# Rapid groundwater decline and some cases of recovery in aquifers globally

Scott Jasechko[1,11]✉, Hansjörg Seybold[2,11], Debra Perrone[3,11], Ying Fan[4], Mohammad Shamsudduha[5], Richard G. Taylor[6], Othman Fallatah[7,8] & James W. Kirchner[2,9,10]

Groundwater resources are vital to ecosystems and livelihoods. Excessive groundwater withdrawals can cause groundwater levels to decline[1–10], resulting in seawater intrusion[11], land subsidence[12,13], streamflow depletion[14–16] and wells running dry[17]. However, the global pace and prevalence of local groundwater declines are poorly constrained, because in situ groundwater levels have not been synthesized at the global scale. Here we analyse in situ groundwater-level trends for 170,000 monitoring wells and 1,693 aquifer systems in countries that encompass approximately 75% of global groundwater withdrawals[18]. We show that rapid groundwater-level declines (>0.5 m year⁻¹) are widespread in the twenty-first century, especially in dry regions with extensive croplands. Critically, we also show that groundwater-level declines have accelerated over the past four decades in 30% of the world's regional aquifers. This widespread acceleration in groundwater-level deepening highlights an urgent need for more effective measures to address groundwater depletion. Our analysis also reveals specific cases in which depletion trends have reversed following policy changes, managed aquifer recharge and surface-water diversions, demonstrating the potential for depleted aquifer systems to recover.

Groundwater is the primary water source for many homes, farms, industries and cities around the globe. Unsustainable groundwater withdrawals and changes in climate can cause groundwater levels to fall[1–10], making groundwater resources less accessible[17]. Global maps of groundwater storage trends are available[7] from the Gravity Recovery and Climate Experiment (GRACE) satellites, although at a resolution that is too coarse (>150,000 km²; ref. 19) to detect local changes and inform local management. Measuring multidecadal groundwater-level declines and managing their consequences—including seawater intrusion[11], land subsidence[12,13], streamflow depletion[14–16] and wells running dry[17]—requires in situ groundwater-level measurements from networks of monitoring wells. Such monitoring-well networks have been used at local and regional scales to estimate groundwater recharge[20,21], characterize streamflow depletion[14], evaluate the risk of wells running dry[17] and test whether surface-water diversions[22,23] or market and policy interventions[24] have succeeded in slowing groundwater losses. However, in situ groundwater-level observations have rarely been analysed at the global scale because we lack a global compilation of in situ groundwater-level time series.

Here we compile and analyse in situ measurements of groundwater-level trends in about 170,000 monitoring wells. The measurements provide new constraints on the prevalence of rapid and accelerating groundwater-level declines and their correlation with land use and climatic drivers. Furthermore, the measurements highlight individual cases in which groundwater levels have recovered following policy changes[25] and inter-basin water transfers[26].

## Local hotspots of groundwater-level changes

We compiled and quality-controlled groundwater-level time series in monitoring wells from more than 40 countries (see Methods and Supplementary Notes 1 and 2). We calculated twenty-first century trends in depth to groundwater level for about 170,000 monitoring wells with time series that span at least 8 years using Theil–Sen robust regression (Fig. 1; analyses based on alternative regression techniques and on different quality-control thresholds yield similar results; see Supplementary Notes 3, 4, 5 and 6). Positive Theil–Sen slopes indicate deepening groundwater levels (red points in Fig. 1). Trends in groundwater levels often differ substantially from well to well, and local hotspots of groundwater decline can be found even in regions in which nearby groundwater levels are stable or rising, and vice versa (Fig. 1), highlighting the importance of analysing groundwater-level trends at the scales defined by the boundaries of individual aquifer systems.

To evaluate aquifer-scale groundwater-level trends, we manually delineated the boundaries of 1,693 aquifer systems—areas underlain by one or more aquifers—using maps and descriptions from 1,236 local and regional studies (see Methods and Supplementary Note 7). We calculated aquifer-scale groundwater-level trends as the median of the

[1]Bren School of Environmental Science & Management, University of California, Santa Barbara, Santa Barbara, CA, USA. [2]Department of Environmental Systems Sciences, ETH Zürich, Zürich, Switzerland. [3]Environmental Studies Program, University of California, Santa Barbara, Santa Barbara, CA, USA. [4]Department of Earth and Planetary Sciences, Rutgers University, New Brunswick, NJ, USA. [5]Institute for Risk and Disaster Reduction, University College London, London, UK. [6]Department of Geography, University College London, London, UK. [7]Department of Nuclear Engineering, Faculty of Engineering, King Abdulaziz University, Jeddah, Saudi Arabia. [8]Center for Training and Radiation Protection, Faculty of Engineering, King Abdulaziz University, Jeddah, Saudi Arabia. [9]Swiss Federal Research Institute WSL, Birmensdorf, Switzerland. [10]Department of Earth and Planetary Science, University of California, Berkeley, Berkeley, CA, USA. [11]These authors contributed equally: Scott Jasechko, Hansjörg Seybold, Debra Perrone. ✉e-mail: jasechko@ucsb.edu

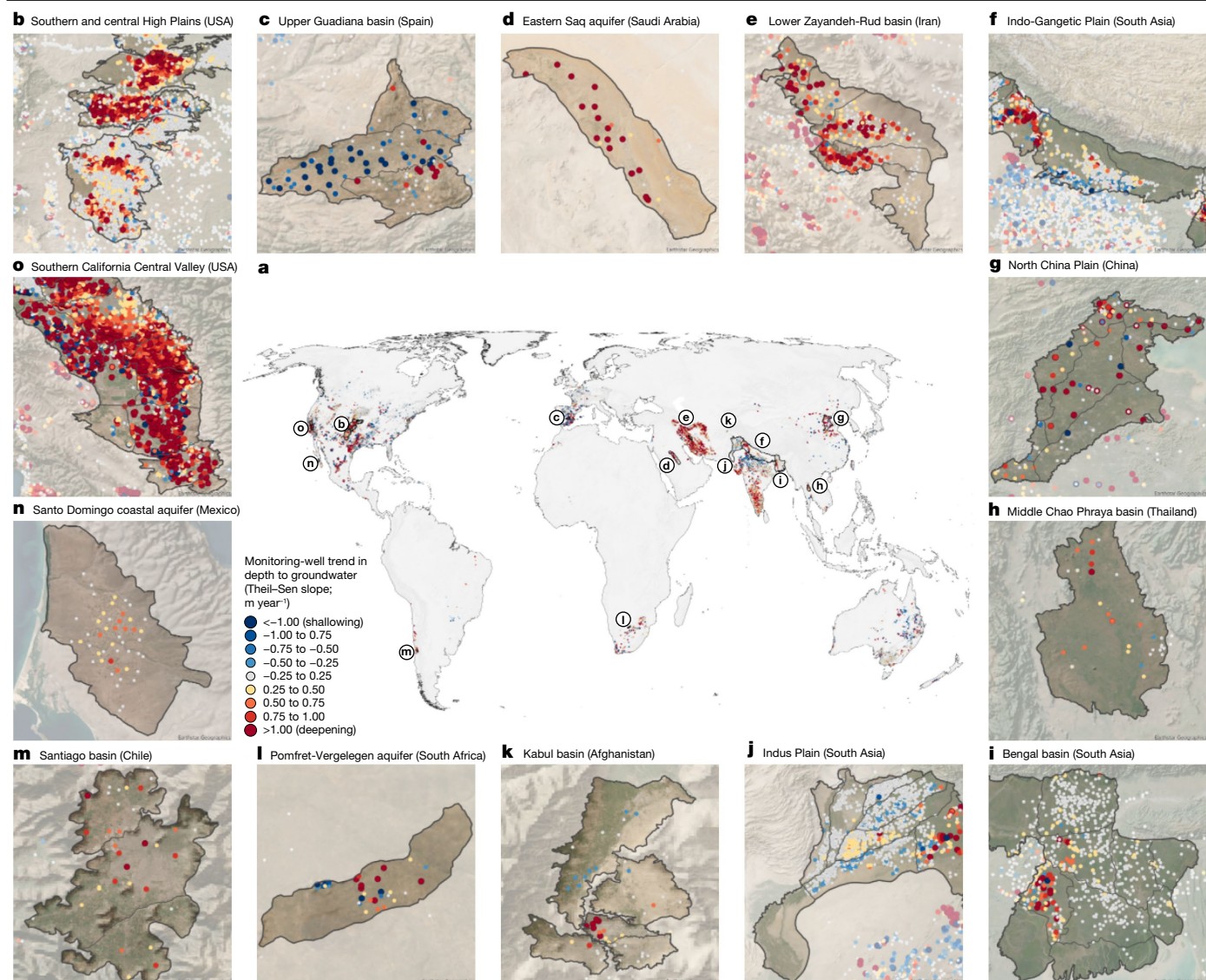

**Fig. 1 | Twenty-first century groundwater-level trends in globally distributed monitoring wells.** Each point represents one monitoring well, coloured to represent the Theil–Sen trend of annual median groundwater levels during the twenty-first century. Blue and red points indicate shallowing and deepening, respectively, of groundwater levels over time, with darker colours indicating faster rates. **a**, Spatial distributions of groundwater-level trends in globally distributed monitoring wells. **b–o**, Regional maps illustrating the substantial spatial variability in groundwater-level trends. Supplementary Notes 16 and 17 show monitoring wells and their groundwater-level trends at subcontinental scales (Supplementary Note 16) and in 207 individual aquifer systems (Supplementary Note 17). Background imagery shown in **b–o** from https://www.arcgis.com/home/item.html?id=10df2279f9684e4a9f6a7f08febac2a9.

Theil–Sen slopes of all monitoring wells located within each aquifer system (Fig. 2). Most aquifer-scale groundwater-level trends range from −0.1 to 0.9 m year⁻¹ (5th to 95th percentiles), in which negative values represent shallowing groundwater levels and positive values indicate deepening groundwater levels.

Groundwater levels became deeper over time at rates exceeding 0.1 m year⁻¹ in 36% of the aquifer systems (617 of 1,693) and exceeding 0.5 m year⁻¹ in 12% (210) of them. Aquifer systems that exhibit groundwater-level deepening and are too small to be detected by GRACE satellite observations (for example, southeastern Spain) highlight the value of in situ groundwater-level measurements to complement global-scale insights[5,7,9,19] made possible by the GRACE (see Methods and Supplementary Note 8).

Groundwater levels became shallower over time faster than −0.1 m year⁻¹ in 6% of the aquifer systems (97 of 1,693) and faster than −0.5 m year⁻¹ in only 1% (13) of them. Some groundwater-shallowing trends may be explained by reductions in groundwater withdrawals, land-cover changes, managed aquifer recharge projects (for example, in Arizona's East Salt River basin[22]) and inter-basin surface-water transfers (for example, the Wanjiazhai water diversion to China's Taiyuan basin[26]).

## Accelerating groundwater-level declines

To place twenty-first century groundwater-level declines into context, we compared them with groundwater-level trends during the late twentieth century (1980–2000); this analysis was possible in 542 of the 1,693 delineated aquifer systems (see Methods and Supplementary Note 9).

In 30% of these aquifer systems, groundwater-level declines accelerated, with early twenty-first century groundwater-level declines outpacing those of the late twentieth century (the red points in Fig. 3a; see the red time series in Fig. 3b and Extended Data Fig. 1 for illustrative examples). These cases of accelerating groundwater-level declines are more than twice as prevalent as one would expect from random fluctuations in the absence of any systematic trends in either time period

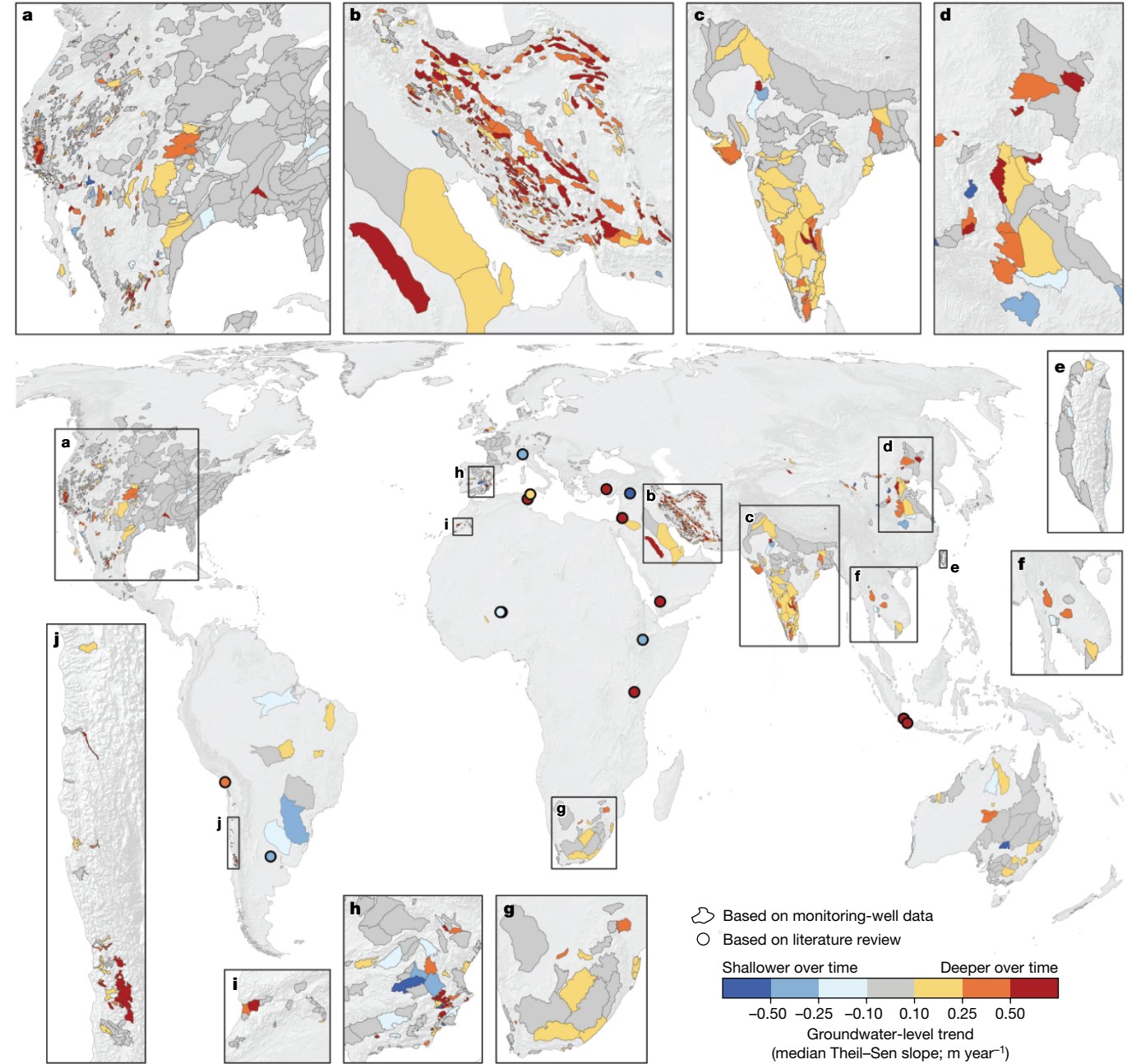

**Fig. 2 | Twenty-first century trends in depth to groundwater in 1,693 globally distributed aquifer systems.** Each polygon represents one aquifer system. Dark grey represents aquifer systems in which groundwater levels have been relatively stable (median Theil–Sen slope between −0.1 and 0.1 m year⁻¹). Yellow, orange and red represent aquifer systems in which groundwater levels became deeper (median Theil–Sen slope >0.1 m year⁻¹). Blue represents aquifer systems in which groundwater levels became shallower (median Theil–Sen slope of <−0.1 m year⁻¹). Darker colours indicate faster rates. Circular points

mark locations for which we lack monitoring-well data but groundwater-level trends have been documented in the literature, with colours indicating the average of the minimum and maximum literature values (Supplementary Note 15). Statistics describing the spatial variability of groundwater-level trends within individual aquifers are presented in Supplementary Note 23. Median Theil–Sen slopes for all 1,693 aquifer systems are tabulated in Supplementary Note 24.

(12.5%; *P*-value < 0.001 by the binomial test). Furthermore, among all cases in which groundwater levels declined in both the late twentieth and early twenty-first centuries, declines in the early twenty-first century outpaced those in the late twentieth century much more often than one would expect by chance (163 red points versus 107 orange points in Fig. 3a; *P*-value < 0.001 by the sign test). If we exclude cases in which groundwater-level trends changed by less than 0.1 m year⁻¹ between these two periods (that is, considering only points lying outside the grey diagonal band in Fig. 3a), we find that accelerating declines (red points) outnumber decelerating declines (orange points) by a ratio of 5:2 (*P*-value < 0.001 by the sign test). In summary, groundwater-level declines have accelerated in a substantial share of the analysed aquifer systems.

To test for a potential statistical relationship between accelerating groundwater-level declines and climate variability, we analysed precipitation rates over the past four decades (Supplementary Note 10). We show that most (>80%) of the aquifer systems exhibiting accelerating groundwater-level declines also experienced a decline in precipitation

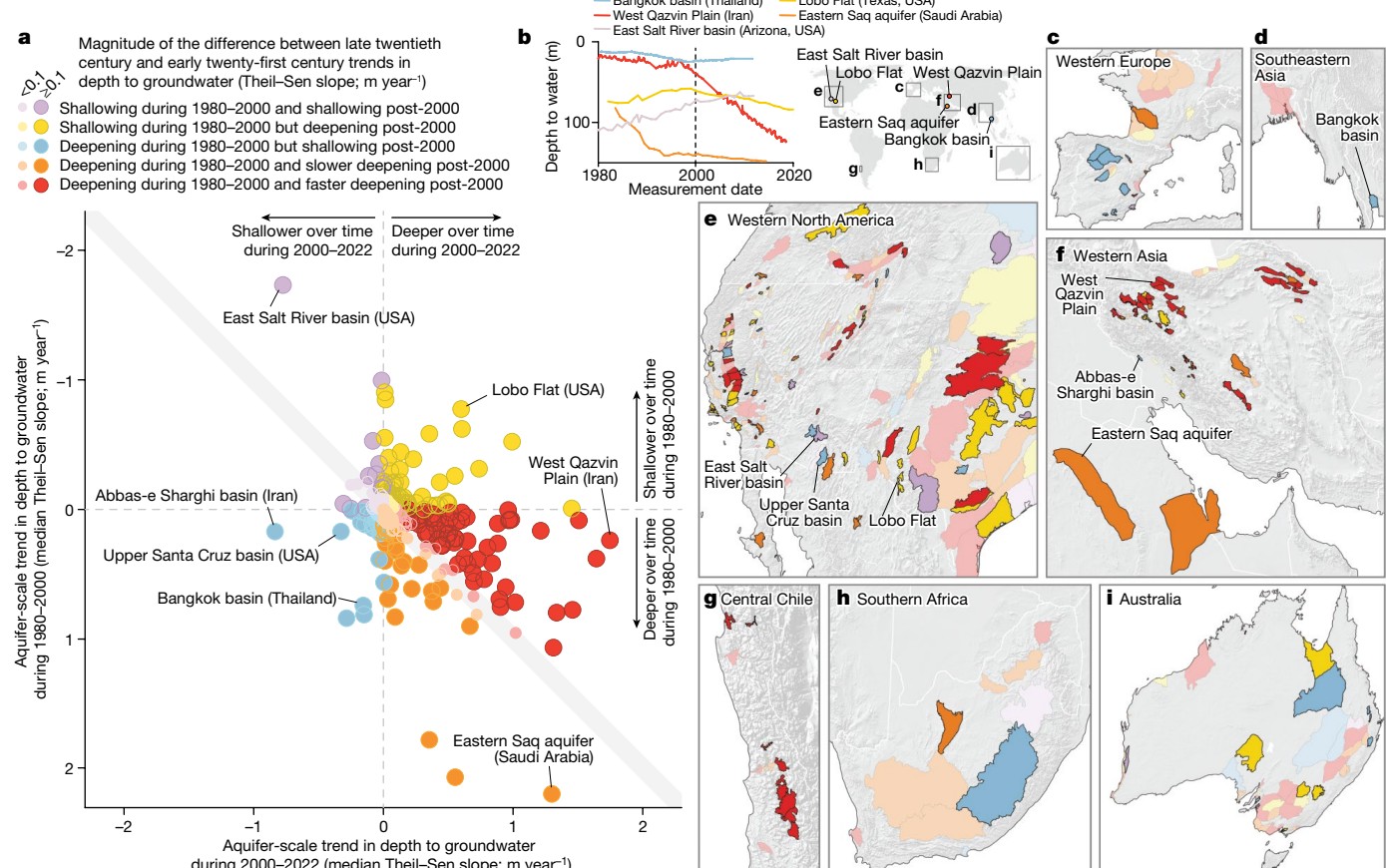

**Fig. 3 | Comparison of aquifer-scale trends in depth to groundwater during the late twentieth and early twenty-first centuries. a**, Scatter plot of aquifer-scale trends (median Theil–Sen slopes) during 2000–2022 (*x*-axis values) and during 1980–2000 (*y*-axis values). The colour of each point indicates one of the following categories of trends: (1) groundwater levels became shallower during 1980–2000 and continued to become shallower (purple points); (2) groundwater levels became shallower during 1980–2000 but have since become deeper (yellow points); (3) groundwater levels became deeper during 1980–2000 but have since become shallower (blue points); (4) groundwater levels became deeper during 1980–2000 and continued to become deeper but at a slower rate (that is, decelerated deepening; orange

points); and (5) groundwater levels became deeper during 1980–2000 and continued to become deeper at a faster rate (that is, accelerated deepening; red points). The intensity of each colour scales with the absolute value (that is, magnitude) of the difference between the late twentieth and early twenty-first century trends in groundwater level (see legend). **b**, Examples of groundwater-level time series illustrating each of our five categories (see legend). **c**–**i**, Maps of aquifer systems categorized by their late twentieth and early twenty-first century trends in groundwater levels (colours correspond to categories in the legend). For an expanded version of this figure, see Supplementary Note 9.

over time (that is, lower average annual precipitation during the early twenty-first century than in the late twentieth century). Declines in precipitation can cause groundwater levels to fall as a result of both indirect impacts (for example, increased groundwater abstractions during droughts) and direct impacts (for example, reduced recharge rates during droughts; see ref. 27). Our finding—that early twenty-first century precipitation rates were lower than in the late twentieth century in most aquifer systems exhibiting accelerating groundwater-level declines—highlights a potential link between decadal-scale climate variability and accelerating groundwater-level declines. Accelerating groundwater-level declines, regardless of their potential drivers, are likely to also accelerate the consequences of those declines, including land subsidence[12,13] and wells running dry[17].

## Slowing and reversing groundwater-level declines

Many previous studies[1–10] have highlighted groundwater losses, but the potential for slowing or reversing these losses has received less attention. Our analysis of groundwater levels suggests that long-term groundwater losses are neither universal nor inevitable. Specifically, in half (49%) of the 542 aquifer systems in our analysis, groundwater-level

declines have decelerated (that is, slowed; orange in Fig. 3; 20%) or reversed (blue in Fig. 3; 16%), or groundwater levels have continued to rise (purple in Fig. 3; 13%).

In 20% of the aquifer systems, groundwater-level deepening has decelerated, as late twentieth century groundwater declines continued in the early twenty-first century, but at a slower rate (the orange points in Fig. 3a; see orange time series in Fig. 3b and Extended Data Fig. 2 for illustrative examples). Although these cases are outnumbered by those for which groundwater declines have accelerated, they demonstrate that it is possible to slow, and potentially even reverse, groundwater-level declines. For example, our analysis shows marked deceleration of groundwater-level deepening in the Eastern Saq aquifer of Saudi Arabia, possibly owing partly to policies designed to reduce agricultural water demands[28] (see labelled orange point in Fig. 3a, which corresponds to the orange line in Fig. 3b).

In 16% of the aquifer systems, groundwater level declines reversed—defined as cases in which groundwater levels declined in the late twentieth century but rose in the early twenty-first century (the blue colours in Fig. 3; see blue time series in Fig. 3b and Extended Data Fig. 3 for examples). For example, in the Bangkok basin (Thailand), groundwater levels deepened during the late twentieth century but shallowed in

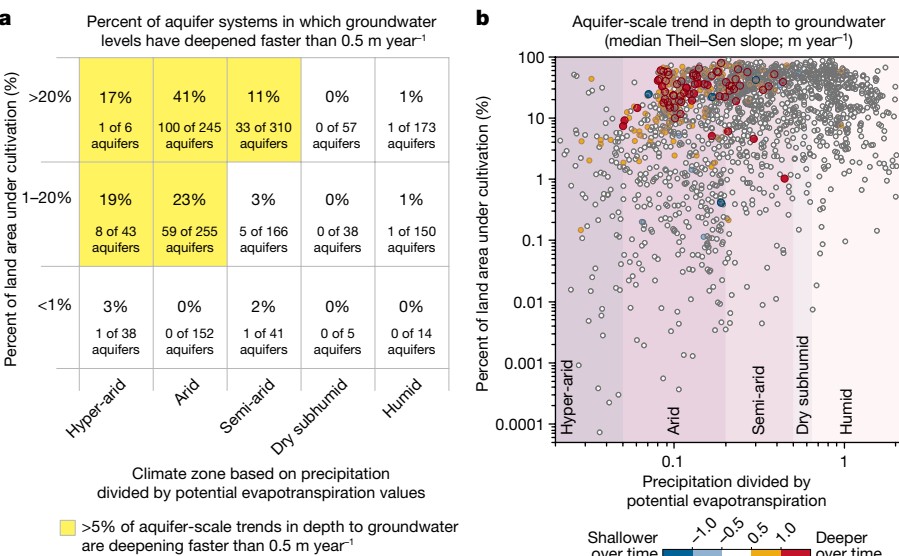

**Fig. 4 | Twenty-first century aquifer-scale trends in depth to groundwater in the context of climate and cultivation. a**, The percentage of aquifer systems with rapidly deepening groundwater (median Theil–Sen slope steeper than 0.5 m year[−1]) when categorized by climate conditions and cropland prevalence. Aquifer systems with rapidly deepening groundwater are most common in hyper-arid, arid and semi-arid climate zones (see categories on the *x* axis) and where a larger proportion of land is under cultivation (see categories on the *y* axis). **b**, Scatter plot of aquifer-scale average annual precipitation divided by potential evapotranspiration[39], and the percentage of land area under cultivation[40] (estimated for the year 2015). The colour of each point

represents the twenty-first century aquifer-scale groundwater-level trend (median Theil–Sen slope). Blue and red points indicate shallowing and deepening, respectively, of groundwater, with darker colours indicating faster rates. Background shades represent climate zones classified by annual precipitation divided by potential evapotranspiration (that is, *x*-axis values). Several aquifer systems are absent from this plot because either no land is under cultivation (incompatible with the log scale of the *y* axis) or precipitation divided by evapotranspiration values fall outside the shown range of *x*-axis values. For alternative versions of this figure showing these aquifer systems, see Supplementary Note 11.

the early twenty-first century (see labelled blue point in Fig. 3a); this reversal has been attributed[25] to regulatory measures (groundwater pumping fees and licensing of wells). Another example is Iran's Abbas-e Sharghi basin, in which twentieth century groundwater-level declines were reversed by the diversion of water to the basin from the Kharkeh Dam[29]. In other areas, groundwater deepening has been reversed following the implementation of managed aquifer recharge projects[22] (for example, west of Tucson, Arizona; Extended Data Fig. 3). Recharge projects are sometimes only viable where excess surface waters are available, emphasizing the importance of coordinating groundwater and surface-water management[30]. Nevertheless, these examples illustrate that interventions of sufficient scope and scale can reverse declining groundwater trends.

In a further 13% of the aquifer systems, groundwater levels rose in both the late twentieth and the early twenty-first centuries (purple colours in Fig. 3; see purple time series in Fig. 3b and Extended Data Fig. 4 for examples). Some of these cases indicate that aquifers that were heavily exploited before 1980 are recovering. Aquifer recovery can potentially ameliorate the consequences of groundwater pumping (for example, land subsidence[31]). In other cases, however, rising groundwater levels can be problematic. For example, rising groundwaters can lead to flooding of coastal cities[32], waterlogging of farmlands[33] and salinization of groundwaters and soils[34]. Rising groundwater levels may be driven by reductions in groundwater withdrawals[25] or increases in recharge rates owing to land clearing[35,36], irrigation[33] or managed aquifer recharge[37]. Our aquifer-scale groundwater-level trends can help predict where rising groundwater levels may pose challenges.

Although these examples illustrate that groundwater declines can be slowed or reversed, several caveats must be kept in mind. In general, rates of groundwater-level shallowing are much slower than rates of groundwater-level decline. Of the aquifer systems in Fig. 3 with rising twenty-first century groundwater levels (blue and purple points), only 6% are rising faster than −0.2 m year[−1]. By contrast, of the

aquifer systems with deepening twenty-first century groundwater levels (yellow, red and orange points in Fig. 3), 25% are falling faster than 0.2 m year[−1]. Furthermore, across these aquifer systems, the average rate of twenty-first century deepening (0.2 m year[−1]) exceeds the average rate of shallowing (−0.05 m year[−1]) by a factor of four. Thus, rapidly rising groundwater levels are rare, but they demonstrate that aquifer recovery is possible, especially following policy changes[25], managed aquifer recharge[37] and inter-basin surface water-transfers[26].

## Groundwater declines in cultivated drylands

Many of the aquifer systems with declining twenty-first century groundwater levels (Fig. 2) underlie drylands, defined[38] as areas in which average precipitation divided by potential evapotranspiration is less than 0.65. Rapidly deepening groundwater levels (faster than 0.5 m year[−1]) are found in 11%, 24% and 8% of aquifers in climate zones classified[38] as hyper-arid, arid and semi-arid, respectively. Notably, aquifer systems with rapidly deepening groundwater levels are virtually absent (<1%) in humid and dry subhumid climate zones. Our 1,693 aquifer-scale groundwater-level trends exhibit a moderately strong rank correlation with precipitation divided by potential evapotranspiration[39] (Spearman $\rho = -0.40$, *P*-value < 0.001; Supplementary Note 11 and Methods), implying that groundwater deepening is more common in drier climates (Fig. 4). As well as rapid groundwater-level declines, we also find that accelerating groundwater-level declines are more common in drier climates, especially underlying cultivated lands (Supplementary Note 9), probably reflecting greater reliance on groundwater for irrigation.

Irrigation is estimated to account for 70% of global groundwater withdrawals[18]. A lack of high-resolution, ground-truthed data quantifying groundwater withdrawals for irrigation precludes statistical tests of their correlation with groundwater-level changes over time. However, using high-resolution global land cover data[40], we can test for statistical

relationships between land-use patterns and groundwater trends (Fig. 4). Aquifer systems with rapidly deepening groundwater levels (>0.5 m year$^{-1}$) are relatively common (17%) where more than one-fifth of the land surface is cultivated, but are virtually absent (0.8%) where cultivation accounts for <1% of the land surface. Across the 1,693 aquifer systems, rates of groundwater-level deepening are significantly correlated with the proportion of land under cultivation[40] (Spearman $\rho$ = 0.17, $P$-value < 0.001; Fig. 4). This statistical relationship becomes stronger when we account for the correlation between cultivation and climatic aridity (partial rank correlation coefficient = 0.32, $P$-value < 0.001; see Supplementary Note 11). Our analyses demonstrate that rapid groundwater declines are most common in cultivated drylands.

Groundwater losses from dryland aquifers pose management challenges. Aquifer recharge is typically slow in drylands[41], meaning that depleted dryland aquifers will generally take longer to recover than aquifers in wetter climates[42], except where recharge rates are artificially increased (for example, seepage from unlined canals in the Indus basin[33]). Moreover, groundwater is often the sole source of perennial drinking water for communities in drylands. As groundwater levels become deeper, shallower wells can run dry[17], compromising local water access. Even where groundwater levels remain stable, groundwater withdrawals can deplete the flow of nearby streams by reducing natural seepage of groundwater to rivers, or even inducing streamwater leakage into underlying aquifers (see discussion of 'capture' by ref. 43). Indeed, leakage from surface waters may replenish pumped aquifers and stabilize groundwater levels at the expense of streamflow. The prevalence of rapid and accelerating groundwater declines in cultivated drylands suggests that, even if management strategies are in place, they have often been insufficient—either in concept or in implementation—to slow or reverse groundwater depletion.

## Depleting and recovering groundwater resources

Our analysis of groundwater-level measurements demonstrates that: (1) groundwater levels are declining rapidly (>0.5 m year$^{-1}$) in many regions (Fig. 2); (2) groundwater declines are accelerating in many aquifer systems around the world (Fig. 3); and (3) both rapid and accelerating groundwater declines are particularly evident in aquifers underlying cultivated drylands (Fig. 4 and Supplementary Notes 9 and 11). Our analysis also identifies cases in which late twentieth century groundwater declines have been reversed in the early twenty-first century (blue points in Fig. 3). However, cases of rapidly rising groundwater levels remain outnumbered by cases of rapidly deepening groundwater levels.

Our results indicate that twenty-first century realities—including climatic trends, hydrogeologic conditions, groundwater withdrawal rates, land uses and management approaches—have resulted in widespread, rapid and accelerating groundwater-level declines. Nevertheless, the compiled in situ observations also capture numerous cases in which declines in groundwater levels have slowed, stopped or reversed following intervention (for example, implementation of regulatory measures[25]). Although our work represents the most extensive analysis of groundwater-level monitoring records so far, it does not cover the globe (see Methods section entitled 'Limitations'). Further, analysed monitoring wells do not represent a randomized sample of global wells and we are only able to analyse groundwater level trends where monitoring data are available. Global maps of groundwater storage changes from GRACE satellite observations[7] suggest that groundwater stores are declining in some regions in which monitoring data are not publicly available and, thus, cannot be evaluated here. GRACE data are also important for characterizing impacts of climate change and variability[9,19,44–46] and evaluating global hydrologic models[47]. Evaluating such models is important because they are widely used to estimate groundwater depletion (see ref. 6 and Table 3 in ref. 48). Our compilation of monitoring-well data could facilitate future efforts to reconcile GRACE-based, model-based and piezometric-based groundwater time series (see refs. 49,50). Combining these diverse data products—and thus exploiting both the high spatial resolution of monitoring-well networks and the global coverage of GRACE[7,9,19] and hydrologic models[2,3,6,16,48]—may yield new insights into the causes, consequences and spatial patterns of groundwater depletion.

Groundwater depletion can threaten ecosystems and economies. Specifically, groundwater depletion can damage infrastructure through land subsidence[12,13], impair fluvial ecosystems through streamflow depletion[14–16], jeopardize agricultural productivity[51] and compromise water supplies as wells run dry[17]. Our methodologically consistent analysis of groundwater-level trends across 1,693 globally distributed aquifer systems demonstrates widespread, rapid and accelerating twenty-first century groundwater-level declines, particularly in cultivated drylands.

Our analysis also documents cases for which groundwater declines have slowed or reversed after: (1) the implementation of groundwater policies; (2) the alleviation of groundwater demand by means of surface-water transfers; or (3) the addition of groundwater storage following managed aquifer recharge projects. To address the growing problem of global groundwater depletion, these kinds of success stories would need to be replicated in dozens of aquifer systems with declining groundwater levels. Thus, our analysis illustrates the potential for depleted aquifers to recover, while demonstrating how much work remains to be done to protect groundwater resources. By documenting global hotspots of groundwater-level decline and recovery, this analysis can inform efforts to address rapid and accelerating groundwater depletion.

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

## Methods

### Delineating global aquifer systems based on literature review of local studies

For each country in our study, we consulted published accounts of local-scale studies[52–1288] (Supplementary Note 7) to delineate 1,693 study areas, each underlain by one or more aquifers and/or low-permeability geologic formations that are, collectively, referred to as an 'aquifer system'. Each aquifer system was delineated by consulting maps and reading descriptions within local-scale reports. Specific steps applied to delineate the boundaries of each aquifer system are detailed in Supplementary Note 7.

### Downloading groundwater-level data

Our study focuses on more than 40 countries for which we compiled monitoring-well data. We analysed groundwater-level time series derived from numerous data repositories (dataset-specific details are available in Supplementary Note 1; some of these datasets are described in refs. 1289–1297). The compiled groundwater-level databases span different time intervals and have different measurement frequencies (see heat map plot and global maps showing monitoring-well time series durations and measurement frequencies in Supplementary Note 12).

### Quality controlling groundwater-level time series

We completed five pre-processing steps before analysing groundwater-level data. First, we identified replicate groundwater-level measurements, defined as cases in which an identical measurement date and an identical groundwater-level measurement were reported from the same monitoring well; in these cases, we retain only one of these replicates. Second, we identified cases in which several groundwater-level measurements from the same monitoring well reported identical measurement dates. In these cases, we calculated the median among all groundwater-level measurements sharing the same measurement date and the adjacent points in the time series (that is, the median of the group of measurements with identical dates and the measurements immediately preceding and following the same-date measurements); we then kept only the single water-level measurement whose value was closest to this calculated median (Supplementary Note 2). Third, we excluded extreme values of depth to groundwater (that is, >1,000 m and <−1,000 m) and implausibly high groundwater elevations (that is, >8,000 m above sea level). Fourth, we excluded groundwater-level measurements with values of '999', '−9,999' or '0', because some databases used these values as a code for missing measurements (see figures in Supplementary Note 2). Fifth, we excluded outlier values detected by a machine-learning algorithm[1298] (based on an additive regression model[1299]; for details, see Supplementary Note 2.1). This algorithm was applied to each monitoring well with more than 15 groundwater-level measurements, yielding a prediction for each time step and its 99% confidence interval. We defined points to be outliers and excluded them if they fell outside the range defined by the predicted groundwater level ±0.75 times this confidence interval. If a large number of measurements within a monitoring well's time series were classified as outliers, we excluded the entire time series from our analysis (in which a 'large number of measurements' is defined as cases for which there were at least five outliers identified by the machine-learning algorithm and for which these outliers comprise >1% of all measurements in the time series; for visualization, see schematics in Supplementary Note 2). Among the approximately 170,000 monitoring wells presented in Fig. 1, only about 12% had one or more outlier points removed by means of this machine-learning approach, highlighting that this machine-learning approach affected only a small proportion of consulted monitoring wells. Furthermore, a comparison of aquifer-scale trends in depth to groundwater with versus without the use of a machine-learning-based outlier-exclusion procedure suggests that our machine-learning approach had no substantial influence on our findings (see Supplementary Note 13).

### Flagging groundwater-level measurements based on rapid increases or decreases

After excluding potential outliers (through the steps outlined in the previous paragraph), we calculated each monitoring well's annual median groundwater levels for each calendar year with at least one measurement. We then visually inspected plots of annual median groundwater levels over time. On visual inspection, we noted that a small number of monitoring wells show 'spikes' in their annual groundwater-level time series, in which a 'spike' is defined as a high-magnitude (absolute value > 20 m year$^{-1}$) groundwater-level change followed directly by another high-magnitude groundwater-level change in the opposite direction (for example, a high-magnitude groundwater-level deepening trend between two adjacent points in the time series, directly followed by a high-magnitude groundwater-level shallowing trend between two adjacent points). We flagged these data points as potentially suspect. The first or last point in each time series was also flagged if it differed by more than 20 m year$^{-1}$ from the second or next-to-last point. We compared groundwater-level trends with and without these flagged points and observed only trivial differences (Supplementary Note 5: 'Similar aquifer-scale trends obtained with and without flagged measurements'). The results presented in the main text (for example, Fig. 1) derive from annual median groundwater-level time series that exclude the flagged measurements.

### Statistical analyses of twenty-first century groundwater-level trends (Figs. 1 and 2)

To evaluate groundwater-level trends since the year 2000, we excluded all previous measurements. Next, we excluded all monitoring wells for which the earliest and most recent annual medians were separated by fewer than 8 years. We calculated trends in annual median groundwater levels for all monitoring wells that met these minimum criteria for analysis (for a similar method, see ref. 1288).

Some data sources report groundwater levels as elevations (metres above sea level) and others report them as depth to groundwater (metres below the land surface, or below the top of the well). In cases in which both were reported, we used the depth to groundwater data. If groundwater levels were only reported as elevations, we reversed the signs of the calculated trends, to obtain trends in depth to groundwater.

Our results in the main text are based on Theil–Sen regression slopes[1300,1301] but we also applied several different regression techniques, including ordinary least squares, iteratively reweighted least squares[1302–1304] and RANSAC (or random sample consensus)[1305], which yielded comparable results (Supplementary Note 3; for non-parametric regression techniques, see Supplementary Note 4 and ref. 1306). We present our results as trends in depth to groundwater, meaning that positive slopes represent groundwater levels becoming deeper over time. We calculated an aquifer-scale groundwater-level trend for each aquifer system by taking the median of the Theil–Sen slopes of all monitoring wells within its boundaries (Fig. 2).

### Comparing groundwater-level trends between the late twentieth and early twenty-first centuries (Fig. 3)

To contextualize twenty-first century trends in depth to groundwater, we identified monitoring wells with sufficient data during two periods: the late twentieth century (1980–2000) and the early twenty-first century (2000–2022). Here well time series are 'sufficient' if their earliest and latest annual medians are separated by at least 8 years within a given time interval (that is, 1980–2000 or 2000–2022). There are 45,911 monitoring wells in the compiled dataset with sufficient groundwater-level data for trend analyses during both periods. For these monitoring wells, we calculated Theil–Sen trends in depth to groundwater for the late twentieth century. Next, we grouped monitoring wells located within

the same aquifer system and calculated aquifer-scale trends for the late twentieth century (medians of the Theil–Sen slopes for all wells in each system; that is, $y$-axis values presented in Fig. 3a). Only aquifer systems with at least five monitoring wells for both time periods (1980–2000 and 2000–2022) satisfying the aforementioned requirements were used to compare late twentieth century and early twenty-first century trends in depth to groundwater. Last, we assigned each aquifer system to one of five categories based on its late twentieth century and early twenty-first century trends in depth to groundwater: (1) groundwater levels became shallower during 1980–2000 and continued to become shallower (purple points in Fig. 3a); (2) groundwater levels became shallower during 1980–2000 but have since become deeper (yellow points in Fig. 3a); (3) groundwater levels became deeper during 1980–2000 but have since become shallower (blue points in Fig. 3a); (4) groundwater levels became deeper during 1980–2000 and continued to become deeper but at a slower rate (that is, decelerated deepening; orange circles in Fig. 3a); and (5) groundwater levels became deeper during 1980–2000 and continued to become deeper at a faster rate (that is, accelerated deepening; red circles in Fig. 3a). Further details are available in Supplementary Note 9.

**Geospatial analysis of potential explanatory variables (Fig. 4)**
To test for statistical relationships between the spatial distributions of environmental conditions and groundwater-level trends, we downloaded two geospatial datasets: (1) long-term mean annual precipitation divided by potential evapotranspiration (the 'CGIAR-CSI Global-Aridity and Global-PET Database'; ref. 39) and (2) the proportion of land area under cultivation (estimated for the year 2015; ref. 40). Next, we averaged each of these geospatial datasets over each of the 1,693 aquifer systems (Fig. 4). We calculated rank correlations between twenty-first century aquifer-scale groundwater-level trends and both of the potential explanatory variables (namely, (1) long-term mean annual precipitation divided by potential evapotranspiration and (2) the proportion of land area under cultivation). We also used multiple regression on the rank transforms of these explanatory variables to account for their covariation (Supplementary Note 11).

**Limitations**
Our analyses are based on the best available measurements but nonetheless have limitations. Here we detail some of these limitations and evaluate how some may affect our main conclusions.
- Although we have used several steps, as outlined above, to detect and remove outliers, we cannot independently verify the accuracy of all groundwater-level time series. Nevertheless, our analysis is based on several layers of robust estimation (for example, Theil–Sen regression on annual medians), minimizing its sensitivity to unreliable data.
- Groundwater-level data from individual monitoring wells span different time intervals and have different measurement frequencies, as detailed in Supplementary Note 12. Furthermore, about 41% of the analysed monitoring wells have discontinuous time series of annual groundwater levels (for which 'discontinuous' time series are defined as those lacking a groundwater-level measurement for at least one of the calendar years that lie between the earliest and most recent twenty-first century groundwater-level measurements; for an example of a discontinuity in an annual groundwater-level time series, see Supplementary Fig. 3c).
- We could not obtain groundwater-level data for many countries around the globe and our conclusions are only directly applicable where we have data. GRACE satellite data[1307–1311] suggest that groundwater storage has declined in some of the areas in which we lack monitoring-well data (Supplementary Note 8). Further, simulation results from a global model suggest that substantial groundwater depletion may have occurred in some of the countries in which we lack monitoring-well data, so groundwater-level deepening may be even more widespread than our results indicate (refs. 16,1312; Supplementary Note 14). We reviewed published and grey literature[20,427,802,1282,1313–1356] to obtain groundwater-level trends for some of the countries in which we lack monitoring-well data (that is, point data in Fig. 2; details available in Supplementary Note 15).

- We highlight that monitoring wells are not distributed evenly across each aquifer system. Consequently, some locations within aquifer systems are not captured by compiled monitoring-well data (see discussion of Dhaka (Bangladesh) in Supplementary Note 15). The aquifer-scale trends that we present in the main text (Figs. 2–4) do not provide insights into the spatial patterns of groundwater-level trends within individual aquifer systems. The high variability in monitoring-well densities within aquifer systems, as well as the substantial variability in groundwater-level trends even among co-located monitoring wells, are presented in a suite of maps for individual aquifer systems in Supplementary Notes 16 and 17. Specifically, our analysis demonstrates that groundwater-level trends can vary greatly among wells within individual aquifer systems (Fig. 1 and Supplementary Notes 16 and 17), implying that local-scale groundwater-level declines may be even more widespread than our Fig. 2 suggests (Supplementary Note 18). Some of the variability in groundwater-level trends among co-located wells may be partly explained by differences in the depths of nearby monitoring wells, as shallow and deep aquifers can have different groundwater-level trends (see Supplementary Note 19).
- We stress that groundwater-level trends may differ between deeper and shallower wells (for example, ref. 1357) owing to, for example, differences in the depths of nearby wells used to extract groundwater and differences in storage coefficients between unconfined and confined aquifers (see, for example, refs. 1358,1359). Steep groundwater-level trends—both upward and downward—are more common in deeper wells than in shallower wells, possibly due in part to the greater prevalence of confined conditions at deeper depths (discussion and analyses available in Supplementary Note 19). 2D geologic data are available at the global scale[1360], but an accurate high-resolution 3D hydrogeologic dataset remains unavailable for the globe, meaning that key hydrogeologic conditions (for example, whether the monitoring well captures unconfined versus confined conditions) cannot be ascribed for deep versus shallow wells at the global scale.
- We highlight that our approach to delineating boundaries for individual aquifer systems—although based on local-scale studies—potentially introduces inconsistencies, because local norms for delineating aquifer-system boundaries may differ. Further, some (16%) of the 170,000 monitoring wells fall outside the boundaries of the aquifer systems delineated here and, therefore, are excluded from our aquifer-scale statistical analyses. We present groundwater-level trends for monitoring wells both within and outside aquifer-system boundaries in a series of regional-scale maps (Supplementary Note 16).
- It is possible that some of monitoring-well-based time series may be truncated where the monitoring well itself has run dry (see ref. 1361), possibly excluding monitoring wells located in areas experiencing rapid groundwater depletion. We analysed monitoring-well depths and depth to groundwater data for 72,000 wells and conclude that it is possible that a small proportion of the groundwater-level time series was truncated owing to well desiccation (see Supplementary Note 20). Thus, rapid and accelerating twenty-first century groundwater-level deepening may be even more prevalent than our analysis indicates.
- Our main-text results are based on annual median groundwater levels. However, we acknowledge that trends in depth to groundwater can differ when based on measurements made during specific seasons (for example, long-term trends in pre-monsoon depth to groundwater can differ from long-term trends in post-monsoon depth to groundwater; see ref. 1362). We highlight that trends in season-specific groundwater levels may differ from trends in annual median groundwater levels (as presented in Fig. 1), especially where intra-annual groundwater-level variability is changing over time (for example, time series from the Bengal basin in Supplementary Note 21; see also the time series presented in refs. 21,1363,1364).

- The compiled groundwater-level time series do not allow us to infer trends over longer (for example, centennial-scale) time intervals. In some areas, substantial groundwater-level changes took place long before the four decades that we focus on here. For example, there is evidence[1365,1366] that substantial accumulation occurred during the twentieth century in parts of South Asia and that groundwater levels were much deeper at the start of the twentieth century than they are today (see, specifically, Fig. 3b in ref. 1365). Some aquifer systems in our dataset, for example, may have been heavily depleted during the mid-twentieth century, but have exhibited relatively stable groundwater levels (or even shallowing groundwater-level trends) during the twenty-first century. Given the potential for such cases, we make no claim that stable twenty-first century groundwater levels necessarily imply a lack of previous or continuing disturbance.

- We do not make claims about aquifer-specific drivers behind rapid and accelerating groundwater declines (although we do make note of case studies in the literature that have identified important drivers; for example, ref. 25). We acknowledge that groundwater abstractions can perturb flow systems and, in many cases, deplete aquifers. Many of the aquifer systems exhibiting rapid groundwater-level declines are being accessed by wells, as evidenced by recorded well-completion events throughout the early twenty-first century (Supplementary Note 22; data described in refs. 17,1367–1369) and by regional-scale research[108,1370,1371]. Further, we acknowledge that climate variability and change can have both direct impacts on groundwater levels (such as through changes in groundwater recharge owing to, for example, changes in temporal variability in precipitation) and also indirect impacts on groundwater levels (for example, through changes in groundwater demand in response to climate variability, such as increased groundwater withdrawals during drier time intervals; see ref. 27). Available precipitation data[1372,1373] suggest that most of the aquifer systems characterized as exhibiting accelerating groundwater-level declines (that is, red points in Fig. 3) are situated in areas in which early twenty-first century annual precipitation rates were lower than late twentieth century annual precipitation rates (Supplementary Note 10), highlighting that, at a minimum, we cannot rule out an influence of climate variability (direct or indirect) on groundwater-level changes over time.

## Data availability

Annual groundwater-level data are available for download in all cases for which we have received permission from a database manager to post data (data are available from Zenodo (https://doi.org/10.5281/zenodo.10003697) and CUAHSI HydroShare (https://www.hydroshare.org/resource/da946dee3ada4a67860d057134916553/)); these datasets include groundwater-level data for: Afghanistan[1289], Austria, Belgium, Brazil, Bulgaria, Canada (Alberta, British Columbia, Manitoba, Northwest Territories, Ontario, Prince Edward Island, Saskatchewan, Yukon), China[1290], Croatia, Czech Republic, Denmark, France[1291], Germany, Guam, Ireland, Israel, Italy, Latvia, Lithuania, New Zealand, Norway, Paraguay, Poland, Slovenia, Sweden, Switzerland and the USA (Groundwater Ambient Monitoring and Assessment Program, U.S. Geological Survey's (USGS) National Water Information System and the Texas Water Development Board). The databases for which we have received written permission to post annual groundwater-level data encompass 59% of annual groundwater-level data analysed here (specifically, we received permission to post 66% ($n$ = 4,170,802 of $n$ = 6,314,793) of all annual 'depth to groundwater' data and 18% ($n$ = 190,879 of $n$ = 1,049,502) of all 'groundwater elevation' data). These datasets are specified in Supplementary Table 1 (see column entitled 'Written permission received to post annual groundwater-level data'). Source data for each of the main-text figures are available here. Supplementary tables associated with this work are available at https://doi.org/10.5281/zenodo.10003697. Geospatial data for the 1,693 aquifer systems studied here are available from CUAHSI HydroShare (https://www.hydroshare.org/resource/73834f47b8b5459a8db4c999e6e3fef6/) and Zenodo (https://doi.org/10.5281/zenodo.10003697). Source data are provided with this paper.

## Code availability

Analyses presented here do not depend on specific code; the approach can be reproduced following the procedures described in the Methods section.

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

Geological Survey Water-Resources Investigations Report 2002-4136. https://pubs.usgs.gov/wri/wri024136/wrir024136.pdf (2002).

Scientific Investigations Report 2017-5020. https://pubs.usgs.gov/sir/2017/5020/sir20175020.pdf (2017).

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

828. Morell, I. Acuíferos detríticos costeros. *Hidrogeol. Aguas Subterrán.* **1**, 31–44 (2003).

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

1000. Roques, C., Bour, O., Aquilina, L. & Dewandel, B. High-yielding aquifers in crystalline basement: insights about the role of fault zones, exemplified by Armorican Massif, France. *Hydrol. J.* **24**, 2157–2170 (2016).

1001. Rosário de Jesus, M. Groundwater protection for public water-supply in Portugal. https://unece.org/fileadmin/DAM/env/water/meetings/groundwater01/portugal.pdf (2001).

1002. Rose, T. P., Davisson, M. L., Smith, D. K. & Kenneally, J. M. Isotope hydrology investigation of regional groundwater flow in central Nevada. Hydrologic Resources Management Program and Underground Test Area Operable Unit FY 1997 Progress Report, Ch. 6. https://core.ac.uk/download/pdf/204554577.pdf#page=62 (1998).

1003. Rose, T. P., Davisson, M. L., Hudson, G. B. & Varian, A. R. Environmental isotope investigation of groundwater flow in the Honey Lake Basin, California and Nevada. Department of Energy Report UCRL-ID-127978 ON: DE98051049. https://www.osti.gov/servlets/purl/620597 (1997).

1004. Rostami, A. A., Isazadeh, M., Shahabi, M. & Nozari, H. Evaluation of geostatistical techniques and their hybrid in modelling of groundwater quality index in the Marand Plain in Iran. *Environ. Sci. Pollut. Res.* **26**, 34993–35009 (2019).

1005. Rostkier-Edelstein, D. et al. Towards a high-resolution climatography of seasonal precipitation over Israel. *Int. J. Climatol.* **34**, 1964–1979 (2014).

1006. Rotzoll, K., Gingerich, S. B., Jenson, J. W. & El-Kadi, A. I. Estimating hydraulic properties from tidal attenuation in the Northern Guam Lens Aquifer, territory of Guam, USA. *Hydrol. J.* **21**, 643–654 (2013).

1007. Rouillard, J. & Maréchal, J.-C. in *Sustainable Groundwater Management: A Comparative Analysis of French and Australian Policies and Implications to Other Countries* (eds Rinaudo, J.-D., Holley, C., Barnett, S. & Montginoul, M.) 17–45 (Springer, 2020).

1008. Rupérez-Moreno, C., Pérez-Sánchez, J., Senent-Aparicio, J. & del Pilar Flores-Asenjo, M. The economic value of conjoint local management in water resources: results from a contingent valuation in the Boquerón aquifer (Albacete, SE Spain). *Sci. Total Environ.* **532**, 255–264 (2015).

1009. Rupérez-Moreno, C. et al. Sustainability of irrigated agriculture with overexploited aquifers: the case of Segura basin (SE, Spain). *Agric. Water Manag.* **182**, 67–76 (2017).

1010. Rushton, K. R. & Rao, S. R. Groundwater flow through a Miliolite limestone aquifer. *Hydrol. Sci. J.* **33**, 449–464 (1988).

1011. Rutulis, M. Aquifer maps of southern Manitoba. Manitoba Water Resources Branch map. https://www.gov.mb.ca/water/pubs/maps/water/1986_rutulis_bedrock_aquifers.pdf (1986).

1012. Ruybal, C. J., Hogue, T. S. & McCray, J. E. Assessment of groundwater depletion and implications for management in the Denver Basin Aquifer System. *J. Am. Water Resour. Assoc.* **55**, 1130–1148 (2019).

1013. Ryder, P. Ground Water Atlas of the United States: Segment 4, Oklahoma, Texas. U.S. Geological Survey Hydrologic Investigations Atlas 730-E. https://pubs.usgs.gov/ha/730e/report.pdf (1996).

1014. Saadatmand, A., Noorollahi, Y., Yousefi, H. & Mohammadi, A. Investigation, modeling and analysis of qualitative parameters of groundwater resources in Kurdistan's Kamyaran plain. *Iran. J. Ecohydrol.* **8**, 357–367 (2021).

1015. Sabzevari, Y., Nasrolahi, A. H. & Yonesi, H. A. Investigation of temporal-spatial variations of groundwater resources quality in Borujerd-Dorood Plain. *Irrig. Water Eng.* **11**, 150–167 (2020).

1016. Sadeghfam, S., Hassanzadeh, Y., Nadiri, A. A. & Khatibi, R. Mapping groundwater potential field using catastrophe fuzzy membership functions and Jenks optimization method: a case study of Maragheh-Bonab plain, Iran. *Environ. Earth Sci.* **75**, 545 (2016).

1017. Sadid, N. Surface-groundwater interaction in the Kabul region basin. Afghanistan Research and Evaluation Unit Report. https://reliefweb.int/sites/reliefweb.int/files/resources/2005-E-Surface-groundwater-interaction-in-the-Kabul-region-basin.pdf (2020).

1018. Saeidi, H., Lashkaripour, G. & Ghafoori, M. Evaluation of land subsidence in Kashmar-Bardaskan plain, NE Iran. *Iran. J. Earth Sci.* **12**, 280–291 (2020).

1019. Saffari, A., Jan Ahmadi, M. & Raeati Shavazi, M. Site selection for suitable flood spreadingand artificial feeding through hybrid, AHP-Fuzzy Model Case Study: (Bushkan Plain, Bushehr Province). *Hydrogeomorphology* **1**, 81–97 (2015).

1020. Saffi, M. H. National alarming on groundwater natural storage depletion and water quality deterioration of Kabul City and immediate response to the drinking water crises. Scientific Investigation Report in Afghanistan, DACAAR report (2019).

1021. Saha, D. & Gor, N. A prolific aquifer system is in peril in arid Kachchh region of India. *Groundw. Sustain. Dev.* **11**, 100394 (2020).

1022. Saha, D. & Ray, R. K. in *Groundwater Development and Management* (ed. Sikdar, P. K.) 19–42 (Springer, 2019).

1023. Saha, D., Shekhar, S., Ali, S., Vittala, S. S. & Raju, N. J. Recent hydrogeological research in India. *Proc. Indian Natl Sci. Acad.* **82**, 787–803 (2016).

1024. Sahoo, S., Dhar, A., Kar, A. & Chakraborty, D. Index-based groundwater vulnerability mapping using quantitative parameters. *Environ. Earth Sci.* **75**, 522 (2016).

1025. Sahu, J. K., Das, P. P., Sahoo, H. K., Mohapatra, P. P. & Sahoo, S. Geospatial analysis and hydrogeochemical investigation of a part of southern Mahanadi delta, Odisha, India. *Himal. Geol.* **39**, 92–100 (2018).

1026. Sahu, S., Gogoi, U. & Nayak, N. C. Patterns of groundwater chemistry: implications of groundwater flow and the relation with groundwater fluoride contamination in the phreatic aquifer of Odisha, India. *Arab. J. Geosci.* **13**, 1272 (2020).

1027. Sajil Kumar, P. J. & James, E. J. Identification of hydrogeochemical processes in the Coimbatore district, Tamil Nadu, India. *Hydrol. Sci. J.* **61**, 719–731 (2016).

1028. Sakai, A. Land subsidence due to seasonal pumping of groundwater in Saga Plain, Japan. *Lowl. Technol. Int.* **3**, 25–40 (2001).

1029. Salehabadi, G. The effect of groundwater in plain settlement in Jovin. *Sci. Res. Q. Geogr. Data* **22**, 30–34 (2021).

1030. Salehi, H. & Zeinivand, H. Evaluation and mapping of groundwater quality for rigation and drinking purposes in Kuhdasht region, Iran. *Environ. Resour. Res.* **4**, 75–89 (2016).

1031. Salemi, H. R. et al. Water management for sustainable irrigated agriculture in the Zayandeh Rud Basin, Esfahan Province, Iran. Report by Iranian Agricultural Engineering Research Institute, Esfahan Agricultural Research Center and the International Water Management Institute, Research Report Number 1 (2000).

1032. Salinas Valley Basin Integrated Sustainability Plan. https://svbgsa.org/wp-content/uploads/2019/03/Valley-Wide-Integrated-Sustainability-Plan-optimized.pdf (2020).

1033. Saltel, M. et al. Paleoclimate variations and impact on groundwater recharge in multi-layer aquifer systems using a multi-tracer approach (northern Aquitaine basin, France). *Hydrol. J.* **27**, 1439–1457 (2019).

1034. Samantaray, S., Rath, A. & Swain, P. C. Conjunctive use of groundwater and surface water in a part of Hirakud Command Area. *Int. J. Eng. Technol.* **9**, 3002–3010 (2017).

1035. Samper, J. et al. Evaluació de los impactos del cambio climático e los acuíferos de la pla a de la galera y del aluvial de Tortosa. Estudios en la Zona no Saturada del Suelo. Vol. X, 359–364. http://zonanosaturada.com/zns11/publications/p359.pdf (2011).

1036. Sanchez, R. & Eckstein, G. Groundwater management in the borderlands of Mexico and Texas: the beauty of the unknown, the negligence of the present, and the way forward. *Water Resour. Res.* **56**, e2019WR026068 (2020).

1037. Sanchez, R., Lopez, V. & Eckstein, G. Identifying and characterizing transboundary aquifers along the Mexico–US border: an initial assessment. *J. Hydrol.* **535**, 101–119 (2016).

1038. Sandberg, G. W. Ground-water resources of selected basins in southwestern Utah. U.S. Geological Survey Open Technical Publication 13. https://waterrights.utah.gov/docSys/v920/w920/w920008c.pdf (1966).

1039. Sandiford, M., Lawrie, K. & Brodie, R. S. Hydrogeological implications of active tectonics in the Great Artesian Basin, Australia. *Hydrol. J.* **28**, 57–73 (2020).

1040. Sanford, W. E. & Buapeng, S. Assessment of a groundwater flow model of the Bangkok Basin, Thailand, using carbon-14-based ages and paleohydrology. *Hydrol. J.* **4**, 26–40 (1996).

1041. Sanford, W. E., Pope, J. P., Selnick, D. L. & Stumvoll, R. F. Simulation of groundwater flow in the shallow aquifer system of the Delmarva Peninsula, Maryland and Delaware. U.S. Geological Survey Open-File Report 2012–1140. https://pubs.usgs.gov/of/2012/1140/pdf/OFR_2012-1140.pdf (2012).

1042. Santha, N., Sangkajan, S. & Saenton, S. Arsenic contamination in groundwater and potential health risk in Western Lampang Basin, Northern Thailand. *Water* **14**, 465 (2022).

1043. Santoni, S. et al. Strontium isotopes as tracers of water-rocks interactions, mixing processes and residence time indicator of groundwater within the granite-carbonate coastal aquifer of Bonifacio (Corsica, France). *Sci. Total Environ.* **573**, 233–246 (2016).

1044. Sanz, D. et al. Modeling aquifer–river interactions under the influence of groundwater abstraction in the Mancha Oriental System (SE Spain). *Hydrol. J.* **19**, 475–487 (2011).

1045. Savoca, M. E., Sadorf, E. M. & Akers, K. K. Ground-water quality in the eastern part of the Silurian-Devonian and Upper Carbonate Aquifers in the eastern Iowa basins, Iowa and Minnesota, 1996. U.S. Geological Survey Water-Resources Investigations Report 98-4224. https://pubs.usgs.gov/wri/1998/wri984224/pdf/wri98-4224.pdf (1999).

1046. Schoewe, W. H. The geography of Kansas: Part II. Physical geography. *Trans. Kans. Acad. Sci.* **52**, 261–333 (1949).

1047. Schrader, G. P. Unconsolidated aquifer systems of Ripley County, Indiana. Indiana Department of Natural Resources, Division of Water report. https://www.in.gov/dnr/water/files/ripley_unconsolidated_text.pdf (2004).

1048. Schult, J. Herbicides, pesticides and nutrients in the Tindall aquifer, Katherine Region. Northern Territory Government, Department of Land Resource Management report. https://landresources.nt.gov.au/__data/assets/pdf_file/0019/282160/GWQ-report.pdf (2016).

1049. Schwennesen, A. T. & Forbes, R. H. Ground water in San Simon Valley, Arizona and New Mexico. U.S. Geological Survey Water Supply Paper 425-A. https://pubs.usgs.gov/wsp/0425a/report.pdf (1919).

1050. Schwennesen, A. T. & Hare, R. F. Ground water in the Animas, Playas, Hachita, and San Luis Basins, New Mexico, with analyses of water and soil. U.S. Geological Survey Water-Supply Paper 422. https://pubs.usgs.gov/wsp/0422/report.pdf (1918).

1051. Scibek, J. & Allen, D. M. Numerical groundwater flow model of the Abbotsford-Sumas aquifer, central Fraser Lowland of BC, Canada, and Washington State, US. Report prepared for Environment Canada. https://www.sfu.ca/personal/dallen/AB_Modeling_Report_Final.pdf (2005).

1052. Scott, L., Hanson, C. & Cressy, C. Groundwater quality investigation of the mid-Waitaki valley. Environment Canterbury Regional Council Kaunihera Taiao ki Waitaha Report No. R12/71. http://citeseerx.ist.psu.edu/viewdoc/download?doi=10.1.1.799.6506&rep=rep1&type=pdf (2012).

1053. Scott, T.-M., Nystrom, E. A. & Reddy, J. E. Groundwater quality in the Lake Champlain and Susquehanna River basins, New York, 2014. U.S. Geological Survey Open-File Report 2016-1153. https://pubs.usgs.gov/of/2016/1153/ofr20161153.pdf (2016).

1054. Selck, B. J. et al. Investigating anthropogenic and geogenic sources of groundwater contamination in a semi-arid alluvial basin, Goshen Valley, UT, USA. *Water Air Soil Pollut.* **229**, 186 (2018).

1055. Semeniuk, V. & Semeniuk, C. A. Sedimentary fill of basin wetlands, central Swan Coastal Plain, southwestern Australia. Part 2: distribution of sediment types and their stratigraphy. *J. R. Soc. West. Aust.* **89**, 185 (2006).

1056. Senthilkumar, M. & Gnanasundar, D. Hydrogeological characterization and hydrological modeling for devising groundwater management strategies for Chennai aquifer system, Southern India. https://www.authorea.com/doi/full/10.22541/au.158990356.67099058 (2020).

1057. Seraphin, P., Gonçalvès, J., Vallet-Coulomb, C. & Champollion, C. Multi-approach assessment of the spatial distribution of the specific yield: application to the Crau plain aquifer, France. *Hydrol. J.* **26**, 1221–1238 (2018).

1058. Serrat, P. & Lenoble, J. L. La surexploitation des aquifères du Roussillon: une ressource patrimoniale en danger. *Houille Blanche* **93**, 71–78 (2007).

1059. Serviço Geológico do Brasil. Aquífero Urucuia Caracterização hidrológica com base en dados secundários. Inistério de Minas e Energia Secretaria de Geologia, Mineração e Transformação Mineral Serviço Geológico do Brasil (CPRM) report. https://rigeo.cprm.gov.br/jspui/handle/doc/20922 (2019).

1060. Shabani, M. Determining the most suitable interpolation method for groundwater chemical characteristics mapping. *Watershed Eng. Manag.* **3**, 196–204 (2012).

1061. Shah, T. Towards a managed aquifer recharge strategy for Gujarat, India: an economist's dialogue with hydro-geologists. *J. Hydrol.* **518**, 94–107 (2014).

1062. Shahmohammadi-Kalalagh, S., Taran, F. & Nasiri, H. Investigating groundwater level fluctuations via analyzing groundwater hydrograph: a case study of Naqadeh plain in north-west of Iran. *Sustain. Water Resour. Manag.* **6**, 8 (2020).

1063. Shalyari, N., Alinejad, A., Hashemi, A. H. G., RadFard, M. & Dehghani, M. Health risk assessment of nitrate in groundwater resources of Iranshahr using Monte Carlo simulation and geographic information system (GIS). *MethodsX* **6**, 1812–1821 (2019).

1064. Shams, M. et al. Drinking water in Gonabad, Iran: fluoride levels in bottled, distribution network, point of use desalinator, and decentralized municipal desalination plant water. *Fluoride* **45**, 138 (2012).

1065. Shamsudduha, M. Spatial variability and prediction modeling of groundwater arsenic distributions in the shallowest alluvial aquifers in Bangladesh. *J. Spat. Hydrol.* **7**, 33–46 (2007).

1066. Sharaf, M. A. & Hussein, M. T. Groundwater quality in the Saq aquifer, Saudi Arabia. *Hydrol. Sci. J.* **41**, 683–696 (1996).

1067. Sharpe, D. R. et al. in: *Canada's Groundwater Resources*, (ed. Rivera, A.) 444–499 (Fitzhenry and Whiteside, 2013).

1068. Shelton, J. L., Fram, M. S., Munday, C. M. & Belitz, K. Groundwater-quality data for the Sierra Nevada study unit, 2008. Results from the California GAMA program. U.S. Geological Survey Data Series 534. https://pubs.usgs.gov/ds/534/ds_534.pdf (2010).

1069. Sheppard, G. M. *The Hydrogeology of the Kaikoura Plains, North Canterbury, New Zealand*. PhD dissertation, Univ. Canterbury (1995).

1070. Shintani, T. et al. Three-dimensional structure and sources of groundwater masses beneath the Osaka Plain, Southwest Japan. *J. Hydrol. Reg. Stud.* **43**, 101193 (2022).

1071. Shterev, K. D. The hydrogeothermal basin of Sofia graben (Bulgaria). *Environ. Geol.* **46**, 651–660 (2004).

1072. Shu, L. C., Liu, P. G. & Ong'or, B. T. I. Environmental impact assessment using FORM and groundwater system reliability concept: case study Jining, China. *Environ. Geol.* **55**, 661–667 (2008).

1073. Siebenthal, C. E. Geology and water resources of the San Luis Valley, Colorado. U.S. Geological Survey Water-Supply Paper 240. https://pubs.usgs.gov/wsp/0240/report.pdf (1910).

1074. Sikandar, P., Bakhsh, A., Arshad, M. & Rana, T. The use of vertical electrical sounding resistivity method for the location of low salinity groundwater for irrigation in Chaj and Rachna Doabs. *Environ. Earth Sci.* **60**, 1113–1129 (2010).

1075. Silar, J. & Silar, J. in *Application of Tracers in Arid Zone Hydrology* (eds Adar, E. M. & Leibundgut, C.) 141–150 (IAHS, 1995).

1076. Simonson, B. M., Schubel, K. A. & Hassler, S. W. Carbonate sedimentology of the early Precambrian Hamersley Group of western Australia. *Precambrian Res.* **60**, 287–335 (1993).

1077. Simpson, M. A. Geology and hydrostratigraphy of the Rosetown Area (72O), Saskatchewan. Saskatchewan Research Council Publication No. 10416-2C98. https://www.wsask.ca/wp-content/uploads/2021/08/Groundwater-Resources-Report-Rosetown.pdf (1998).

1078. Singaraja, C. et al. A study on the status of saltwater intrusion in the coastal hard rock aquifer of South India. *Environ. Dev. Sustain.* **17**, 443–475 (2015).

1079. Singh, J., Erenstein, O., Thorpe, W. R. & Varma, A. Crop-livestock interactions and livelihoods in the Gangetic Plains of Uttar Pradesh, India: a regional synthesis. International Livestock Research Institute (2007).

1080. Singh, Y. & Dubey, D. P. in *Watershed Management for Sustainable Development* (eds Tiwari, R. N. & Pandey, G. P.) 122–134 (Excellent Publishing House, 2014).

1081. Sinsakul, S. Late quaternary geology of the lower central plain, Thailand. *J. Asian Earth Sci.* **18**, 415–426 (2000).

1082. Sloan, M., Gillies, J. A. & Norum, D. I. Using poor quality groundwater for irrigation in Saskatchewan, Canada. *Can. Water Resour. J.* **16**, 45–64 (1991).

1083. Smedley, P. L., Zhang, M., Zhang, G. & Luo, Z. Mobilisation of arsenic and other trace elements in fluviolacustrine aquifers of the Huhhot Basin, Inner Mongolia. *Appl. Geochem.* **18**, 1453–1477 (2003).

1084. Smerdon, B. D. & Ramsley, T. R. Water resource assessment for the Surat region. A technical report to the Australian Government from the CSIRO Great Artesian Basin Water Resource Assessment. https://publications.csiro.au/rpr/download?pid=csiro:EP132644&dsid=DS4 (2012).

1085. Smerdon, B. D., Ramsley, T. R., Radke, B. M., Kellett, J. R. Water resource assessment for the Great Artesian Basin. A technical report to the Australian Government from the CSIRO Great Artesian Basin Water Resource Assessment. https://publications.csiro.au/rpr/download?pid=csiro:EP132685&dsid=DS3 (2012).

1086. Smit, P. J. Groundwater recharge in the dolomite of the Ghaap Plateau near Kuruman in the Northern Cape, Republic of South Africa. *Water SA* **4**, 81–92 (1978).

1087. Smith, D. W., Buto, S. G. & Welborn, T. L. Groundwater-level change and evaluation of simulated water levels for irrigated areas in Lahontan Valley, Churchill County, west-central Nevada, 1992–2012. U.S. Geological Survey Scientific Investigations Report 2016-5045. https://pubs.usgs.gov/sir/2016/5045/sir20165045.pdf (2016).

1088. Smith, K. *Assessing the Hydrogeologic Characteristics and Sources of Groundwater Recharge and Flow in the Elandsfontein Aquifer, West Coast, Western Cape, South Africa*. MSc thesis, Univ. Western Cape (2020).

1089. Smith, L. N. Hydrologic framework of the Lolo-Bitterroot Area ground-water characterization study. Montana Bureau of Mines and Geology. Montana Ground-Water Assessment Atlas 4-B-02. http://mbmg.mtech.edu/pdf-publications/GWAA04B-02.pdf (2006).

1090. Smith, L. N., LaFave, J. I. & Patton, T. W. Groundwater resources of the Lolo-Bitterroot area: Mineral, Missoula, and Ravalli counties, Montana. Montana Bureau of Mines and Geology. Montana Groundwater Assessment Atlas No. 4. http://www.mbmg.mtech.edu/pdf-publications/gwaa4a.pdf (2013).

1091. Smith, L. N. Hydrogeologic framework of the southern part of the Flathead Lake Area, Flathead, Lake, Missoula, and Sanders counties, Montana. Montana Bureau of Mines and Geology. Montana Ground-Water Assessment Atlas 2-B-10. http://mbmggwic.mtech.edu/gwcpmaps/gwaa02map10untiled.pdf (2004).

1092. Smith, S. J., Fontaine, K. & Lewis, S. J. Regional hydrogeological characterisation of the St Vincent Basin, South Australia. Technical Report for the National Collaboration Framework Regional Hydrogeology Project. Geoscience Australia Record 2015/16. https://d28rz98at9flks.cloudfront.net/78884/Rec2015_016.pdf (2015).

1093. Smith, S. J. et al. Hydrogeology and model-simulated groundwater availability in the Salt Fork Red River aquifer, southwestern Oklahoma, 1980–2015. U.S. Geological Survey Scientific Investigations Report 2021-5003. https://pubs.usgs.gov/sir/2021/5003/sir20215003.pdf (2021).

1094. Smith, S. J., Ellis, J. H., Wagner, D. L. & Peterson, S. M. Hydrogeology and simulated groundwater flow and availability in the North Fork Red River aquifer, southwest Oklahoma, 1980–2013. U.S. Geological Survey Scientific Investigations Report 2017-5098. https://pubs.usgs.gov/sir/2017/5098/sir20175098.pdf (2017).

1095. Smolensky, D. A., Buxton, H. T. & Shernoff, P. K. Hydrologic framework of Long Island, New York. U.S. Geological Survey Hydrologic Atlas 709. https://pubs.usgs.gov/ha/709/plate-1.pdf (1990).

1096. Sneed, M., Brandt, J. T. & Solt, M. Land subsidence, groundwater levels, and geology in the Coachella Valley, California, 1993–2010. U.S. Geological Survey Scientific Investigations Report 2014-5075. https://pubs.usgs.gov/sir/2014/5075/pdf/sir2014-5075.pdf (2014).

1097. Sohrabi, N., Chitsazan, M., Amiri, V. & Nezhad, T. M. Evaluation of groundwater resources in alluvial aquifer based on MODFLOW program, case study: Evan plain (Iran). *Int. J. Agric. Crop Sci.* **5**, 1164–1170 (2013).

1098. Soldo, B., Mahmoudi Sivand, S., Afrasiabian, A. & Đurin, B. Effect of sinkholes on groundwater resources in arid and semi-arid karst area in Abarkooh, Iran. *Environments* **7**, 26 (2020).

1099. Soltani Mohammadi, A., Sayadi Shahraki, A. & Naseri, A. A. Simulation of groundwater quality parameters using ANN and ANN+ PSO models (case study: Ramhormoz Plain). *Pollution* **3**, 191–200 (2017).

1100. Soltani, S., Asghari Moghaddam, A., Barzegar, R. & Kazemian, N. Evaluation of nitrate concentration and vulnerability of the groundwater by GODS and AVI methods (case study: Kordkandi-Duzduzan Plain, East Azarbaijan province). *Iran. J. Ecohydrol.* **3**, 517–531 (2016).

1101. Soltani, S., Moghaddam, A. A., Barzegar, R., Kazemian, N. & Tziritis, E. Hydrogeochemistry and water quality of the Kordkandi-Duzduzan plain, NW Iran: application of multivariate statistical analysis and PoS index. *Environ. Monit. Assess.* **189**, 455 (2017).

1102. Sorensen, J. P. et al. The influence of groundwater abstraction on interpreting climate controls and extreme recharge events from well hydrographs in semi-arid South Africa. *Hydrogeol. J.*, 1–15 (2021).

1103. Souid, F., Birkle, P. & Worrall, F. Water-rock interaction of the Jilh and Tawil aquifers in the Wadi Sirhan Basin, NW Saudi Arabia. *E3S Web Conf.* **98**, 01047 (2019).

1104. South African Department of Water Affairs. Aquifer classification of South Africa. https://www.dws.gov.za/Groundwater/documents/Aquifer%20Classification.pdf (2012).

1105. Squeo, F. A. et al. Groundwater dynamics in a coastal aquifer in north-central Chile: implications for groundwater recharge in an arid ecosystem. *J. Arid. Environ.* **67**, 240–254 (2006).

1106. Sreenivas, A., Gowtham, B., Vinodh, K. & Kumaresan, K. Aquifer mapping of hard rock terrain in parts of Dindigul district, Tamil Nadu. *Int. J. Anal. Exp. Modal Anal.* **12**, 200–211 (2020).

1107. Srivastava, M. & Poonia, O. P. Transboundary aquifers in Rajasthan, issues & management. Bhujal News, 28–36. https://hindi.indiawaterportal.org/articles/transboundary-aquifers-rajasthan-issues-management (2010).

1108. Stamos, C. L., Christensen, A. H. & Langenheim, V. Preliminary hydrogeologic assessment near the boundary of the Antelope Valley and El Mirage Valley groundwater basins, California. U.S. Geological Survey Scientific Investigations Report 2017-5065. https://pubs.usgs.gov/sir/2017/5065/sir20175065.pdf (2017).

1109. Standen, A. R. & Kane, J. A. The spatial distribution of radiological contaminants in the Hickory aquifer and other aquifers overlying the Llano Uplift, Central Texas. *Austin Geol. Soc. Bull.* **1**, 87–101 (2023).

1110. Stapinsky, M. et al. Groundwater resources assessment in the Carboniferous Maritimes Basin: preliminary results of the hydrogeological characterization, New Brunswick, Nova Scotia, and Prince Edward Island. Geological Survey of Canada Current Research Report 2002-D8. http://www.gov.pe.ca/photos/original/cle_WA10.pdf (2002).

1111. State of New Mexico, Office of the State Engineer. Nutt-Hockett Basin Hydrographic Survey Report. https://www.ose.state.nm.us/HydroSurvey/legal_ose_hydro_nutt-hocket.php (1998).

1112. Steinbrügge, G., Muñoz Pardo, J. F. & Fernández, B. Análisis probabilístico y optimización de los recursos de agua subterránea: el caso del acuífero Maipo-Mapocho, Chile. Ingeniería hidraulica en Mexico, XX, 85–97. https://repositorio.uc.cl/dspace/bitstreams/2172bd6b-172e-4233-806a-c9c2b0af5c13/download (2005).

1113. Steinich, B., Escolero, O. & Marín, L. E. Salt-water intrusion and nitrate contamination in the Valley of Hermosillo and El Sahuaral coastal aquifers, Sonora, Mexico. *Hydrol. J.* **6**, 518–526 (1998).

1114. Stephenson, D. A. Hydrogeology of glacial deposits of the Mahomet Bedrock Valley in east-central Illinois. Illinois State Geological Survey Circular 409. https://www.ideals.illinois.edu/items/35335/bitstreams/112693/data.pdf (1967).

1115. Stephenson, L. W. The ground-water resources of Mississippi. U.S. Geological Survey Water-Supply Paper 576. https://pubs.usgs.gov/wsp/0576/report.pdf (1941).

1116. Steuer, A., Helwig, S. L. & Tezkan, B. Aquifer characterization in the Ouarzazate Basin (Morocco): a contribution by TEM and RMT data. *Near Surf. Geophys.* **6**, 5–14 (2008).

1117. Stolp, B. J. et al. Age dating base flow at springs and gaining streams using helium-3 and tritium: Fischa-Dagnitz system, southern Vienna Basin, Austria. *Water Resour. Res.* **46**, W07503 (2010).

1118. Story, J. & Lopez-Gunn, E. Comparing conflict in transboundary aquifer management: some insights from a comparative study between Spain and Australia. https://unesdoc.unesco.org/ark:/48223/pf0000190140 (2010).

1119. Strom, E. W. & Mallory, M. J. Hydrogeology and simulation of ground-water flow in the Eutaw-McShan Aquifer and in the Tuscaloosa aquifer system in northeastern Mississippi. U.S. Geological Survey Water-Resources Investigations Report 94-4223. https://pubs.usgs.gov/wri/1994/4223/report.pdf (1995).

1120. Subramanian, S. & Balasubramanian, A. Hydrochemical studies of Tiruchendur Coast, Tamilnadu, India. Regional Workshop on Environmental Aspects of Groundwater Development (1994).

1121. Sun, X. et al. Analysis and evaluation of the renewability of the deep groundwater in the Huaihe River Basin, China. *Environ. Earth Sci.* **80**, 104 (2021).

1122. Sun, Y., Zhou, J., Zho, Y., Zeng, Y. & Chen, Y. Influencing factors of groundwater organic pollution around the Bosten Lake area of Xinjiang, China. *E3S Web Conf.* **98**, 09029 (2019).

1123. Sureshjani, M. K., Amanipoor, H. & Battaleb-Looie, S. The effects of industrial wastewater on groundwater quality of the Boroujen aquifer, Southwest Iran. *Nat. Resour. Res.* **29**, 3719–3741 (2020).

1124. Sweetkind, D. S., Faunt, C. C. & Hanson, R. T. Construction of 3-D geologic framework and textural models for Cuyama Valley groundwater basin, California. U.S. Geological Survey Scientific Investigations Report 2013-5127. https://pubs.usgs.gov/sir/2013/5127/pdf/sir2013-5127.pdf (2013).

1125. Szczucińska, A., Dłużewski, M., Kozłowski, R. & Niedzielski, P. Hydrochemical diversity of a large alluvial aquifer in an arid zone (Draa river, S Morocco). *Ecol. Chem. Eng. S* **26**, 81–100 (2019).

1126. Szynkiewicz, A., Medina, M. R., Modelska, M., Monreal, R. & Pratt, L. M. Sulfur isotopic study of sulfate in the aquifer of Costa de Hermosillo (Sonora, Mexico) in relation to upward intrusion of saline groundwater, irrigation pumping and land cultivation. *Appl. Geochem.* **23**, 2539–2558 (2023).

1127. Tafreshi, G. M., Nakhaei, M. & Lak, R. Land subsidence risk assessment using GIS fuzzy logic spatial modeling in Varamin aquifer, Iran. *GeoJournal* **86**, 1203–1223 (2019).

1128. Tagma, T., Hsissou, Y., Bouchaou, L., Bouragba, L. & Boutaleb, S. Groundwater nitrate pollution in Souss-Massa basin (south-west Morocco). *Afr. J. Environ. Sci. Technol.* **3**, 301–309 (2009).

1129. Taheri Zangi, S. & Vaezihir, A. Vulnerability of Shazand Plain subsidence caused by groundwater level reduction using weighting model and its validation analysis using radar interferometry. *Iran. J. Ecohydrol.* **7**, 183–194 (2020).

1130. Taheri, K., Missimer, T. M., Amini, V., Bahrami, J. & Omidipour, R. A GIS-expert-based approach for groundwater quality monitoring network design in an alluvial aquifer: a case study and a practical guide. *Environ. Monit. Assess.* **192**, 684 (2020).

1131. Talebi, M. S. & Fatemi, M. Assessment of the quality and quantity of groundwater in Bahadoran plain using neural network methods, geostatistical and multivariate statistical analysis. *J. Appl. Res. Water Wastewater* **7**, 144–151 (2020).

1132. Tanachaichoksirikun, P. & Seeboonruang, U. Distributions of groundwater age under climate change of Thailand's Lower Chao Phraya basin. *Water* **12**, 3474 (2020).

1133. Tanaka, T. Groundwater resources, development and management in the Kanto Plain, Japan. https://core.ac.uk/download/pdf/76125416.pdf (2004).

1134. Tanigawa, K., Hyodo, M. & Sato, H. Holocene relative sea-level change and rate of sea-level rise from coastal deposits in the Toyooka Basin, western Japan. *Holocene* **23**, 1039–1051 (2013).

1135. Taniguchi, M. Estimated recharge rates from groundwater temperatures in the Nara Basin, Japan. *Appl. Hydrogeol.* **2**, 7–14 (1994).

1136. Taucare, M. et al. Connectivity of fractures and groundwater flows analyses into the Western Andean Front by means of a topological approach (Aconcagua Basin, Central Chile). *Hydrol. J.* **28**, 2429–2438 (2020).

1137. Tauchen, P. et al. Wind/Bighorn River Basin Water Plan Update Groundwater Study Level 1 (2008–2011). Groundwater Determination. Wyoming Water Development Commission Technical Memorandum. https://waterplan.state.wy.us/plan/bighorn/2010/gw-finalrept/gw-finalrept.pdf (2012).

1138. Tavassoli, S. & Mohammadi, F. Critically assessment of groundwater quality based on WQI and its vulnerability to saltwater intrusion in a coastal city, Iran. *Mod. Adv. Geogr. Environ. Earth Sci.* **2**, 126–138 (2021).

1139. Taweesin, K., Seeboonruang, U. & Saraphirom, P. The influence of climate variability effects on groundwater time series in the lower central plains of Thailand. *Water* **10**, 290 (2018).

1140. Taylor, C. B. et al. Sources and flow of north Canterbury plains groundwater, New Zealand. *J. Hydrol.* **106**, 311–340 (1989).

1141. Taylor, C. J & Nelson Jr, H. L. A compilation of provisional karst geospatial data for the Interior Low Plateaus physiographic region, central United States. U.S. Geological Survey Data Series 339. https://pubs.usgs.gov/ds/339/pdf/ds339_web.pdf (2008).

1142. Taylor, G. C. & Ghosh, P. K. Artesian water in the Malabar coastal plain of southern Kerala, India. U.S. Geological Survey Water Supply Paper 1608-D. https://pubs.usgs.gov/wsp/1608d/report.pdf (1964).

1143. Teng, Y. et al. Risk assessment framework for nitrate contamination in groundwater for regional management. *Sci. Total Environ.* **697**, 134102 (2019).

1144. Tezangi, M. F. Studying the effects of drought on groundwater aquifers of Zarand, Kerman. *Int. J. Pharm. Res. Allied Sci.* **5**, 437–447 (2016).

1145. Thamke, J. N., LeCain, G. D., Ryter, D. W., Sando, R. & Long, A. J. Hydrogeologic framework of the uppermost principal aquifer systems in the Williston and Powder River structural basins, United States and Canada. U.S. Geological Survey Scientific Investigations Report 2014-5047. https://pubs.usgs.gov/sir/2014/5047/pdf/sir2014-5047.pdf (2014).

1146. Thiros, S. A., Stolp, B. J., Hadley, H. K. & Steiger, J. I. Hydrology and simulation of ground-water flow in Juab Valley, Juab County, Utah. State of Utah Department of Natural Resources, Division of Water Rights Technical Publication No. 114. https://waterrights.utah.gov/docSys/v920/y920/y920000j.pdf (1996).

1147. Thiros, S. A. Hydrogeology of shallow basin-fill deposits in areas of Salt Lake Valley, Salt Lake County, Utah. U.S. Geological Survey Water-Resources Investigations Report 03-4029. https://pubs.usgs.gov/wri/wri034029/pdf/wri034029.pdf (2003).

1148. Thomas, H. E. Ground water in Tooele Valley, Tooele County, Utah. State of Utah Department of Natural Resources, Division of Water Rights Technical Publication No. 4. https://waterrights.utah.gov/docSys/v920/w920/w9200083.pdf (1946).

1149. Thorleifson, L. H. et al. Hydrogeology and hydrogeochemistry of the Red River Valley/Interlake region of Manitoba. Manitoba Energy and Mines, Minerals Division Report of Activities, 172–185 (1998).

1150. Tickell, S. J. Groundwater resources of the Oolloo Dolostone. Department of Infrastructure Planning and Environment, Natural Resources Division Report 17/2002. https://citeseerx.ist.psu.edu/viewdoc/download?doi=10.1.1.932.9762&rep=rep1&type=pdf (2002).

1151. Tillman, F. D., Cordova, J. T., Leake, S. A., Thomas, B. E. & Callegary, J. B. Water availability and use pilot: methods development for a regional assessment of groundwater availability, southwest alluvial basins, Arizona. U.S. Geological Survey Scientific Investigations Report 2011-5071. https://pubs.usgs.gov/sir/2011/5071/sir2011-5071_text.pdf (2011).

1152. Tillman, F. D., Garner, B. D. & Truini, M. Preliminary groundwater flow model of the basin-fill aquifers in Detrital, Hualapai, and Sacramento Valleys, Mohave County, northwestern Arizona. U.S. Geological Survey Scientific Investigations Report 2013-5122. http://pubs.usgs.gov/sir/2013/5122/ (2013).

1153. Timms, N. E. et al. Sedimentary facies analysis, mineralogy and diagenesis of the Mesozoic aquifers of the central Perth Basin, Western Australia. *Mar. Pet. Geol.* **60**, 54–78 (2015).

1154. Tizro, T. A., Voudouris, K. S. & Kamali, M. Comparative study of step drawdown and constant discharge tests to determine the aquifer transmissivity: the Kangavar aquifer case study, Iran. *J. Water Resour. Hydraul. Eng.* **3**, 12–21 (2014).

1155. Tokarsky, O. Hydrogeologic profile Alberta-Saskatchewan boundary. Report prepared for the Prairie Provinces Water Board. https://www.ppwb.ca/uploads/media/5c81764eb01c3/ppwb-report-78-no-maps-en.pdf?v1 (1985).

1156. Tokarsky, O. Hydrogeologic profile Saskatchewan-Manitoba boundary. Report prepared for the Prairie Provinces Water Board. https://www.ppwb.ca/uploads/media/5c81764f23261/ppwb-report-79-no-maps-en.pdf?v1 (1985).

1157. Tomás, R., Lopez-Sanchez, J. M., Delgado, J., Mallorquí Franquet, J. J. & Herrera García, G. in *Droughts: Causes, Effects and Predictions* (ed. Sánchez, J. M.) 253–276 (Nova Science, 2008).

1158. Tomás, R. et al. Mapping ground subsidence induced by aquifer overexploitation using advanced Differential SAR Interferometry: Vega Media of the Segura River (SE Spain) case study. *Remote Sens. Environ.* **98**, 269–283 (2005).

1159. Tomozawa, Y., Onodera, S. I. & Saito, M. Estimation of groundwater recharge and salinization in a coastal alluvial plain and Osaka megacity, Japan, using $\delta^{18}O$, $\delta D$, and $Cl^-$. *Geomate J.* **16**, 153–158 (2019).

1160. Torak, L. J. & Painter, J. A. Geostatistical estimation of the bottom altitude and thickness of the Mississippi River Valley alluvial aquifer. U.S. Geological Survey Scientific Investigations Map 3426. https://pubs.er.usgs.gov/publication/sim3426 (2019).

1161. Torkamanitombeki, H., Rahnamarad, J. & Saadatkhah, N. Groundwater chemical indices changed due to water-level decline, Minab Plain, Iran. *Environ. Earth Sci.* **77**, 269 (2018).

1162. Torres-Martínez, J. A., Mora, A., Knappett, P. S., Ornelas-Soto, N. & Mahlknecht, J. Tracking nitrate and sulfate sources in groundwater of an urbanized valley using a multi-tracer approach combined with a Bayesian isotope mixing model. *Water Res.* **182**, 115962 (2020).

1163. Torres-Martínez, J. A. et al. Estimation of nitrate pollution sources and transformations in groundwater of an intensive livestock-agricultural area (Comarca Lagunera), combining major ions, stable isotopes and MixSIAR model. *Environ. Pollut.* **269**, 115445 (2021).

1164. Torres-Martinez, J. A. et al. Constraining a density-dependent flow model with the transient electromagnetic method in a coastal aquifer in Mexico to assess seawater intrusion. *Hydrol. J.* **27**, 2955–2972 (2019).

1165. Torres-Rondon, L., Carrière, S. D., Chalikakis, K. & Valles, V. An integrative geological and geophysical approach to characterize a superficial deltaic aquifer in the Camargue plain, France. *C. R. Geosci.* **345**, 241–250 (2013).

1166. Tosaki, Y. et al. Deep incursion of seawater into the Hiroshima Granites during the Holocene transgression: evidence from $^{36}Cl$ age of saline groundwater in the Hiroshima area, Japan. *Geochem. J.* **51**, 263–275 (2017).

1167. Tournoud, M. G., Payraudeau, S., Cernesson, F. & Salles, C. Origins and quantification of nitrogen inputs into a coastal lagoon: application to the Thau lagoon (France). *Ecol. Model.* **193**, 19–33 (2006).

1168. Tran, D. A. et al. Groundwater quality evaluation and health risk assessment in coastal lowland areas of the Mekong Delta, Vietnam. *Groundw. Sustain. Dev.* **15**, 100679 (2021).

1169. Trapp Jr, H. Hydrology of sand-and-gravel aquifer in central and southern Escambia County, Florida. U.S. Geological Survey Open-File Report 74-218. https://pubs.usgs.gov/of/1974/0218/report.pdf (1973).

1170. Trapp Jr, H. & Horn, M. A. Ground water atlas of the United States: Segment 11, Delaware, Maryland, New Jersey, North Carolina, Pennsylvania, Virginia, West Virginia. U.S. Geological Survey Hydrologic Investigations Atlas 730-L. https://pubs.usgs.gov/ha/730l/report.pdf (1997).

1171. Treu, F. et al. Intrinsic vulnerability of the Isonzo/Soča high plain aquifer (NE Italy–W Slovenia). *J. Maps* **13**, 799–810 (2017).

1172. Truong, P. V. Hydrogeochemistry characteristics and salinity of groundwater in Quaternary sediments in the coastal zone of Ha Tinh province. *Vietnam J. Earth Sci.* **37**, 70–78 (2015).

1173. Tucci, P. Use of a three-dimensional model for the analysis of the ground-water flow system in Parker Valley, Arizona and California. U.S. Geological Survey Open-File Report 82-1006. https://pubs.usgs.gov/of/1982/1006/report.pdf (1982).

1174. U.S. Geological Survey. National water summary 1984: hydrologic events, selected water-quality trends, and ground-water resources. U.S. Geological Survey Water-Supply Paper 2275. https://pubs.usgs.gov/wsp/2275/report.pdf (1984).

1175. Umvoto Africa. The assessment of water availability in the Berg Catchment (WMA 19) by means of Water Resource Related Models. Department of Water Affairs and Forestry report. https://www.dws.gov.za/Documents/Other/WMA/19/Reports/Rep9-Vol5-GW%20Cape%20Flats%20Aquifer.pdf (2008).

1176. United States Bureau of Reclamation. Final feasibility-level special study report. Odessa subarea special study. https://www.usbr.gov/pn/programs/eis/odessa/finaleis/final.pdf (2012).

1177. University of Greenwich and Gujarat Institute of Desert Ecology. Ecosystem assessment of the coastal plain natural area of Kachchh District: planning for biodiversity and livelihoods into the future. Project presentation. https://gala.gre.ac.uk/id/eprint/16221/1/16221%20BARTLETT_Coastal_Plain_of_Kachchh_2016.pdf (2016).

1178. Upson, J. E. & Thomasson, H. G. Geology and water resources of the Santa Ynez river basin, Santa Barbara County, California, Vol. 2. U.S. Geological Survey Water-Supply Report 1107. https://pubs.usgs.gov/wsp/1107/report.pdf (1951).

1179. Urresti-Estala, B., Gavilán, P. J., Pérez, I. V. & Cantos, F. C. Assessment of hydrochemical trends in the highly anthropised Guadalhorce River basin (southern Spain) in terms of compliance with the European groundwater directive for 2015. *Environ. Sci. Pollut. Res.* **23**, 15990–16005 (2016).

1180. Urrutia, J. et al. Hydrogeology and sustainable future groundwater abstraction from the Agua Verde aquifer in the Atacama Desert, northern Chile. *Hydrol. J.* **26**, 1989–2007 (2018).

1181. US Army Corps of Engineers. Water resources assessment of El Salvador. https://www.sam.usace.army.mil/Portals/46/docs/military/engineering/docs/WRA/ElSalvador/El%20Salvador%20WRA%20English.pdf (1998).

1182. Uthman, W. & Beck J. Hydrogeology of the Upper Beaverhead Basin near Dillon, Montana. Montana Bureau of Mines and Geology Open-File Report 384. https://dnrc.mt.gov/_docs/water/Hydro_science_data/mbmg_open-file_report_384.pdf (1998).

1183. Uugulu, S. & Wanke, H. Estimation of groundwater recharge in savannah aquifers along a precipitation gradient using chloride mass balance method and environmental isotopes, Namibia. *Phys. Chem. Earth A/B/C* **116**, 102844 (2020).

1184. Vaccaro, J. J., Hansen, A. J. & Jones, M. A. Hydrogeologic framework of the Puget Sound aquifer system, Washington and British Columbia. U.S. Geological Survey Professional Paper 1424-D. https://pubs.usgs.gov/pp/1424d/report.pdf (1998).

1185. Vaccaro, J. J. et al. Groundwater availability of the Columbia Plateau Regional Aquifer System, Washington, Oregon, and Idaho. U.S. Geological Survey Professional Paper 1817. https://doi.org/10.3133/pp1817 (2015).

1186. Vaezihir, A. & Tabarmayeh, M. Total vulnerability estimation for the Tabriz aquifer (Iran) by combining a new model with DRASTIC. *Environ. Earth Sci.* **74**, 2949–2965 (2015).

1187. Valin, Z. C. & McLaughlin, R. J. Locations and data for water wells of the Santa Rosa Valley, Sonoma County, California. U.S. Geological Survey Open File Report 2005-1318. https://pubs.usgs.gov/of/2005/1318/of2005-1318.pdf (2005).

1188. van Geldern, R. et al. Pleistocene paleo-groundwater as a pristine fresh water resource in southern Germany–evidence from stable and radiogenic isotopes. *Sci. Total Environ.* **496**, 107–115 (2014).

1189. Van Lam, N., Van Hoan, H. & Duc Nhan, D. Investigation into groundwater resources in southern part of the Red River's Delta Plain, Vietnam by the use of isotopic techniques. *Water* **11**, 2120 (2019).

1190. Varma, A. Groundwater resource and governance in Kerala. Status, issues and prospects. Forum for Policy Dialogue on Water Conflicts in India. Kerala Resource Centre report. https://www.soppecom.org/pdf/Groundwater-Resource-and-Governance-in-Kerala.pdf (2017).

1191. Varma, S. & Michael, K. Impact of multi-purpose aquifer utilisation on a variable-density groundwater flow system in the Gippsland Basin, Australia. *Hydrol. J.* **20**, 119–134 (2012).

1192. Vazquez Sanchez, E., Cortes, A., Jaimes Palomera, R., Fritz, P. & Aravena, R. Hidrogeologia isotopica de los valles de Cuautla y Yautepec, Mexico. *Geofís. Int.* **28**, 245–264 (1989).

1193. Vazquez, J. G., Grande, J. A., Barragán, F. J., Ocaña, J. A. & De La Torre, M. L. Nitrate accumulation and other components of the groundwater in relation to cropping system in an aquifer in Southwestern Spain. *Water Resour. Manag.* **19**, 1–22 (2005).

1194. Vega-Granillo, E. L., Cirett-Galán, S., De la Parra-Velasco, M. L. & Zavala-Juárez, R. Hidrogeología de Sonora, México. Panorama de la geología de Sonora, México (ed. Calmus, T.) 267–298. Universidad Nacional Autónoma de México, Instituto de Geología, Boletín 118. https://boletin.geologia.unam.mx/index.php/boletin/issue/view/14/12 (2011).

1195. Vergnes, J. P. et al. The AquiFR hydrometeorological modelling platform as a tool for improving groundwater resource monitoring over France: evaluation over a 60-year period. *Hydrol. Earth Syst. Sci.* **24**, 633–654 (2020).

1196. Vetrimurugan, E., Elango, L. & Rajmohan, N. Sources of contaminants and groundwater quality in the coastal part of a river delta. *Int. J. Environ. Sci. Technol.* **10**, 473–486 (2013).

1197. Veve, T. D. & Taggart, B. E. Atlas of Ground-Water Resources in Puerto Rico and the U.S. Virgin Islands. U.S. Geological Survey Water-Resources Investigations Report 94-4198. https://pubs.usgs.gov/wri/1994/4198/report.pdf (1996).

1198. Villanueva-Hernández, H., Tovar-Cabañas, R. & Vargas-Castilleja, R. Classification of aquifers in the Mina field, Nuevo Leon, using geographic information systems. *Tecnol. Cienc. Agua* **10**, 96–123 (2019).

1199. Villegas, P., Paredes, V., Betancur, T. & Ribeiro, L. Assessing the hydrochemistry of the Urabá Aquifer, Colombia by principal component analysis. *J. Geochem. Explor.* **134**, 120–129 (2013).

1200. Virbulis, J., Bethers, U., Saks, T., Sennikovs, J. & Timuhins, A. Hydrogeological model of the Baltic Artesian Basin. *Hydrol. J.* **21**, 845–862 (2013).

1201. Vizintin, G., Souvent, P., Veselič, M. & Curk, B. C. Determination of urban groundwater pollution in alluvial aquifer using linked process models considering urban water cycle. *J. Hydrol.* **377**, 261–273 (2009).

1202. Vogel, J. C., Talma, A. S., Heaton, T. H. E. & Kronfeld, J. Evaluating the rate of migration of an uranium deposition front within the Uitenhage Aquifer. *J. Geochem. Explor.* **66**, 269–276 (1999).

1203. Vroblesky, D. A. & Fleck, W. B. Hydrogeologic Framework of the Coastal Plain of Maryland, Delaware, and the District of Columbia. U.S. Geological Survey Professional Paper 1404-E. https://pubs.usgs.gov/pp/1404e/report.pdf (1991).

1204. Wacker, M. A., Cunningham, K. J. & Williams, J. H. Geologic and hydrogeologic frameworks of the Biscayne aquifer in central Miami-Dade County, Florida. U.S. Geological Survey Scientific Investigations Report 2014-5138. https://pubs.usgs.gov/sir/2014/5138/sir2014-5138.pdf (2014).

1205. Wade, S. & Jigmond, M. Groundwater availability model of west Texas Bolsons (Presidio and Redford) Aquifer. Texas Water Development Board report. https://www.twdb.texas.gov/groundwater/models/gam/prbl/PRBL_ModelFinalReport.pdf (2013).

1206. Wallace, J. & Lowe, M. Ground-water quality classification for the Principal Basin-fill Aquifer, Salt Lake Valley, Salt Lake County, Utah. Utah Geological Survey Open-File Report 560. https://ugspub.nr.utah.gov/publications/open_file_reports/ofr-560.pdf (2009).

1207. Wang, D., Yang, C. & Shao, L. The spatiotemporal evolution of hydrochemical characteristics and groundwater quality assessment in Urumqi, Northwest China. *Arab. J. Geosci.* **14**, 161 (2021).

1208. Wang, L. & Iwao, Y. Groundwater characteristics of the Saga Plain, Japan. *J. Nepal Geol. Soc.* **22**, 343–350 (2000).

1209. Wang, S. J., Lee, C. H., Yeh, C. F., Choo, Y. F. & Tseng, H. W. Evaluation of climate change impact on groundwater recharge in groundwater regions in Taiwan. *Water* **13**, 1153 (2021).

1210. Wang, S. et al. Shallow groundwater dynamics in North China plain. *J. Geog. Sci.* **19**, 175–188 (2009).

1211. Washington State Department of Ecology. Puget Sound groundwater toxics loading analysis: direct discharge pathway. Publication No. 10-03-122. https://apps.ecology.wa.gov/publications/documents/1003122.pdf (2010).

1212. Water and Marine Resources Division. Tasmanian Aquifer Framework. Groundwater Management Report Series. Report No. GW 2012/02. https://nre.tas.gov.au/Documents/Tasmanian%20Aquifer%20Framework.pdf (2012).

1213. Watts, K. R. Hydrogeology and quality of ground water in the upper Arkansas River Basin from Buena Vista to Salida, Colorado, 2000–2003. U.S. Geological Survey Scientific Investigations Report 2005-5179. https://pubs.usgs.gov/sir/2005/5179/pdf/SIR2005-5179.pdf (2005).

1214. Wei, M., Allen, D. M., Carmichael, V. & Ronneseth, K. State of understanding of the hydrogeology of the Grand Forks aquifer. Water Stewardship Division, BC Ministry of Environment Report. https://www.grandforks.ca/wp-content/uploads/reports/2010-Hydrogeology-Study-of-Grand-Forks-area.pdf (2010).

1215. Weiss, J. S. Geohydrologic units of the coastal lowlands aquifer system, south-central United States. U.S. Geological Survey regional aquifer-system analysis. https://pubs.usgs.gov/pp/1416c/report.pdf (1990).

1216. Welch, A. H., Sorey, M. L. & Olmsted, F. H. Hydrothermal system in Southern Grass Valley, Pershing County, Nevada. U.S. Geological Survey Open-File Report 81-915. https://www.osti.gov/servlets/purl/5119283-5mJ8YB/ (1981).

1217. Welder, G. E. Geohydrologic framework of the Roswell ground-water basin, Chaves and Eddy Counties, New Mexico. New Mexico State Engineer Technical Report 42. https://www.ose.state.nm.us/Library/TechnicalReports/TechReport-042.pdf (1983).

1218. Welder, G. E. Plan of study for the regional aquifer system analysis of the San Juan structural basin, New Mexico, Colorado, Arizona, and Utah. U.S. Geological Survey Water-Resources Investigations Report 85-4294. https://pubs.usgs.gov/wri/1985/4294/report.pdf (1986).

1219. Wellman, T. P. Evaluation of groundwater levels in the South Platte River alluvial aquifer, Colorado, 1953–2012, and design of initial well networks for monitoring groundwater levels. U.S. Geological Survey Scientific Investigations Report 2015-5015. https://pubs.usgs.gov/sir/2015/5015/pdf/sir2015-5015.pdf (2015).

1220. Welsh, W. D. Spatial and temporal water balance estimates using a GIS. Engineers Australia. https://openresearch-repository.anu.edu.au/bitstream/1885/43108/2/HYDRO2005_bowen2.pdf (2005).

1221. Westjohn, D. B. & Weaver, T. L. Hydrogeologic framework of the Michigan Basin regional aquifer system. U.S. Geological Survey Professional Paper 1418. https://pubs.usgs.gov/pp/1418/report.pdf (1998).

1222. Whitcomb, H. A. & Lowry, M. E. Ground-water resources and geology of the Wind River Basin area, central Wyoming. U.S. Geological Survey Hydrologic Atlas 270. https://pubs.usgs.gov/ha/270/report.pdf (1968).

1223. White, P. A. & Reeves, R. R. The volume of groundwater in New Zealand 1994 to 2001. Statistics New Zealand, Client Report 2002/79. https://docs.niwa.co.nz/library/public/volume-of-groundwater-in-nz-2001%5B1%5D.pdf (2002).

1224. White, W. N. Preliminary report on the ground-water supply of Mimbres Valley, New Mexico. U.S. Geological Survey Water Supply Paper 637. https://pubs.usgs.gov/wsp/0637B/report.pdf (1931).

1225. Whitehead, E. J. & Lawrence, A. R. The Chalk aquifer of Lincolnshire. British Geological Survey Research Report RR/06/03. http://nora.nerc.ac.uk/id/eprint/3699/1/RR06003.pdf (2006).

1226. Whitehead, R. L. Geohydrologic framework of the Snake River Plain regional aquifer system, Idaho and eastern Oregon. U.S. Geological Survey Professional Paper 1408-B. https://pubs.usgs.gov/pp/1408b/report.pdf (1992).

1227. Whittlemore, D. O., Macfarlane, P. A. & Wilson, B. B. Water Resources of the Dakota Aquifer in Kansas. Kansas Geological Survey Bulletin 260. http://www.kgs.ku.edu/Publications/Bulletins/260/Bulletin_260_Dakota.pdf (2014).

1228. Wildermuth Environmental. Chino Basin Optimum Basin Management Program. State of the Basin Report – 2004. Report prepared for Chino Basin Watermaster. http://www.cbwm.org/docs/engdocs/isob/ISOB_Final_FullVersion.pdf (2005).

1229. Wilkes, P. Baseline assessment of groundwater characteristics in the Beetaloo Sub-basin, NT. GISERA Project Order. https://gisera.csiro.au/wp-content/uploads/2018/10/Water-16-Project-Order-1.pdf (2018).

1230. Williams, L. J. & Kuniansky, E. L. Revised hydrogeologic framework of the Floridan aquifer system in Florida and parts of Georgia, Alabama, and South Carolina. U.S. Geological Survey Professional Paper 1807. https://pubs.usgs.gov/pp/1807/pdf/pp1807.pdf (2016).

1231. Willmes, M. et al. Mapping of bioavailable strontium isotope ratios in France for archaeological provenance studies. *Appl. Geochem.* **90**, 75–86 (2018).

1232. Wilson, D. D. The significance of geology in some current water resource problems, Canterbury Plains, New Zealand. *J. Hydrol. (New Zeal.)* **12**, 103–118 (1973).

1233. Wilson, H. D. Ground-water appraisal of Santa Ynez River basin, Santa Barbara County, California, 1945-52. U.S. Geological Survey Water-Supply Paper 1467. https://pubs.usgs.gov/wsp/1467/report.pdf (1959).

1234. Wilson, J. E., Brown, S., Schreier, H., Scovill, D. & Zubel, M. Arsenic in groundwater wells in Quaternary deposits in the Lower Fraser Valley of British Columbia. *Can. Water Resour. J.* **33**, 397–412 (2008).

1235. Wilson, J. T. Water-quality assessment of the Cambrian-Ordovician aquifer system in the northern Midwest, United States. U.S. Geological Survey Scientific Investigations Report 2011-5229. https://pubs.usgs.gov/sir/2011/5229/pdf/SIR20115229_web.pdf (2012).

1236. Winner Jr, M. D. & Coble, R. W. Hydrogeologic framework of the North Carolina Coastal Plain aquifer system. U.S. Geological Survey Open-File Report 87-690. https://pubs.usgs.gov/of/1987/0690/report.pdf (1989).

1237. Wolfgang, C. Hydrogeology of the Pilliga sandstone aquifer in the Western Coonamble embayment and its implications for water resource management. PhD thesis, Australia National Univ. (2000).

1238. Wood, P. R. Geology and ground-water features of the Butte Valley region, Siskiyou County, California. U.S. Geological Survey Water-Supply Paper 1491. https://pubs.usgs.gov/wsp/1491/report.pdf (1960).

1239. Wood, P. R. & Davis, G. H. Ground-water conditions in the Avenal-McKittrick Area Kings and Kern Counties California. U.S. Geological Survey Water-Supply Paper 1457. https://pubs.usgs.gov/wsp/1457/report.pdf (1959).

1240. Woodman, N. D., Burgess, W. G., Ahmed, K. M. & Zahid, A. A partially coupled hydro-mechanical analysis of the Bengal Aquifer System under hydrological loading. *Hydrol. Earth Syst. Sci.* **23**, 2461–2479 (2019).

1241. Woodward, D. G., Gannett, M. W. & Vaccaro, J. J. Hydrogeologic framework of the Willamette Lowland aquifer system, Oregon and Washington. U.S. Geological Survey Professional Paper 1424-B. https://pubs.usgs.gov/pp/1424b/report.pdf (1998).

1242. Woolfenden, L. R. & Nishikawa, T. Simulation of groundwater and surface-water resources of the Santa Rosa Plain watershed, Sonoma County, California. U.S. Geological Survey Scientific Investigations Report 2014-5052. https://pubs.usgs.gov/sir/2014/5052/pdf/sir2014-5052.pdf (2014).

1243. Worts, G. F. & Thomasson, H. G. Geology and ground-water resources of the Santa Maria Valley area, California. U.S. Geological Survey Water-Supply Paper 1000. https://pubs.usgs.gov/wsp/1000/report.pdf (1951).

1244. Wright, P. R. Hydrogeology and water quality in the Snake River alluvial aquifer at Jackson Hole Airport, Jackson, Wyoming, water years 2011 and 2012. U.S. Geological Survey Scientific Investigations Report 2013-5184. https://pubs.usgs.gov/sir/2013/5184/pdf/sir2013-5184.pdf (2013).

1245. Wurl, J. & Imaz-Lamadrid, M. A. Coupled surface water and groundwater model to design managed aquifer recharge for the valley of Santo Domingo, BCS, Mexico. *Sustain. Water Resour. Manag.* **4**, 361–369 (2018).

1246. Xiao, Y. et al. Hydrogeochemical constraints on groundwater resource sustainable development in the arid Golmud alluvial fan plain on Tibetan plateau. *Environ. Earth Sci.* **80**, 750 (2021).

1247. Xu, N., Gong, J. & Yang, G. Using environmental isotopes along with major hydro-geochemical compositions to assess deep groundwater formation and evolution in eastern coastal China. *J. Contam. Hydrol.* **208**, 1–9 (2018).

1248. Xu, Y. S., Shen, S. L., Ma, L., Sun, W. J. & Yin, Z. Y. Evaluation of the blocking effect of retaining walls on groundwater seepage in aquifers with different insertion depths. *Eng. Geol.* **183**, 254–264 (2014).

1249. Xue, Z., Du, P., Li, J. & Su, H. Sparse graph regularization for robust crop mapping using hyperspectral remotely sensed imagery with very few in situ data. *ISPRS J. Photogramm. Remote Sens.* **124**, 1–15 (2017).

1250. Yamamoto, S. The groundwater hydrology of river valley (2) on the groundwater of Kinokawa valley. *Geogr. Rev. Jpn.* **24**, 8–16 (1951).

1251. Yang, W. Q., Shen, L., Xiao, H. & Wang, Y. Z. Impact of shallow groundwater quality evolution in Kunming Urban by human activities. *Adv. Mater. Res.* **788**, 302–306 (2013).

1252. Yangouliba, G. I. et al. Modelling past and future land use and land cover dynamics in the Nakambe River Basin, West Africa. *Model. Earth Syst. Environ.* **9**, 1651–1667 (2022).

1253. Yazdi, Z. & Niroumand, H. Assessing land subsidence in Qazvin plain caused by groundwater level drop, using finite elements and finite difference methods. GeoTerrace-2020-043. https://eage.in.ua/wp-content/uploads/2020/12/GeoTerrace-2020-043.pdf (2020).

1254. Yeh, H. F. Spatiotemporal variation of the meteorological and groundwater droughts in central Taiwan. *Front. Water* **3**, 636792 (2021).

1255. Yeh, H. F., Lin, H. I., Lee, C. H., Hsu, K. C. & Wu, C. S. Identifying seasonal groundwater recharge using environmental stable isotopes. *Water* **6**, 2849–2861 (2014).

1256. Yoneda, M. et al. Groundwater deterioration caused by induced recharge: field survey and verification of the deterioration mechanism by stochastic numerical simulation. *Water Air Soil Pollut.* **127**, 125–146 (2001).

1257. Yonesi, H. et al. Evaluating groundwater quality in Zayandehrood southern sub-basin aquifers. *Desert Ecosyst. Eng. J.* **9**, 103–115 (2020).

1258. Yoosefdoo, I. & Khashei Siuki, A. Determine the vulnerability of the aquifer using the standard drastic and data-based methods (case study: Kochisfahan Aquifer). *Iran. J. Remote Sens. GIS* **9**, 99–116 (2018).

1259. Yoshioka, Y. et al. Multiple-indicator study of the response of groundwater recharge sources to highly turbid river water after a landslide in the Tedori River alluvial fan, Japan. *Hydrol. Process.* **34**, 3539–3554 (2020).

1260. Yoshioka, Y. & Yoshioka, H. Spatiotemporal variability of hydrogen stable isotopes at a local scale in shallow groundwater during the warm season in Tottori Prefecture, Japan. *Hydrol. Res. Lett.* **16**, 25–31 (2022).

1261. Young, H. L. Hydrogeology of the Cambrian-Ordovician aquifer system in the northern Midwest, United States. U.S. Geological Survey Professional Paper 1405-B. https://pubs.usgs.gov/pp/1405b/report.pdf (1992).

1262. Young, H. W. Reconnaissance of ground-water resources in the Mountain Home plateau area, southwest Idaho. U.S. Geological Survey Water-Resources Investigations Report 77-108. https://pubs.usgs.gov/wri/1977/0108/report.pdf (1977).

1263. Young, R. A. & Carpenter, C. H. Ground-water conditions and storage in the Central Sevier Valley, Utah. U.S. Geological Survey Water-Supply Paper 1787. https://pubs.usgs.gov/wsp/1787/report.pdf (1965).

1264. Yu, H. L. & Chu, H. J. Recharge signal identification based on groundwater level observations. *Environ. Monit. Assess.* **184**, 5971–5982 (2012).

1265. Yu, H. L. & Chu, H. J. Understanding space–time patterns of groundwater system by empirical orthogonal functions: a case study in the Choshui River alluvial fan, Taiwan. *J. Hydrol.* **381**, 239–247 (2010).

1266. Yustres, Á., Navarro, V., Asensio, L., Candel, M. & García, B. Groundwater resources in the Upper Guadiana Basin (Spain): a regional modelling analysis. *Hydrol. J.* **21**, 1129 (2013).

1267. Zandi, R., Ghahraman, K. & Asadi, Z. Monitoring the land subsidence and its associated landforms using remote sensing techniques in Feyzabad Plain (north-east Iran). *J. Hydrosci. Environ.* **3**, 43–51 (2019).

1268. Zare, M. & Koch, M. Computation of the irrigation water demand in the Miandarband plain, Iran, using FAO-56-and satellite-estimated crop coefficients. *Interdiscip. Res. Rev.* **12**, 15–25 (2017).

1269. Zarour, H., Aitchison-Earl, P., Scott, M., Peaver, L. & De Silva, J. Current state of the groundwater resource in the Orari-Temuka-Opihi-Pareora area. Environment Canterbury Regional Council Report No. R16/41. https://api.ecan.govt.nz/TrimPublicAPI/documents/download/2964277 (2018).

1270. Zaryab, A., Nassery, H. R. & Alijani, F. Identifying sources of groundwater salinity and major hydrogeochemical processes in the Lower Kabul Basin aquifer, Afghanistan. *Environ. Sci. Process. Impacts* **23**, 1589–1599 (2021).

1271. Zeng, Y., Zhou, Y., Zhou, J., Jia, R. & Wu, J. Distribution and enrichment factors of high-arsenic groundwater in Inland Arid area of PR China: a case study of the Shihezi area, Xinjiang. *Expos. Health* **10**, 1–13 (2018).

1272. Zhang, B. et al. The renewability and quality of shallow groundwater in Sanjiang and Songnen Plain, Northeast China. *J. Integr. Agric.* **16**, 229–238 (2017).

1273. Zhang, G., Deng, W., Yang, Y. S. & Salama, R. B. Evolution study of a regional groundwater system using hydrochemistry and stable isotopes in Songnen Plain, northeast China. *Hydrol. Process.* **21**, 1055–1065 (2007).

1274. Zhang, H., Xu, Y., Cheng, S., Li, Q. & Yu, H. Application of the dual-isotope approach and Bayesian isotope mixing model to identify nitrate in groundwater of a multiple land-use area in Chengdu Plain, China. *Sci. Total Environ.* **717**, 137134 (2020).

1275. Zhang, H., Yang, R., Wang, Y. & Ye, R. The evaluation and prediction of agriculture-related nitrate contamination in groundwater in Chengdu Plain, southwestern China. *Hydrol. J.* **27**, 785–799 (2019).

1276. Zhang, L., Stauffacher, M., Walker, G. R. & Dyce, P. Recharge estimation in the Liverpool Plains (NSW) for input groundwater models. CSIRO Technical Report 10/97 (1997).

1277. Zhang, Q. et al. Predicting the risk of arsenic contaminated groundwater in Shanxi Province, Northern China. *Environ. Pollut.* **165**, 118–123 (2012).

1278. Zhang, W. et al. Using noble gases to trace groundwater evolution and assess helium accumulation in Weihe Basin, central China. *Geochim. Cosmochim. Acta* **251**, 229–246 (2019).

1279. Zhang, Y., Gable, C. W., Zyvoloski, G. A. & Walter, L. M. Hydrogeochemistry and gas compositions of the Uinta Basin: A regional-scale overview. *AAPG Bull.* **93**, 1087–1118 (2009).

1280. Zhang, Y. et al. Land subsidence and uplift due to long-term groundwater extraction and artificial recharge in Shanghai, China. *Hydrol. J.* **23**, 1851–1866 (2015).

1281. Zhen, L. & Martin, P. Geohydrology, simulation of regional groundwater flow, and assessment of water-management strategies, Twentynine Palms area, California. U.S. Geological Survey Scientific Investigations Report 2010-5249. https://pubs.usgs.gov/sir/2010/5249/pdf/sir20105249.pdf (2011).

1282. Zhong, Y. et al. Groundwater depletion in the West Liaohe River Basin, China and its implications revealed by GRACE and in situ measurements. *Remote Sens.* **10**, 493 (2018).

1283. Zhou, J., Hu, B. X., Cheng, G., Wang, G. & Li, X. Development of a three-dimensional watershed modelling system for water cycle in the middle part of the Heihe rivershed, in the west of China. *Hydrol. Process.* **25**, 1964–1978 (2011).

1284. Zhou, Y., Wang, Y., Li, Y., Zwahlen, F. & Boillat, J. Hydrogeochemical characteristics of central Jianghan Plain, China. *Environ. Earth Sci.* **68**, 765–778 (2013).

1285. Zhou, Z. & Zhong, J. Role of atmospheric temperature and seismic activity in spring water hydrogeochemistry in Urumqi, China. *Int. J. Environ. Res. Public Health* **19**, 12004 (2022).

1286. Zhu, G. F., Li, Z. Z., Su, Y. H., Ma, J. Z. & Zhang, Y. Y. Hydrogeochemical and isotope evidence of groundwater evolution and recharge in Minqin Basin, Northwest China. *J. Hydrol.* **333**, 239–251 (2007).

1287. Zulfic, D., Harrington, N. & Evans, S. Uley Basin groundwater modelling project, volume 2: groundwater flow model. DWLBC Report 2007/04, Department of Water, Land and Biodiversity Conservation. https://www.waterconnect.sa.gov.au/Content/Publications/DEW/ki_dwlbc_report_2007_04.pdf (2006).

1288. GebreEgziabher, M., Jasechko, S. & Perrone, D. Widespread and increased drilling of wells into fossil aquifers in the USA. *Nat. Commun.* **13**, 2129 (2022).

1289. Taher, M. R., Chornack, M. P. & Mack, T. J. Groundwater levels in the Kabul Basin, Afghanistan, 2004–2013. U.S. Geological Survey Open-File Report 2013-1296. https://doi.org/10.3133/ofr20131296 (2014).

1290. Gong, H. et al. Long-term groundwater storage changes and land subsidence development in the North China Plain (1971–2015). *Hydrol. J.* **26**, 1417–1427 (2018).

1291. Winckel, A., Ollagnier, S. & Gabillard, S. Managing groundwater resources using a national reference database: the French ADES concept. *SN Appl. Sci.* **4**, 217 (2022).

1292. Ascott, M. J. et al. In situ observations and lumped parameter model reconstructions reveal intra-annual to multidecadal variability in groundwater levels in sub-Saharan Africa. *Water Resour. Res.* **56**, e2020WR028056 (2020).

1293. Tao, S. et al. Changes in China's water resources in the early 21st century. *Front. Ecol. Environ.* **18**, 188–193 (2020).

1294. Adamson, J. K. et al. Significance of river infiltration to the Port-Au-Prince metropolitan region: a case study of two coastal aquifers in Haiti. *Hydrol. J.* **30**, 1367–1386 (2022).

1295. Vongphachanh, S., Gupta, A. D., Milne-Home, W., Ball, J. E. & Pavelic, P. Hydrogeological reconnaissance of Sukhuma District, Champasak Province, Southern Laos. *J. Hydrol. (New Zeal.)* **56**, 79–96 (2017).

1296. Fallatah, O. A. Groundwater quality patterns and spatiotemporal change in depletion in the regions of the Arabian shield and Arabian shelf. *Arab. J. Sci. Eng.* **45**, 341–350 (2020).

1297. Hsu, Y. J. et al. Assessing seasonal and interannual water storage variations in Taiwan using geodetic and hydrological data. *Earth Planet. Sci. Lett.* **550**, 116532 (2020).

1298. Taylor, S. J. & Letham, B. Forecasting at scale. *Am. Stat.* **72**, 37–45 (2018).

1299. Friedman, J. H. & Stuetzle, W. Projection pursuit regression. *J. Am. Stat. Assoc.* **76**, 817–823 (1981).

1300. Theil, H. A rank-invariant method of linear and polynomial regression analysis. *Indag. Math.* **12**, 386–392 (1950).

1301. Sen, P. K. Estimates of the regression coefficient based on Kendall's tau. *J. Am. Stat. Assoc.* **63**, 1379–1389 (1968).

1302. Holland, P. W. & Welsch, R. E. Robust regression using iteratively reweighted least-squares. *Commun. Stat. Theory Methods* **6**, 813–827 (1977).

1303. Kirchner, J. W. Quantifying new water fractions and transit time distributions using ensemble hydrograph separation: theory and benchmark test. *Hydrol. Earth Syst. Sci.* **23**, 303–349 (2019).

1304. Kirchner, J. W. & Knapp, J. L. A. Technical note: Calculation scripts for ensemble hydrograph separation. *Hydrol. Earth Syst. Sci.* **24**, 5539–5558 (2020).

1305. Fisher, M. & Bolles, R. Random sample consensus: a paradigm for model fitting with applications to image analysis and automated cartography. *Commun. ACM* **24**, 381–395 (1981).

1306. Önöz, B. & Bayazit, M. Block bootstrap for Mann–Kendall trend test of serially dependent data. *Hydrol. Process.* **26**, 3552–3560 (2012).

1307. Shamsudduha, M. & Taylor, R. G. Groundwater storage dynamics in the world's large aquifer systems from GRACE: uncertainty and role of extreme precipitation. *Earth Syst. Dyn.* **11**, 755–774 (2020).

1308. Landerer, F. W. & Swenson, S. C. Accuracy of scaled GRACE terrestrial water storage estimates. *Water Resour. Res.* **48**, W04531 (2012).

1309. Watkins, M. M., Wiese, D. N., Yuan, D.-N., Boening, C. & Landerer, F. W. Improved methods for observing Earth's time variable mass distribution with GRACE using spherical cap mascons. *J. Geophys. Res. Solid Earth* **120**, 2648–2671 (2015).

1310. Wiese, D. N., Landerer, F. W. & Watkins, M. M. Quantifying and reducing leakage errors in the JPL RL05M GRACE mascon solution. *Water Resour. Res.* **52**, 7490–7502 (2016).

1311. Biancale, R. et al. 3 Years of Geoid Variations from GRACE and LAGEOS Data at 10-day Intervals from July 2002 to March 2005. CNES/GRGS data product (2006).

1312. de Graaf, I. D., Sutanudjaja, E. H., Van Beek, L. P. H. & Bierkens, M. F. P. A high-resolution global-scale groundwater model. *Hydrol. Earth Syst. Sci.* **19**, 823–837 (2015).

1313. Duran-Llacer, I. et al. Lessons to be learned: groundwater depletion in Chile's Ligua and Petorca watersheds through an Interdisciplinary approach. *Water* **12**, 2446 (2020).

1314. Narvaez-Montoya, C. et al. Predicting adverse scenarios for a transboundary coastal aquifer system in the Atacama Desert (Peru/Chile). *Sci. Total Environ.* **806**, 150386 (2022).

1315. Oiro, S., Comte, J. C., Soulsby, C., MacDonald, A. & Mwakamba, C. Depletion of groundwater resources under rapid urbanisation in Africa: recent and future trends in the Nairobi Aquifer System, Kenya. *Hydrol. J.* **28**, 2635–2656 (2020).

1316. Castellazzi, P., Garfias, J. & Martel, R. Assessing the efficiency of mitigation measures to reduce groundwater depletion and related land subsidence in Querétaro (Central Mexico) from decadal InSAR observations. *Int. J. Appl. Earth Obs. Geoinf.* **105**, 102632 (2021).

1317. Nguyen, M. et al. Assessment of long-term ground subsidence and groundwater depletion in Hanoi, Vietnam. *Eng. Geol.* **299**, 106555 (2022).

1318. Bui, L. K. et al. Recent land deformation detected by Sentinel-1A InSAR data (2016–2020) over Hanoi, Vietnam, and the relationship with groundwater level change. *GISci. Remote Sens.* **58**, 161–179 (2021).

1319. Moshfika, M., Biswas, S. & Mondal, M. S. Assessing groundwater level declination in Dhaka city and identifying adaptation options for sustainable water supply. *Sustainability* **14**, 1518 (2022).

1320. Sohail, M. T. et al. Groundwater budgeting of Nari and Gaj formations and groundwater mapping of Karachi, Pakistan. *Appl. Water Sci.* **12**, 267 (2022).

1321. Dehghani, F., Mohammadi, Z. & Zare, M. Assessment of groundwater depletion in a heterogeneous aquifer: historical reconnaissance and current situation. *Environ. Earth Sci.* **80**, 582 (2021).

1322. Gautam, A., Rai, S. C. & Rai, S. P. Impact of anthropogenic activities on the alluvial aquifers of north-east Punjab, India. *Environ. Monit. Assess.* **192**, 527 (2020).

1323. Sajjad, M. M. et al. Impact of climate and land-use change on groundwater resources, study of Faisalabad district, Pakistan. *Atmosphere* **13**, 1097 (2022).

1324. Ouassanouan, Y. et al. Multi-decadal analysis of water resources and agricultural change in a Mediterranean semiarid irrigated piedmont under water scarcity and human interaction. *Sci. Total Environ.* **834**, 155328 (2022).

1325. Goode, D. J., Senior, L. A., Subah, A. & Jaber, A. Groundwater-level trends and forecasts, and salinity trends, in the Azraq, Dead Sea, Hammad, Jordan Side Valleys, Yarmouk, and Zarqa groundwater basins, Jordan. U.S. Geological Survey Open-File Report 2013-1061. http://pubs.usgs.gov/of/2013/1061/ (2013).

1326. Naeem, U. A. et al. Impact of urbanization on groundwater levels in Rawalpindi City, Pakistan. *Pure Appl. Geophys.* **178**, 491–500 (2021).

1327. Snoussi, J., Jerbi, H. & Tarhouni, J. Integrated groundwater flow modeling for managing a complex alluvial aquifer case of study Mio-Plio-Quaternary Plain of Kairouan (Central Tunisia). *Water* **14**, 668 (2022).

1328. Zghibi, A. et al. Implications of groundwater development and seawater intrusion for sustainability of a Mediterranean coastal aquifer in Tunisia. *Environ. Monit. Assess.* **191**, 696 (2019).

1329. Cotterman, K. A., Kendall, A. D., Basso, B. & Hyndman, D. W. Groundwater depletion and climate change: future prospects of crop production in the Central High Plains Aquifer. *Clim. Change* **146**, 187–200 (2018).

1330. Orhan, O. Monitoring of land subsidence due to excessive groundwater extraction using small baseline subset technique in Konya, Turkey. *Environ. Monit. Assess.* **193**, 174 (2021).

1331. Xia, J. et al. Evaluating the dynamics of groundwater depletion for an arid land in the Tarim Basin, China. *Water* **11**, 186 (2019).

1332. Custodio, E. et al. Groundwater mining: benefits, problems and consequences in Spain. *Sustain. Water Resour. Manag.* **3**, 213–226 (2017).

1333. Taher, T. M. Groundwater abstraction management in Sana'a Basin, Yemen: a local community approach. *Hydrol. J.* **24**, 1593–1605 (2016).

1334. Delinom, R. M. in *Groundwater and Subsurface Environments* (ed. Taniguchi, M.) 113–125 (Springer, 2011).

1335. Taufiq, A. et al. Impact of excessive groundwater pumping on rejuvenation processes in the Bandung basin (Indonesia) as determined by hydrogeochemistry and modeling. *Hydrol. J.* **26**, 1263–1279 (2018).

1336. Zaryab, A., Nassery, H. R. & Alijani, F. The effects of urbanization on the groundwater system of the Kabul shallow aquifers, Afghanistan. *Hydrol. J.* **30**, 429–443 (2022).

1337. Carrillo, M., Gomez, Y. A., Valle, S. & Prado, J. V. Behavior of groundwater levels in Texcoco Aquifer (1507) when they are lowered by excessive pumping from 1968 through 2014. 2016 ASABE Annual International Meeting. American Society of Agricultural and Biological Engineers. https://elibrary.asabe.org/abstract.asp?aid=47273 (2016).

1338. Ojha, C., Werth, S. & Shirzaei, M. Groundwater loss and aquifer system compaction in San Joaquin Valley during 2012–2015 drought. *J. Geophys. Res. Solid Earth* **124**, 3127–3143 (2019).

1339. Noori, R. et al. Anthropogenic depletion of Iran's aquifers. *Proc. Natl Acad. Sci.* **118**, e2024221118 (2021).

1340. Ashraf, S., Nazemi, A. & AghaKouchak, A. Anthropogenic drought dominates groundwater depletion in Iran. *Sci. Rep.* **11**, 9135 (2021).

1341. Saowiang, K. & Giao, P. H. Numerical analysis of subsurface deformation induced by groundwater level changes in the Bangkok aquifer system. *Acta Geotech.* **16**, 1265–1279 (2021).

1342. Shi, W. et al. Spatial-temporal evolution of land subsidence and rebound over Xi'an in western China revealed by SBAS-InSAR analysis. *Remote Sens.* **12**, 3756 (2020).

1343. Sartirana, D. et al. Data-driven decision management of urban underground infrastructure through groundwater-level time-series cluster analysis: the case of Milan (Italy). *Hydrol. J.* **30**, 1157–1177 (2022).

1344. Houspanossian, J. et al. Agricultural expansion raises groundwater and increases flooding in the South American plains. *Science* **380**, 1344–1348 (2023).

1345. Galanter, A. E. & Curry, L. T. S. Estimated 2016 groundwater level and drawdown from predevelopment to 2016 in the Santa Fe Group aquifer system in the Albuquerque area, central New Mexico. U.S. Geological Survey Scientific Investigations Map 3433. https://doi.org/10.3133/sim3433 (2019).

1346. Hao, Y., Xie, Y., Ma, J. & Zhang, W. The critical role of local policy effects in arid watershed groundwater resources sustainability: a case study in the Minqin oasis, China. *Sci. Total Environ.* **601**, 1084–1096 (2017).

1347. Furi, W., Razack, M., Haile, T., Abiye, T. A. & Legesse, D. The hydrogeology of Adama-Wonji basin and assessment of groundwater level changes in Wonji wetland, Main Ethiopian Rift: results from 2D tomography and electrical sounding methods. *Environ. Earth Sci.* **62**, 1323–1335 (2011).

1348. Özel, N., Bozdağ, Ş. & Baba, A. Effect of irrigation system on groundwater resources in Harran Plain (Southeastern Turkey). *J. Food Sci. Eng.* **9**, 45–51 (2023).

1349. Duran-Llacer, I. et al. A new method to map groundwater-dependent ecosystem zones in semi-arid environments: a case study in Chile. *Sci. Total Environ.* **816**, 151528 (2022).

1350. Pino, E. et al. Factors affecting depletion and pollution by marine intrusion in the La Yarada's coastal aquifer, Tacna, Peru. *Tecnol. Cienc. Agua* **10**, 177–213 (2019).

1351. Vu, T. T. & Tran, N. V. T. Assessment of urbanization impact on groundwater resources in Hanoi, Vietnam. *J. Environ. Manag.* **227**, 107–116 (2018).

1352. Roy, S. K. & Zahid, A. Assessment of declining groundwater levels due to excessive pumping in the Dhaka District of Bangladesh. *Environ. Earth Sci.* **80**, 333 (2021).

1353. Taher, T., Bruns, B., Bamaga, O., Al-Weshali, A. & Van Steenbergen, F. Local groundwater governance in Yemen: building on traditions and enabling communities to craft new rules. *Hydrol. J.* **20**, 1177–1188 (2012).

1354. Rybakov, V. Water crisis in Yemen: speculations, realities and mitigation actions. https://static1.squarespace.com/static/5eb18d627d53aa0e85b60c65/t/5eda46ed1c956a6bc14ae36c/1591363321836/Report-victor.pdf (2012).

1355. Abidin, H. Z. et al. Land subsidence and groundwater extraction in Bandung Basin, Indonesia. IAHS publication 329, 145–156 (2009).

1356. Livoreil, B. et al. Systematic searching for environmental evidence using multiple tools and sources. *Environ. Evid.* **6**, 23 (2017).

1357. Malakar, P. et al. Three decades of depth-dependent groundwater response to climate variability and human regime in the transboundary Indus-Ganges-Brahmaputra-Meghna mega river basin aquifers. *Adv. Water Res.* **149**, 103856 (2021).

1358. Taylor, C. J. & Alley, W. M. Ground-water-level monitoring and the importance of long-term water-level data. U.S. Geological Survey Circular 1217 (2001).

1359. Russo, T. A. & Lall, U. Depletion and response of deep groundwater to climate-induced pumping variability. *Nat. Geosci.* **10**, 105–108 (2017).

1360. Hartmann, J. & Moosdorf, N. The new global lithological map database GLiM: a representation of rock properties at the Earth surface. *Geochem. Geophys. Geosyst.* **13**, Q12004 (2012).

1361. Hora, T., Srinivasan, V. & Basu, N. B. The groundwater recovery paradox in South India. *Geophys. Res. Lett.* **46**, 9602–9611 (2019).

1362. Patle, G. T. et al. Time series analysis of groundwater levels and projection of future trend. *J. Geol. Soc. India* **85**, 232–242 (2015).

1363. Shamsudduha, M., Taylor, R. G., Ahmed, K. M. & Zahid, A. The impact of intensive groundwater abstraction on recharge to a shallow regional aquifer system: evidence from Bangladesh. *Hydrol. J.* **19**, 901–916 (2011).

1364. Rushton, K. R., Zaman, M. A. & Mehedi Hasan, M. Sustainable abstraction due to unconfined conditions in multi-layered aquifers: examples from northwest Bangladesh. *Groundw. Sustain. Dev.* **20**, 100901 (2023).

1365. MacDonald, A. M. et al. Groundwater quality and depletion in the Indo-Gangetic Basin mapped from in situ observations. *Nat. Geosci.* **9**, 762–766 (2016).

1366. MacAllister, D. J., Krishan, G., Basharat, M., Cuba, D. & MacDonald, A. M. A century of groundwater accumulation in Pakistan and northwest India. *Nat. Geosci.* **15**, 390396 (2022).

1367. Perrone, D. & Jasechko, S. Dry groundwater wells in the western United States. *Environ. Res. Lett.* **12**, 104002 (2017).

1368. Perrone, D. & Jasechko, S. Deeper well drilling an unsustainable stopgap to groundwater depletion. *Nat. Sustain.* **2**, 773–782 (2019).

1369. Jasechko, S. & Perrone, D. Hydraulic fracturing near domestic groundwater wells. *Proc. Natl Acad. Sci.* **114**, 13138–13143 (2017).

1370. Mukherji, A., Rawat, S. & Shah, T. Major insights from India's minor irrigation censuses: 1986-87 to 2006-07. *Econ. Political Wkly.* **48**, 115–124 (2013).

1371. Laghari, A. N., Vanham, D. & Rauch, W. The Indus basin in the framework of current and future water resources management. *Hydrol. Earth Syst. Sci.* **16**, 1063–1083 (2012).

1372. Abatzoglou, J. T., Dobrowski, S. Z., Parks, S. A. & Hegewisch, K. C. TerraClimate, a high-resolution global dataset of monthly climate and climatic water balance from 1958–2015. *Sci. Data* **5**, 170191 (2018).

1373. Karger, D. N., Wilson, A. M., Mahony, C. & Zimmermann, N. E. Global daily 1 km land surface precipitation based on cloud cover-informed downscaling. *Sci. Data* **8**, 307 (2021).

**Acknowledgements** We gratefully acknowledge the contributions from individuals in dozens of organizations who are responsible for the generation of the primary datasets used in this study (see Supplementary Table 1). This material is based on work supported by the National Science Foundation under grant nos. EAR-2048227 and EAR-2234213. This research was supported by funding from the Zegar Family Foundation. This material is based on work supported by the U.S. Geological Survey (USGS) through the California Institute for Water Resources (CIWR) under grant/cooperative agreement no. G21AP10611-00. The views and conclusions contained in this document are those of the authors and should not be interpreted as representing the opinions or policies of the USGS/CIWR. Mention of trade names or commercial products does not constitute their endorsement by the USGS/CIWR. R.G.T. acknowledges the support of a fellowship (ref. 7040464) from the Canadian Institute for Advanced Research under the Earth 4D programme. S.J. acknowledges the Jack and Laura Dangermond Preserve (https://doi.org/10.25497/D7159W), the Point Conception Institute and the Nature Conservancy for their support of this research.

**Author contributions** S.J., D.P., M.S. and R.G.T. conceived the idea to analyse global piezometric records and S.J., H.S., D.P., Y.F., M.S., R.G.T. and J.W.K. co-developed the approach to analyse these records. S.J., H.S. and D.P. compiled groundwater-level data. M.S. compiled GRACE satellite data. O.F. accessed Saudi Arabian groundwater-level data. S.J. and H.S. completed geospatial and statistical analyses. S.J. delineated aquifer-system boundaries and wrote the first draft of the manuscript. S.J., H.S., D.P., Y.F., M.S., R.G.T. and J.W.K. contributed to writing and editing the manuscript.

**Competing interests** The authors declare no competing interests.

**Additional information**
**Correspondence and requests for materials** should be addressed to Scott Jasechko.

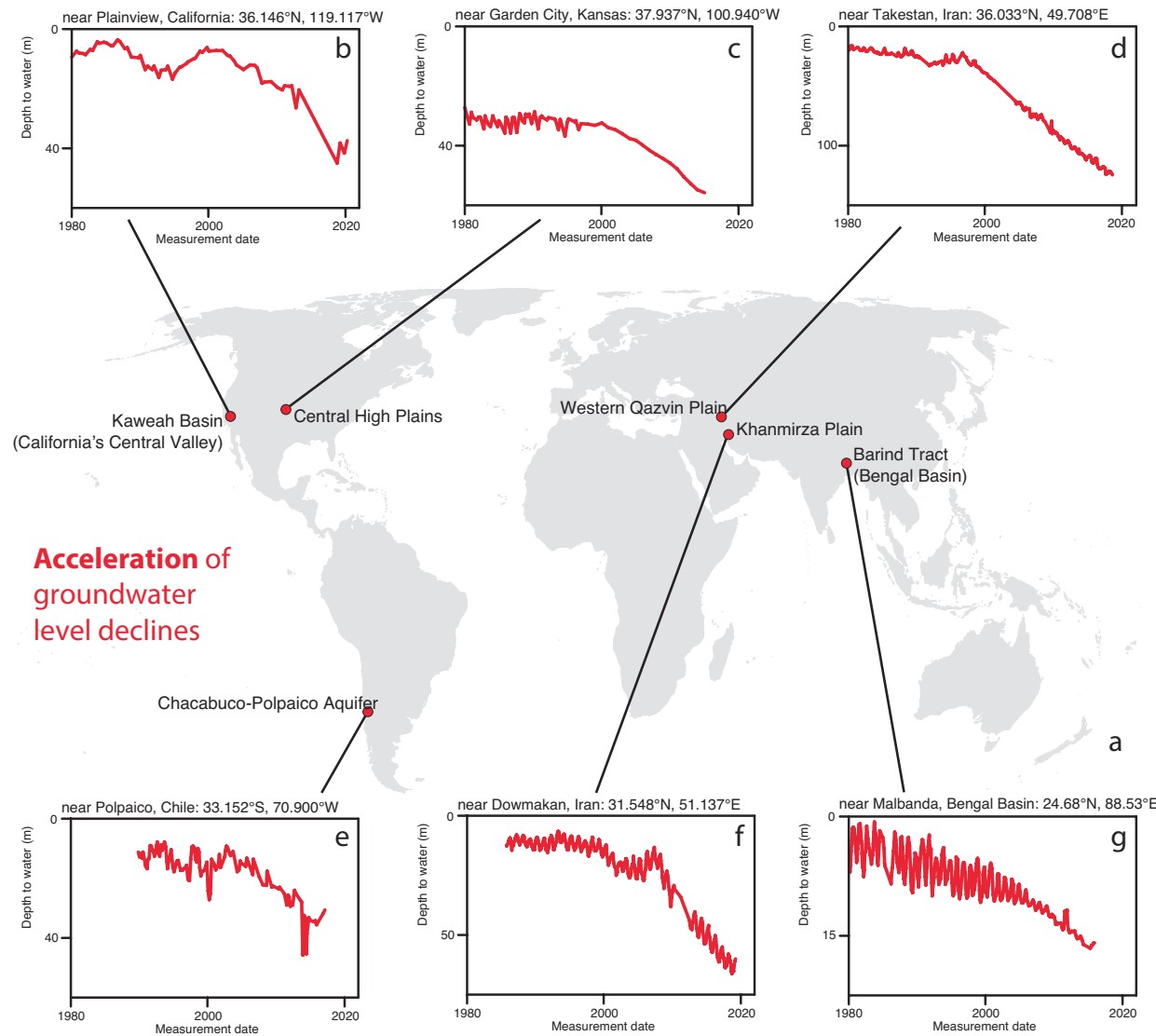

**Extended Data Fig. 1 | Illustrative examples of individual monitoring wells that record cases for which groundwater levels declined during late twentieth century and continued to decline at a faster rate in the early twenty-first century (that is, accelerated deepening). a**, Global map depicting the locations of the six monitoring wells (that is, each point represents one monitoring well). The aquifer system that each monitoring well lies in is labelled next to each point. **b**–**g**, Measured groundwater-level variations over time for individual monitoring wells. Each panel presents groundwater-level data for a single monitoring well.

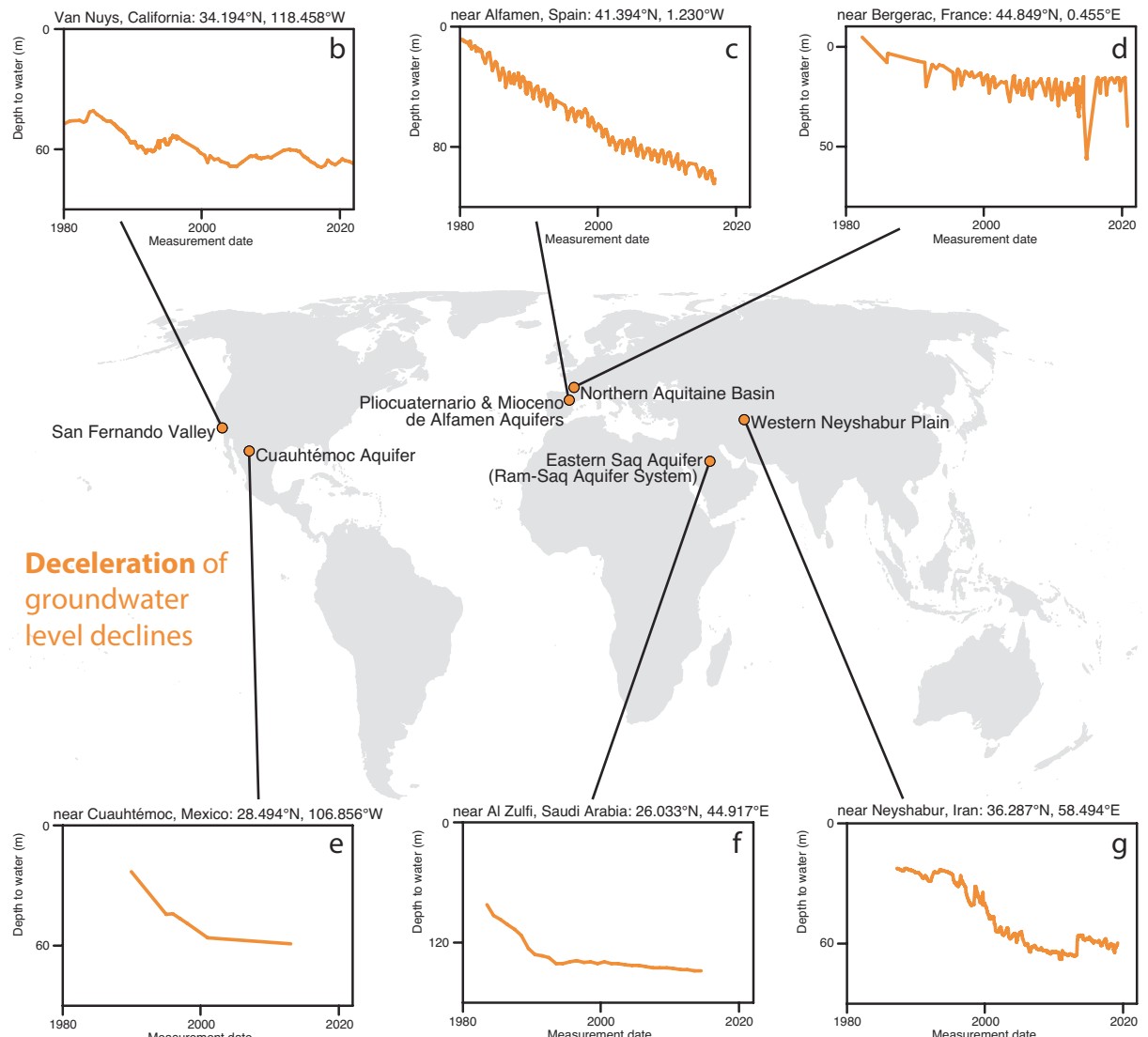

**Extended Data Fig. 2 | Illustrative examples of individual monitoring wells that record cases for which groundwater levels declined during late twentieth century and continued to decline but at a slower rate in the early twenty-first century (that is, decelerated deepening). a**, Global map depicting the locations of the six monitoring wells (that is, each point represents one monitoring well). The aquifer system that each monitoring well lies in is labelled next to each point. **b**–**g**, Measured groundwater-level variations over time for individual monitoring wells. Each panel presents groundwater-level data for a single monitoring well.

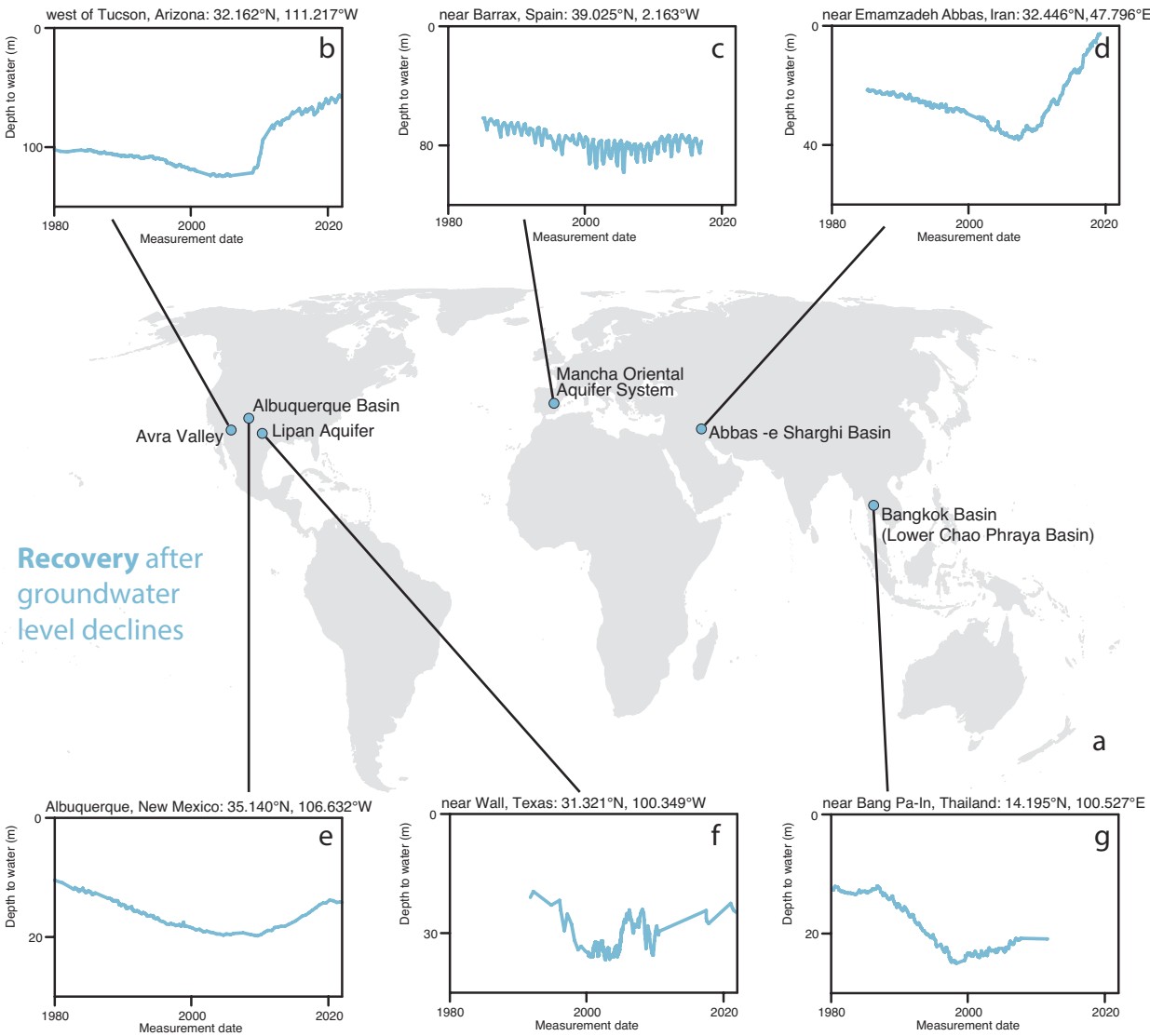

**Extended Data Fig. 3 | Illustrative examples of individual monitoring wells that record cases for which groundwater levels declined during late twentieth century but rose during the early twenty-first century (that is, cases of groundwater level recovery). a**, Global map depicting the locations of the six monitoring wells (that is, each point represents one monitoring well). The aquifer system that each monitoring well lies in is labelled next to each point. **b**–**g**, Measured groundwater-level variations over time for individual monitoring wells. Each panel presents groundwater-level data for a single monitoring well.

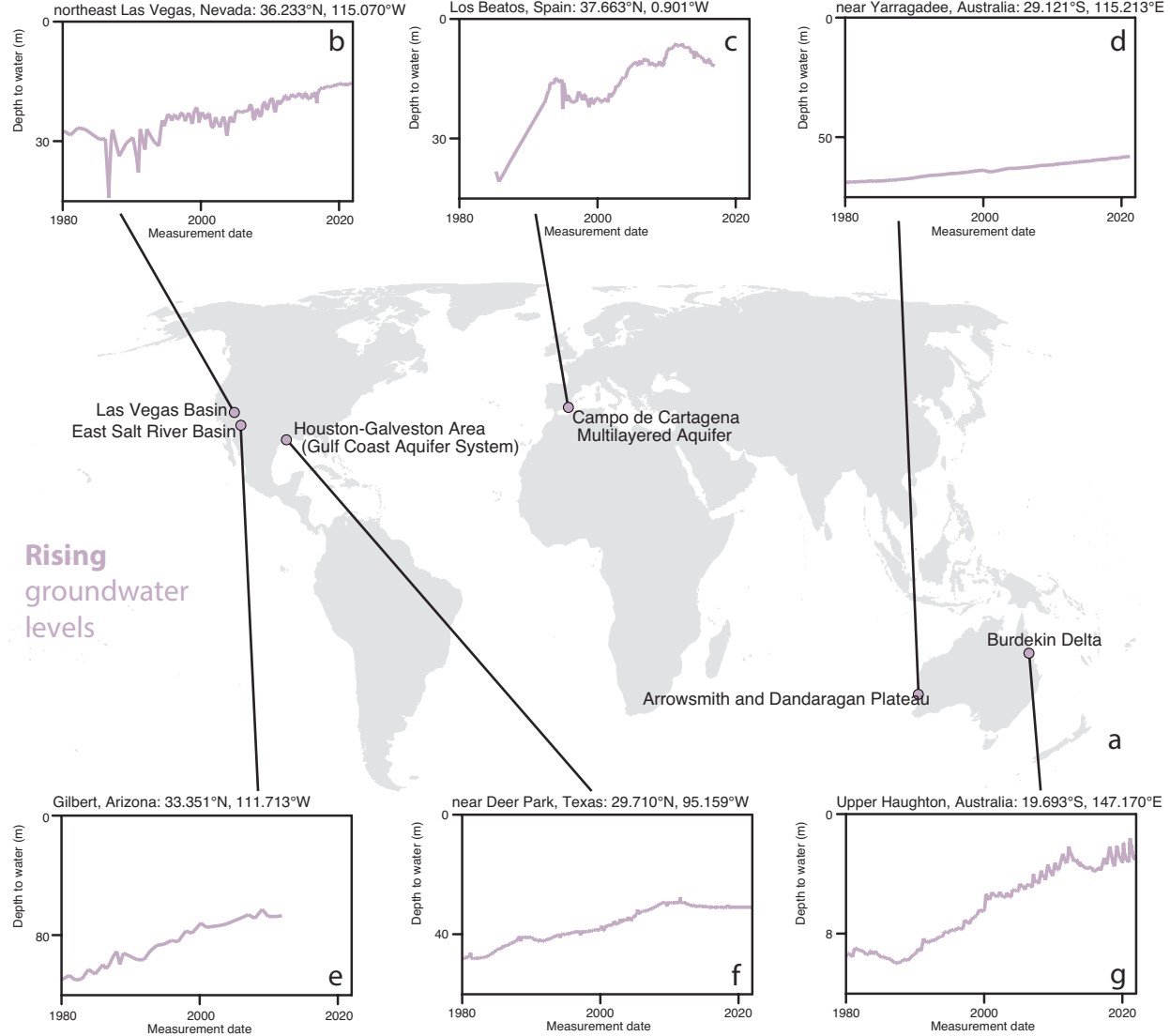

**Extended Data Fig. 4 | Illustrative examples of individual monitoring wells that record cases for which groundwater levels rose during late twentieth century, and continued to rise during the early twenty-first century. a**, Global map depicting the locations of the six monitoring wells (that is, each point represents one monitoring well). The aquifer system that each monitoring well lies in is labelled next to each point. **b**–**g**, Measured groundwater-level variations over time for individual monitoring wells. Each panel presents groundwater-level data for a single monitoring well.