## [Peer Review File · Nature]

Manuscript Title: Rapid groundwater declines in many aquifers globally but cases of recovery

Reviewer Comments & Author Rebuttals

Reviewer Reports on the Initial Version:

Referees' comments:

Referee #1 (Remarks to the Author):

Thank you for the opportunity to review the paper "Rapid and accelerating groundwater level declines in global cultivated drylands". This is an important research topic and the paper is well written. The authors collate well-level data around the world and assess groundwater level trends. They show that groundwater levels have declined in the past few decades in many of the world's regional aquifers. However, this study suffers from a major methodological limitation and is also similar to several other studies in the literature. For this reason, unfortunately, this study is not suitable for a publication in Nature at the present time.

Methods:

The authors compiled water level time series from ~1.5 million monitoring wells spanning more than 30 countries. However, these data were then filtered to exclude potential outliers and wells lacking sufficiently long time series for trend analysis. Once all is said and done, the authors are left with ~178,000 wells.

This means that only ~12% of the well data is used in the analysis. I am concerned that such a strong selection bias will lead to unrepresentative results. There is a lot of information for the 1.2 million wells that are not included in this study that may lead to different findings. The wells that remain in the analysis are unique, especially since only 12% of them pass the statistical requirements of the authors. This means that the results will reflect these unique wells and may make it difficult to draw generalizable conclusions.

What is unique about the included wells? How are they different to the excluded wells? How do the results change when all wells are considered (even using two data points for time series for wells with short time periods available).

Related studies:

- Fan et al (2013) "Global patterns of groundwater table depth", Science. In this study, the authors also use direct well measurements to estimate groundwater levels around the world.
- Jasechko and Perrone (2021) "Global groundwater wells at risk of running dry", Science. In this study, several of the study authors map and quantify well construction depth over time, in order to improve our understanding of spatiotemporal patterns of well locations and depths. Well locations and depths from this study would be closely correlated with groundwater pumping locations and levels (what this study shows).
- Perrone and Jasechko (2019) "Deeper well drilling an unsustainable stopgap to groundwater depletion", Nature Sustainability. In this study, the authors analyze spatial patterns of groundwater well depths and purposes in the US. Importantly, they find a strong correlation between groundwater-level time-series trends and groundwater-well construction depth time-series trends. This means that the findings of the submitted paper are similar to those in Jasechko and Perrone (2021).

- Several high-profile studies have shown that groundwater levels are declining around the world using different methods to the submitted paper. For example, several studies have used satellite imagery or models to show that global groundwater levels are declining. The main novelty in this study then is that well-level data is used to show that groundwater levels are declining. However, this is not the first time that well-level records have been used to reveal global groundwater levels (e.g., as in Fan et al 2013) or relevant trends (e.g., as in Jasechko and Perrone et al 2021). For this reason, I do not think using well-level data is sufficiently novel to warrant publication in Science for this study.

Data availability:

It is unfortunate that the authors are not able to provide the full suite of well data used in the study, making replication analyses difficult (about 12 countries are provided, out of over 30 countries included in the study; this is <50% of the data). It is also my understanding that several other related studies by some of the study authors have not been able to make the data necessary to replicate the results available. It is critical to share data to ensure reproducibility of scientific studies. I encourage the authors to share their data in future studies to support the scientific process.

Referee #2 (Remarks to the Author):

General comments:

This is a well written, expansive, technically sound, and rigorous study. I thoroughly enjoyed reading it. The methodology is well described and defensible. You have managed to pull together a unique and substantial global dataset which is a brilliant achievement. The data could do with improved description (i.e., particularly the nature of the time series used in the analysis, some additional visualisation may help). Overall, the results are very interesting, but I think you need to be careful when discussing their implications. I think the results indicate a much more nuanced picture than your discussion suggests. In fact, I would argue that the 46% of aquifer systems that have seen groundwater level declines slowing, reversed, or continuing to rise in the 21st century, have as much importance as the 54% of aquifers where declines have accelerated or remained constant. I think this nuance needs to be recognised in your discussion and the implications considered. There are important management lessons, which have not really been drawn out very effectively in the text, from areas where declines have slowed or reversed. I am concerned that your analysis was conducted with a presupposition that groundwater level decline dominates globally (see for example the narrative in Supplementary Note 10, looking for evidence of groundwater deepening) and that this has skewed the discussion of the findings.

My view is that in its present form the paper is not suitable for publication in Nature. The real novelty and importance of the findings has perhaps been drowned out by a focus on groundwater depletion, when, as highlighted above, it in fact it appears that there is a much more nuanced story globally. That is not to say that significant depletion is occurring and is not a serious issue, but that your results suggest that there is almost an equal number of aquifer systems where depletion has slowed, reversed, or not occurred at all in the last 40 years or so. Perhaps, if this nuance was recognised and discussed then the paper would be suitable for publication in Nature. However, I think that even then the authors need to make a much stronger case for its suitability for publication in Nature given the dominant (and defensible) narrative of groundwater depletion already in the literature and in the areas where you have data (e.g., Bierkens and Wada, 2019; Konikow and Kendy, 2005; Wada et al, 2010, Aeschbach-Hertig and Gleeson, 2012; and other related literature much of which you have already cited).

Specific comments:

Line 41 – 42: I would argue that the global prevalence of groundwater depletion is already well known in the areas where your data is concentrated (e.g. Scanlon et al, 2012 and Konikow 2014 in the United States; Chen et al., 2016 in Australia; Dangar et al., 2021 India; Feng et al., 2013 in China; and many other example in the literature) and that many of these studies also use in-situ

data. Having said this, I recognise that there are no other studies (at least as far as I am aware) that have integrated data from all these regions into one global analysis, and I think this is a big achievement, but it doesn't mean that the phenomenon was previously unknown.

Line 46 – 47: I am not sure this statement is true. What do you mean by a disproportionate share of the world's regional aquifers. Your results show that at least 46% of aquifers have experienced slowing depletion, reversals, or continued groundwater level rise.

Line 48 – 50: I think a key missing message here is the success in reversing depletion in at least 16% of aquifers and slowing depletion in another 18% of aquifers.

Line 77: What period does your study cover? Is it up to 2022? It would be helpful to summarise the length of the time series used in the analysis, you could perhaps use a heatmap (or several heatmaps by region for example) to do this for all sites.

Fig 1 – hard to read in its present form. Can you improve the resolution of the main and inset maps?

Line 100 – 102: Did you consider using a Mann-Kendal analysis to look at trends? If not, why not? It could add significant confidence to your results here, particularly if you used a bootstrapping method. (Having now read the methodology, why not mention that you also used other methods here and that all methods produced consistent results).

Fig 2 – this figure would lend itself to use of a Mann-Kendal analysis. It could potentially make it more rigorous. Hard to read in its present form. Can you improve the resolution of the main and inset maps?

Line 146 – 155: Could you not look at increases in well numbers as a proxy for this? This type of data is certainly available from a significant number of areas to make it possible for a selection of basins (e.g., IGB in South Asia).

Line 167 – 169: What do you mean by long time spans here? Recent research shows that groundwater levels increased significantly in the Indus basin over a period of 60 – 70 years, this is within one lifetime. While the development of surface water irrigation played a large part in that it does demonstrate that groundwater accumulation can happen quickly given the right conditions.

Line 227 – 229: You say deepening groundwater levels followed by shallowing groundwater levels is rare, however the preceding sentence suggests that as many as 16% of sites have experienced this. What is your definition of rare?

Line 212 – 249: So, 16% of sites experienced increased groundwater levels and 18% experienced slowing declines. That equates to 34% of sites where the situation is better now than it was at the end of the 20th century. This is larger overall than the percentage of sites where groundwater level decline has accelerated in the 21st century. This is a significant finding. Overall, 46% of sites have experienced, slowing decline or rising groundwater level. This is surprising and I think is cause for optimism overall. It also illustrates that if you get management right, it can have significant positive impacts (although your point about the challenges associated with rising groundwater levels is well taken). I think you should stress this more in the manuscript.

Fig 4d – where are these time-series from? It would be good to provide a location perhaps on your inset world map adjacent to this plot.

Fig 4e – i – these are not all that clear. It took me a while to work out some of the locations, could you add an outline to each landmass?

Line 273 – 274: I do not necessarily take this message from Fig 4 or the preceding section of the manuscript, as discussed above. They are accelerating in 54% of aquifers, which is the majority, but not necessarily a disproportionate number of aquifer systems. In fact, this is a better picture than I might have imagined, and it is very interesting.

Line 276: Equally they may mask regions of groundwater accumulation (for example south-western Indian Punjab, which when aggregated at the scale of your analysis would not show up and is perhaps visible in your Supplementary Fig. 38).

Line 277 – 279: They also indicate significant management success in some key areas. I think this also needs to be discussed.

Line 297 – 299: As does the evidence of groundwater level rises in at least 28% of aquifers in the 21st century. This is an important part of the story, and if we don't learn from success (particularly those 16% of aquifers that have seen declines reversed) then what can we learn from?

Line 312 – 313: Could you add a heatmap to the supplementary materials that illustrates the time

series for each well, and highlights data gaps, for each country/continent (or whatever scale you think might be most appropriate)?

Line 327 – 336: What was the name of the machine learning algorithm you used? Can you provide more details about what it does?

Line 369 – 372: This approach gives the reader more confidence in the robustness of your analysis. I would suggest mentioning in the main text. Did you consider using a bootstrapped Mann-Kendall analysis as well? I think it would be valuable.

Line 439 – 442 and Supplementary Note 11: The regional analyses conducted are very interesting. I think they also illustrate the same point as I have tried to make above, mainly that the overall depletion vs accumulation narrative is much more nuanced than your discussion would suggest. Take for example, Supplementary Fig. 25, 26, 29, 30, 31, to my eye there are almost as many areas where accumulation is occurring as there is areas where depletion is occurring. Supplementary Fig 27 is a notable exception, depletion certainly appears to be the dominant trend here. The plots in Supplementary Note 13 leave me with similar impressions.

Line 459 – 462: Could you not test this hypothesis? It would potentially be useful to try and get a hold on what percentage of wells this might have applied to. A graphical summary of the time-series might help but there are possibly other ways you could assess this, perhaps looking at trends prior to data gaps or looking at absolute values (e.g. if groundwater is very shallow just prior to a data gap, might it not suggest a cause other than drying for the termination of the time series)?

Line 466 – 467: the evidence from South Asia in the 20th century is that a period of substantial accumulation took place and that at the start of the 20th century groundwater levels were much deeper than they are now (despite the significant levels of groundwater depletion in the last 30 years or so). It would be good to be explicit about that here.

References

- Aeschbach-Hertig, Werner, and Tom Gleeson. "Regional strategies for the accelerating global problem of groundwater depletion." *Nature Geoscience* 5.12 (2012): 853-861.
- Bierkens, Marc FP, and Yoshihide Wada. "Non-renewable groundwater use and groundwater depletion: a review." *Environmental Research Letters* 14.6 (2019): 063002.
- Chen, J. L., Wilson, C. R., Tapley, B. D., Scanlon, B., & Güntner, A. (2016). Long-term groundwater storage change in Victoria, Australia from satellite gravity and in situ observations. *Global and Planetary change*, 139, 56-65.
- Dangar, Swarup, Akarsh Asoka, and Vimal Mishra. "Causes and implications of groundwater depletion in India: A review." *Journal of Hydrology* 596 (2021): 126103.
- Feng, W., Zhong, M., Lemoine, J. M., Biancale, R., Hsu, H. T., & Xia, J. (2013). Evaluation of groundwater depletion in North China using the Gravity Recovery and Climate Experiment (GRACE) data and ground-based measurements. *Water Resources Research*, 49(4), 2110-2118.
- Konikow, L. F. (2015). Long-term groundwater depletion in the United States. *Groundwater*, 53(1), 2-9.
- Konikow, L. F., & Kendy, E. (2005). Groundwater depletion: A global problem. *Hydrogeology Journal*, 13, 317-320.
- Scanlon, B. R., Faunt, C. C., Longuevergne, L., Reedy, R. C., Alley, W. M., McGuire, V. L., & McMahon, P. B. (2012). Groundwater depletion and sustainability of irrigation in the US High Plains and Central Valley. *Proceedings of the national academy of sciences*, 109(24), 9320-9325.
- Wada, Y., Van Beek, L. P., Van Kempen, C. M., Reckman, J. W., Vasak, S., & Bierkens, M. F. (2010). Global depletion of groundwater resources. *Geophysical research letters*, 37(20).

Referee #3 (Remarks to the Author):

Review of Nature manuscript 2023-04-05873, "Rapid and accelerating groundwater level declines in global cultivated drylands", Scott et al.

This manuscript details a compilation and analysis of in-situ measurements of groundwater-level trends in 178,000 globally distributed wells and provide new constraints on the prevalence of rapid and accelerating groundwater level declines and their correlation with land use and climatic drivers.

The work's novelty lies in its impressive groundwater datasets ~1.5 million monitoring wells spanning more than 30 countries and calculated groundwater level trends for the remaining ~178,000 wells via robust regression of annual median groundwater levels (these Theil-Sen slopes, for each monitoring well). The dataset itself is of clear scientific merit and will be an asset to researchers and regional water resource managers, and the corresponding analysis and discussion implicate contextualization as vital for future groundwater sustainability and management.

My comments and minor points are outlined below. I recommend the manuscript may be published with some clarifications or corrections.

Concerns :

1. Authors studied 207 aquifer systems/ trends in 1525 globally distributed aquifer systems and mention that the groundwater level trend show that the levels are becoming deeper over time but in many regions there are multiple aquifers and in most of the cases it has been found that the shallow aquifers are becoming dry in one season but yielding water in the rainy or after rainy season. Have the authors thought in this direction? (However, authors have mentioned in the limitations section about this, but still the location details are not clear and should be mentioned in the text)

2. In other case, the shallow aquifers are dried up and abandoned and the deeper aquifers are tapped for getting water.

So in either case whether observations derived mostly from shallow aquifers are sufficiently representative, in addition to whether the length of time over which the observations are taken is sufficient, to draw robust conclusions based on this analysis.

If most of the wells that authors used are shallow and those do not provide the clear picture of the recent groundwater pumping.

I wonder if authors can provide separate analysis from the shallow and deep wells to show that there are considerable differences in the trends or not.

3. Authors have taken a good number of wells for their study but the local variations are too high in some regions, given these uncertainties in observations, it is not clear that how many wells truly represent the selected regions for the estimation of water level changes.

4. Although there is spatial variability in the wells of a region in 21st century, Is there any spatial correlation in wells for a selected region?

5. Authors mention that even management strategies are insufficient for check the decline in groundwater levels, authors should provide a detailed examination in that. It has been found that in some of the aquifer systems the management strategies are fruitful in augmenting groundwater levels.

6. Authors have mentioned about the groundwater level trends but not provided any details about the precipitation variability (interannual, decadal and that from climate change)

7. Please make sure color scales used in figures are colorblind friendly.

Author Rebuttals to Initial Comments:

Referee comments are in **bold black text**

Our replies to comments are in **purple text (Times New Roman)**

Text that we revised upon reflecting on one or more Referee comments is in **light blue text (Arial)**

Referees' comments:

Referee #1 (Remarks to the Author):

Thank you for the opportunity to review the paper “Rapid and accelerating groundwater level declines in global cultivated drylands”. This is an important research topic and the paper is well written. The authors collate well-level data around the world and assess groundwater level trends.

We thank Reviewer #1 for their review and helpful comments. The revisions we made after reflecting on these comments have improved the manuscript.

They show that groundwater levels have declined in the past few decades in many of the world’s regional aquifers. However, this study suffers from a major methodological limitation and is also similar to several other studies in the literature. For this reason, unfortunately, this study is not suitable for a publication in Nature at the present time.

We thank Reviewer #1 for their recommendation to revisit our filtering process. We have made considerable updates to our analyses in response to their helpful recommendation, and detail these below (please see details specified in our reply comment immediately below this).

Methods:

The authors compiled water level time series from ~1.5 million monitoring wells spanning more than 30 countries. However, these data were then filtered to exclude potential outliers and wells lacking sufficiently long time series for trend analysis. Once all is said and done, the authors are left with ~178,000 wells.

This means that only ~12% of the well data is used in the analysis. I am concerned that such a strong selection bias will lead to unrepresentative results. There is a lot of information for the 1.2 million wells that are not included in this study that may lead to different findings. The wells that remain in the analysis are unique, especially since only 12% of them pass the statistical requirements of the authors. This means that the results will reflect these unique wells and may make it difficult to draw generalizable conclusions.

What is unique about the included wells? How are they different to the excluded wells? How do the results change when all wells are considered (even using two data points for time series for wells with short time periods available).

We thank you for this helpful comment. This comment helped us to realize that our original manuscript was insufficiently clear with respect to our data filtering process. We have made two updates to our work in response to this comment, and each made the manuscript clearer, in our view (please see “Update One of Two” and “Update Two of Two” below):

First, we added a new sensitivity analysis to our work that examines how our results would differ if we apply a less-stringent data screening process, allowing us to include more wells in our analyses. Specifically, we re-ran our entire analysis under three scenarios that require the earliest and the most-recent annual median groundwater level to be offset by (i) at least 8 years (main text results), or (ii) at least 5 years, or (iii) at least 3 years. The latter two scenarios (minimum offset between the first and most-recent measurement of at least 5 years or at least 3 years) lead to more wells being included in calculations of aquifer-scale trends than those considered in our main text analyses; specifically, scenario (i) leads to 169,717 monitoring wells with adequate data for analyses, scenario (ii) leads to 221,521 wells, and scenario (iii) leads to 256,407 wells. In the revised Supplementary Information, we present a map of these wells (under each of the three scenarios) and calculate aquifer-scale trends in depth to groundwater based on these alternate scenarios. This sensitivity analysis improved our manuscript, and demonstrates that our findings are not substantially different if we reduce the stringency of our criteria for including monitoring wells in our analyses. Our new supplementary information section and its figures reads as follows:

“Our main text (e.g., Figs. 1 and 2) focuses on groundwater level trends during the early 21st century. One of the criteria used to exclude monitoring wells with insufficiently long time series is that the first and the most-recent calendar year in the time series must be separated by eight years. Here, we present groundwater level trends for a variety of other time intervals, and apply a less stringent threshold for exclusion of monitoring wells (with respect to the temporal offset between the earliest and most-recent groundwater level measurements for a given time interval). The following table details these additional trend analyses and the minimum temporal offset between the earliest and most-recent groundwater level measurements for a given time interval (in order for a well to be included in the analysis).”

Additional trends in depth to groundwater presented in this section

Time interval	Minimum temporal offset between the earliest and most-recent measurements in the time series	Number of monitoring wells meeting criteria for analyses	Figure where these results are presented
2000-2022	8 years	169,717	*main text Figs.
2000-2022	5 years (“alternate scenario A”)	221,521 (pink in map below)	Fig. below
2000-2022	3 years (“alternate scenario B”)	256,407 (yellow in map below)	Fig. below

** for context, this row of the table provides information about the number of wells meeting our criteria for analyses as presented in the main text*

Supplementary Fig. 11. Groundwater monitoring wells meeting varying criteria for analyses. Dark grey points represent the $n=169,717$ monitoring wells meeting our criteria for analyses in the main text (i.e., these are the ~170 thousand points plotted in Fig. 2 and that form the basis for the aquifer-scale statistics presented in main text Fig. 2; these monitoring wells have 21st century groundwater level time series with at least 8 years between the earliest and most-recent 21st-century annual median groundwater level). Pink points represent monitoring wells that did not have an adequate number of years for inclusion in our main text Fig. 1, but do have groundwater level time series with at least 5 years separating the earliest and most-recent groundwater level measurement in the 21st century (see “scenario A” in supplementary table above). Yellow points represent monitoring wells that did not have an adequate number of years for inclusion in our main text Fig. 1 nor enough for inclusion in our “alternative scenario A”, but do have groundwater level time

series with at least 3 years separating the earliest and most-recent groundwater level measurement in the 21st century (see “scenario B” in supplementary table above). a) Global map. b-k) Maps for individual regions. Overall, including groundwater level time series that span a fewer number of years than the sites presented in main text (i.e., the locations with pink and yellow in this map that do not have grey points) does not add considerably to the spatial coverage of monitoring wells; nevertheless, the few regions that are not represented in our main text results but could have been had we made our minimum criteria for a monitoring well to be studied less stringent (e.g., lowering the minimum number of years separating the earliest and most-recent annual median groundwater level from 8 years to either 5 or 3 years) include Poland, Lithuania and northern Italy.

Supplementary Fig. 12. Sensitivity of aquifer-scale trends in depth to groundwater to the selection of a minimum number of years separating the earliest and most-recent annual median groundwater level (5 years versus 8 years presented here). Each circle represents one aquifer system and its 21st

century trends in groundwater to groundwater (median Theil-Sen slope of monitoring wells within the boundaries of the aquifer system). The x-axis values are the aquifer-scale trends in depth to groundwater presented in the main text (e.g., Fig. 2). Points on the y-axis are aquifer-scale trends in depth to groundwater if we reduce the stringency of our criteria to include a monitoring well in our analyses, specifically by reducing the minimum temporal offset between the earliest and most-recent calendar year with at least one measurement from 8 years (criteria for inclusion in the main text) to, instead, just 5 years. Points within the light grey shaded area have x-axis and the y-axis values that differ by less than 0.1 m/year. Overall, the great majority of aquifer-scale trends in depth to groundwater do not change substantially when we reduce our minimum temporal offset from 8 years to 3 years, as evidenced by the 91.7% of aquifer-scale trends in depth to groundwater that differ by less than 0.1 m/year (points within the light grey diagonal band) and the 97.9% of aquifer-scale trends in depth to groundwater that differ by less than 0.3 m/year.

Supplementary Fig. 13. Sensitivity of aquifer-scale trends in depth to groundwater to the selection of a minimum number of years separating the earliest and most-recent annual median groundwater level (3 years versus 8 years presented here). Each circle represents one aquifer system and its 21st century trends in groundwater to groundwater (median Theil-Sen slope of monitoring wells within the boundaries of the aquifer system). The x-axis values are the aquifer-scale trends in depth to groundwater

presented in the main text (e.g., Fig. 2). Points on the y-axis are aquifer-scale trends in depth to groundwater if we reduce the stringency of our criteria to include a monitoring well in our analyses, specifically by reducing the minimum temporal offset between the earliest and most-recent calendar year with at least one measurement from 8 years (criteria for inclusion in the main text) to, instead, just 3 years. Points within the light grey shaded area have x-axis and the y-axis values that differ by less than 0.1 m/year. Overall, the great majority of aquifer-scale trends in depth to groundwater do not change substantially when we reduce our minimum temporal offset from 8 years to 3 years, as evidenced by the 89% of aquifer-scale trends in depth to groundwater that differ by less than 0.1 m/year (points within the light grey diagonal band) and the 96.8% of aquifer-scale trends in depth to groundwater that differ by less than 0.3 m/year.”

— Update Two of Two —

We have added a new Supplementary Note devoted to highlighting how groundwater level time series were analysed. Specifically, we created a new Supplementary Figure to better highlight how only a small proportion of monitoring wells have any data of relevance to 21st century groundwater level trends. For example, our revised work now includes a statement highlighting that fewer than half of the monitoring wells in our dataset have more than one measurement in two different years; a new statement in our Supplementary Note 2 reads as follows:

“There are 1.23 million monitoring wells in our dataset that report at least one well water level measurement... But, of these 1.23 million wells, only ~45% (559 thousand wells) report at least one water level measurement in two different calendar years... Furthermore, only 28% (340 thousand wells) report at least one water level measurement in two different calendar years during the 21st century”

The Reviewer’s comment was helpful and motivated us to create this figure; our original manuscript created a misleading view that we were analyzing only a small share of available records when, in fact, only a small share of monitoring well data have more than one water level measurement in the 21st century. The new figure we created also draws attention to the new supplementary analysis detailed earlier in this response (i.e., “Update One of Two” above) by stating the number of monitoring wells meeting these less-stringent criteria for analyses (i.e., the values on the y-axes of the new supplementary figures, which as also presented on the preceding pages of this response letter).

The new figure (appearing in a new Supplementary Note 2 in our revised work) is shown below:

Supplementary Fig. 4 Monitoring wells in the compiled database that meet a suite of different criteria. (top row) There are 1.23 million monitoring wells in our dataset that report at least one well water level measurement. (second row). But, of these 1.23 million wells, only ~45% (559 thousand wells) report at least one water level measurement in two different calendar years. (third row) Furthermore, only 28% (340 thousand wells) report at least one water level measurement in two different calendar years during the 21st century. (fourth row) Of these monitoring wells, only 251 thousand report at least one well water level measurement in at least two different 21st-century calendar years that are separated by at least 3 years. (fifth row) Of these monitoring wells, only 222 thousand report at least one well water level

measurement in at least two different 21st-century calendar years that are separated by at least 5 years. (bottom row) Of these monitoring wells, only 170 thousand report at least one well water level measurement in at least two different 21st-century calendar years that are separated by at least 8 years (results for these monitoring wells are presented in main text Fig. 1, and these 170 thousand monitoring wells form the foundation of the analyses presented in the main text). For a comparison of aquifer-scale trends in depth to groundwater presented in the main text versus results that are based on a less-stringent criteria (i.e., the earliest and most-recent 21st century calendar years with at least one groundwater level measurement must be separated by at least 3 years (or at least 5 years), see Supplementary Note 6.

Related studies:

- **Fan et al (2013) “Global patterns of groundwater table depth”, Science. In this study, the authors also use direct well measurements to estimate groundwater levels around the world.**
- **Jasechko and Perrone (2021) “Global groundwater wells at risk of running dry”, Science. In this study, several of the study authors map and quantify well construction depth over time, in order to improve our understanding of spatiotemporal patterns of well locations and depths. Well locations and depths from this study would be closely correlated with groundwater pumping locations and levels (what this study shows).**
- **Perrone and Jasechko (2019) “Deeper well drilling an unusustainable stopgap to groundwater depletion”, Nature Sustainability. In this study, the authors analyze spatial patterns of groundwater well depths and purposes in the US. Importantly, they find a strong correlation between groundwater-level time-series trends and groundwater-well construction depth time-series trends. This means that the findings of the submitted paper are similar to those in Jasechko and Perrone (2021).**
- **Several high-profile studies have shown that groundwater levels are declining around the world using different methods to the submitted paper. For example, several studies have used satellite imagery or models to show that global groundwater levels are declining. The main novelty in this study then is that well-level data is used to show that groundwater levels are declining. However, this is not the first time that well-level records have been used to reveal global groundwater levels (e.g., as in Fan et al 2013) or relevant trends (e.g., as in Jasechko and Perrone et al 2021). For this reason, I do not think using well-level data is sufficiently novel to warrant publication in Science for this study.**

We appreciate Reviewer #1’s comment because it inspired us to revise our manuscript to be clearer about how our work moves beyond the prior literature. Specifically, we made multiple additions to the manuscript to highlight how this work builds on existing literature:

- 1) We added a new sentence to our summary paragraph (abstract) highlighting how our work evidences cases of groundwater recovery, highlighting a more nuanced picture of global groundwater level change over time. The new sentence in our revised manuscript reads as follows:

“However, our data also reveal specific cases in which depletion trends have reversed following policy changes and surface-water diversions, demonstrating the potential for depleted aquifer systems to recover”

- 2) We also added new statements to our main text, that draws attention to work that has been done previously and then highlights how cases where groundwater losses have been slowed or reversed have received less attention. We thank Reviewer 1 for motivating us to be clearer about how our work is distinct from previous publications; the specific paragraph where we make this case (by highlighting how our work focuses on quantifying cases of slowing and recovering groundwater level trends) reads as follows:

“Many previous studies^{1,2,3,4,5,6,7,8,9,10} have highlighted groundwater losses, but the potential for slowing or reversing these losses has received less attention. Our analysis of groundwater levels suggests that long-term groundwater losses are neither universal nor inevitable. Specifically, in half (49%) of the 542 aquifer systems in our analysis, groundwater level declines have decelerated (i.e., slowed; orange in Fig. 3; 20%) or reversed (blue in Fig. 3; 16%), or groundwater levels have continued to rise (purple in Fig. 3; 13%).”

- 3) One aspect of our work that is novel is our delineation of 1,693 aquifer systems using geographic information system software (these data are included as a supplementary dataset: specifically, Source Data for Fig. 2). We include the following statement in our revised manuscript to highlight this novel dataset:

“To evaluate aquifer-scale groundwater level trends, we manually delineated the boundaries of 1,693 aquifer systems—areas underlain by one or more aquifers—using maps and descriptions from 1,236 local and regional studies (see Methods and Supplementary Note 7).”

- 4) The compiled monitoring well data allow us to examine not only differences in groundwater level changes over time in different x,y locations, but also enable an analyses of differences in groundwater level trends among shallower versus deeper wells; to the best of our knowledge this vertical variability in groundwater level changes over time is not widely discussed in the literature on global groundwater depletion. The Reviewer’s comment motivated us to substantially revise and add to our analysis that juxtaposes shallow versus deep groundwater monitoring wells by specifically calculating aquifer-scale trends based solely on shallower monitoring wells (<30 m) versus deeper monitoring wells (>100 m). These new results highlight how some aquifer systems are undergoing rapid hydraulic head declines in deeper portions of the aquifer system (relative to hydraulic head changes over time in shallower portions of these same aquifer systems), and are presented in our Supplementary Information, which reads as follows:

“To further compare groundwater level trends in shallow versus deep wells, we identified aquifer systems with both deep (>100 m) and shallow (<30 m) monitoring wells.

Specifically, we identified n=232 aquifer systems with at least 4 shallow (<30 m deep) monitoring wells and at least 4 deep (>100 m) monitoring wells with sufficient data to calculate a 21st century groundwater level trend (median Theil-Sen slope; see main text methods). For each of these n=232 aquifer systems, we calculated two aquifer-scale groundwater level trends: (i) an aquifer-scale groundwater level trend based solely on shallow (<30 m) monitoring wells (median Theil-Sen slope of all shallow monitoring wells), and (ii) an aquifer-scale groundwater level trend based solely on deep (>100 m) monitoring wells (median Theil-Sen slope of all shallow monitoring wells). We then compared these aquifer-scale groundwater level trends across all n=232 aquifer systems (figure below).

Among these n=232 aquifer systems, we show that aquifer-scale groundwater level trends based on deeper monitoring wells tend to have higher-magnitude values than trends based on shallow monitoring wells. Further, aquifer-scale groundwater level trends based on deep monitoring wells exceed trends based on shallow monitoring wells in a disproportionate number of aquifer systems (152 of 232 aquifer systems), implying that deepening groundwater level trends (or slower groundwater level shallowing trends) are more prevalent among deep monitoring wells compared to shallow monitoring wells (in these n=232 aquifer systems). Some of the aquifer systems where groundwater level deepening trends based on deep monitoring wells are outpacing deepening trends observed in shallow monitoring wells are in California’s Central Valley (e.g., Tulare Lake Basin, Kaweah Basin, Kings Basin, Chowchilla Basin; see figure below). That groundwater levels are falling faster in deeper monitoring wells (relative to shallower monitoring wells) may imply a strengthening of downward-oriented hydraulic gradients in these areas, which can speed the movement of shallower groundwater to deeper aquifers, potentially impacting the quality of these deep groundwaters if the shallow groundwater contains mobile contaminants.

Supplementary Fig. 282. 21st-century aquifer-scale trends in depth to groundwater based on shallow versus deep monitoring wells. Each point represents one aquifer system. The x-axis values are the aquifer-scale groundwater level trend based solely on monitoring wells that have total depths of no more than 30 m (“shallow” monitoring wells; aquifer-scale trend determined by the median of all shallow monitoring wells with sufficient data to determine a 21st century groundwater level trend – see main text methods for details). The y-axis values represent aquifer-scale groundwater level trend based solely on monitoring wells that have total depths of greater than 100 m (“deep” monitoring wells; aquifer-scale trend determined by the median of all deep monitoring wells with sufficient data to determine a 21st century groundwater level trend). Larger-sized points represent aquifer systems where the aquifer-scale groundwater level trend based on shallow (<30 m) monitoring wells differs from the aquifer-scale

groundwater level trend based on deep (<100 m) monitoring wells by more than 0.1 m/year; among these points (i.e., large purple points and large green points), aquifer-scale trends based on deep wells tend to be greater than aquifer-scale trends based on shallow monitoring wells (i.e., the large purple circles outnumber the large green circles by a ratio of about 3:1). This analysis emphasizes that groundwater level trends in deep and shallow monitoring wells can differ even when these wells are located in the same aquifer system. However, we also highlight that in most of the aquifer systems shown here, the aquifer-scale groundwater level trend based on shallow monitoring wells differs from the aquifer-scale groundwater level trend based on deep monitoring wells by less than 0.1 m/year (147 of 232 aquifer systems).”

Data availability:

It is unfortunate that the authors are not able to provide the full suite of well data used in the study, making replication analyses difficult (about 12 countries are provided, out of over 30 countries included in the study; this is <50% of the data). It is also my understanding that several other related studies by some of the study authors have not been able to make the data necessary to replicate the results available. It is critical to share data to ensure reproducibility of scientific studies. I encourage the authors to share their data in future studies to support the scientific process.

We thank Reviewer #1 for their helpful comment. We agree, wholeheartedly, in the importance of sharing data for reproducibility and for the broader scientific process, as well as the importance of ensuring that shared data is in a format that is user friendly. We have made numerous efforts to make groundwater level data available (please see these details specified in “(A)” and “(B)” to follow):

(A) Obtained permission from more database managers to post groundwater level data:

- When we submitted our initial manuscript in April-2023, we had received permission to repost groundwater level data for n=16 regions: (1) Afghanistan, (2) Brussels and Wallonia (Belgium), (3) Alberta (Canada), (4) British Columbia (Canada), (5) Prince Edward Island (Canada), (6) Ontario (Canada), (7) Yukon (Canada), (8) China (Gong et al., 2018), (9) Croatia, (10) Czech Republic, (11) France, (12) Ireland, (13) New Zealand, (14) Paraguay, (15) Sweden, (16) USGS NWIS (USA).
- At the time we are submitting this revised manuscript, we have received permission to post groundwater level data from a much larger number of agencies; specifically, we have received permission to post groundwater level data for n=36 regions: (1) Afghanistan, (2) Belgium: Brussels, (3) Belgium: Wallonia, (4) Brazil, (5) Canada: Alberta, (6) Canada: British Columbia, (7) Canada: Manitoba, (8) Canada: Northwest Territories, (9) Canada: Ontario, (10) Canada: Prince Edward Island, (11) Canada: Saskatchewan, (12) Canada: Yukon, (13) China: Gong et al. (2018), (14) Croatia, (15) Czech Republic, (16) Austria, (17) Bulgaria, (18) Denmark, (19) Italy, (20) Latvia, (21) Lithuania, (22) Poland, (23) Slovenia, (24) Switzerland, (25) France and French Guiana, (26) Germany, (27) Guam, (28) Ireland, (29) Israel, (30) New Zealand, (31) Norway, (32) Paraguay, (33) Sweden, (34) United States of America: GAMA, (35) United States of America: USGS, (36) United States of America: TWDB. In the event that we receive permission to post groundwater level data from a manager of one of the remaining databases that we analyse in our work, we shall add these data to the compilation of groundwater level data that we have received permission to post.
- We added a new column to our Supplementary Information table devoted to describing datasets entitled “*Written permission received to post annual ground-water level data*”. We have added the word “Yes” in all cases where we have received permission to post groundwater level data (and

we include groundwater level data as a supplementary data file appended to this publication).

- We also calculated the total proportion of annual groundwater levels that we have now received permission to share. We added the following statement to our ‘Data Availability Statement’ within the main text highlighting that we have received permission to share more than half of the annual groundwater levels that we analyse in our study:

“Annual groundwater level data are available here in all cases where we have received permission from a database manager to post data; the supplementary dataset provides groundwater level data for: Afghanistan¹⁰⁸, Austria, Belgium, Brazil, Bulgaria, Canada (Alberta, British Columbia, Manitoba, Northwest Territories, Ontario, Prince Edward Island, Saskatchewan, Yukon), China¹⁰⁹, Croatia, Czech Republic, Denmark, France¹¹⁰, Germany, Guam, Ireland, Israel, Italy, Latvia, Lithuania, New Zealand, Norway, Paraguay, Poland, Slovenia, Sweden, Switzerland, the United States (Groundwater Ambient Monitoring and Assessment Program, US Geological Survey’s (USGS) National Water Information System, and the Texas Water Development Board). The databases for which we have received written permission to post annual groundwater level data encompass 59% of annual groundwater level data analysed here (specifically, we received permission to post 66% (n=4,170,802 of n=6,314,793) of all annual ‘depth to groundwater’ data, and 18% (n=190,879 of n=1,049,502) of all ‘groundwater elevation’ data). These datasets are specified in Supplementary Table 1 (see column entitled ‘Written permission received to post annual groundwater level data’). Source Data for each of the main text figures are available here.”

(B) Revised our Supplementary Information to provide specific instructions to access datasets:

- We have added further details to our supplementary information in an effort to help others access groundwater level data. Specifically, we added (and, in cases, updated) hyperlinks and names of agencies for each database in an effort to help others access underlying datasets, even those where we ourselves did not receive written permission from a database manager to post groundwater level data.

Specifically, our revised Supplementary Information provides a more detailed explanation of the steps we took to download and pre-process individual databases. For example, for our revised manuscript we re-downloaded the United States Geological Survey’s NWIS dataset (i.e., the results presented in this revised manuscript are based on a more recent download (on May-25-2023) of this USGS database than the data analysed and presented in our original submission). We were motivated to re-download (and re-complete our quality control steps) the NWIS database after reading a USGS blog detailing changes made to their groundwater-level observation database (<https://waterdata.usgs.gov/blog/groundwater-data-update/>). The updates that the USGS made to their database enabled us to identify well water level measurements that do not represent static water levels (e.g., water level measurements made during pumping tests). Our revised manuscript excludes such measurements; consequently, the number of monitoring wells with sufficient data for analyses changed from our originally submitted manuscript (178 thousand monitoring wells) to this revised manuscript (170 thousand monitoring wells). The specific steps we took to exclude well water level measurements that may not represent static water levels are detailed in our revised Supplementary Note 1.34, which reads as follows:

*“We downloaded groundwater level data on May 25, 2023 from the United States Geological Survey REST portal (e.g., <https://waterservices.usgs.gov/nwis/gwlevels/?format=rdb&countyCd=COUNTY&startDT=1800-01-01&endDT=2023-05-25&siteType=GW&siteStatus=all>, where the word “COUNTY” was replaced with the corresponding county’s FIPS code; e.g., “01009”). Second, we retrieved station information (e.g., locations, well depths) from the National Water Information System (<https://waterdata.usgs.gov/nwis>) and joined the groundwater level data to the station information via the field entitled “site_no”. We completed several data cleaning steps prior to analysing these data:
First, we excluded the following site types (based on values in the Site Type Code heading “site_tp_cd”):*

- We excluded all rows in the dataset with a “site_tp_cd” value of “GW-MW” (“Multiple wells” – “A group of wells that are pumped through a single header and for which little or no data about the individual wells are available.” quoting <https://maps.waterdata.usgs.gov/mapper/help/sitetype.html>, accessed May 29, 2023);
- We excluded all rows in the dataset with a “site_tp_cd” value of “GW-IW” (“Interconnected wells” – “Collector or drainage wells connected by an underground lateral.”)
- We excluded all rows in the dataset with a “site_tp_cd” value of “GW-CR” (“Collector or Ranney type well” – “An infiltration gallery consisting of one or more underground laterals through which groundwater is collected and a vertical caisson from which groundwater is removed. Also known as a “horizontal well”. These wells produce large yield with small drawdown.”)
- We excluded all rows in the dataset with a “site_tp_cd” value of “GW-TH” (“Test hole not completed as a well” – “An uncased hole (or one cased only temporarily) that was drilled for water, or for geologic or hydrogeologic testing. It may be equipped temporarily with a pump in order to make a pumping test, but if the hole is destroyed after testing is completed, it is still a test hole. A core hole drilled as a part of mining or quarrying exploration work should be in this class.”)

Second, we excluded the following water level measurements (based on data recorded in the field “lev_status_cd”):

- We excluded all rows in the dataset with a “lev_status_cd” value of “2” (corresponding to “True value is below reported value due to local conditions”)
- We excluded all rows in the dataset with a “lev_status_cd” value of “3” (corresponding to “True value is above reported value due to local conditions”)
- We excluded all rows in the dataset with a “lev_status_cd” value of “6” (corresponding to “Measurement unable to be obtained due to local conditions”)
- We excluded all rows in the dataset with a “lev_status_cd” value of “7” (corresponding to “Groundwater level affected by brackish or saline water”)
- We excluded all rows in the dataset with a “lev_status_cd” value of “C” (corresponding to “Frozen”)
- We excluded all rows in the dataset with a “lev_status_cd” value of “D” (corresponding to “Dry”)
- We excluded all rows in the dataset with a “lev_status_cd” value of “O” (corresponding to “Obstructed”)
- We excluded all rows in the dataset with a “lev_status_cd” value of “P” (corresponding to “Pumping”)

Third, we excluded one groundwater level measurement with a “lev_src_cd” value of “0” (implying this water level measurement was only accurate to the nearest foot: “Water level accuracy to nearest foot” quoting https://help.waterdata.usgs.gov/code/water_level_acc_cd_query?fmt=html). Fourth, we excluded groundwater level measurements with a Water-level Approval-Status Code of “P” – “Provisional data subject to revision”. Fifth, upon examining recorded measurement dates within the dataset, we found a large number of measurements had a recorded date (“lev_dt” values) of January 1, 1900; we excluded

these measurements from our compiled database (as we were uncertain if these recorded measurement dates in the dataset were accurate).

Dates that were reported to the nearest month were reported to the first day of the month. Similarly, dates that were reported to the nearest year were reported to the first day of the year. We downloaded station metadata (e.g., well depths, information about the types of aquifers (e.g., unconfined or confined) some of the monitoring well water level times series capture) on May 29, 2023 via <https://waterdata.usgs.gov/nwis/inventory>, and joined this station information to the well water level data via a join based on values in the field entitled "site_no". Two fields in the dataset provide information about the depths of wells – ("well_depth_va" and "hole_depth_va"). If a non-zero value was recorded for "well_depth_va" then we recorded this value as the estimated well depth; if no depth estimate was available via the field "well_depth_va" but a non-zero value was recorded in the field "hole_depth_va", we recorded this "hole_depth_va" as the estimate for the well depth.

Last, we excluded n=25 stations that lack decimal latitude and longitude values in the NWIS 'Station Information' portal (e.g., station number 485037104070301)."

Referee #2 (Remarks to the Author):

General comments:

This is a well written, expansive, technically sound, and rigorous study. I thoroughly enjoyed reading it. The methodology is well described and defensible. You have managed to pull together a unique and substantial global dataset which is a brilliant achievement.

We thank Reviewer #2 for their positive comments and their helpful recommendations. The revisions we have made to our manuscript (specified below) resulting from their comments improved our manuscript considerably. Thank you.

Further, we thank Reviewer 2 for their specific positive comment regarding the global dataset. Since we submitted our original manuscript, we have also received permission to post groundwater level data for a considerable number of databases. Should our work be accepted for publication, these data will be posted with the hope that they prove valuable to other research groups.

The data could do with improved description (i.e., particularly the nature of the time series used in the analysis, some additional visualisation may help).

Thank you for this helpful recommendation. Reviewer 1's comments also highlighted how our original manuscript failed to provide adequate clarity about our underlying data and its processing.

We have now added numerous new visualisations in Supplementary Note 2 to better highlight the steps applied to screen the compiled data. The new visualisations of our data screening process have, in our view, improved the clarity of the work. Thank you for recommending we do this. The new visualisations are detailed in the following material derived from our revised work (i.e., the following content is quoted from revised Supplementary Information):

"2. GROUNDWATER LEVEL DATA PRE-PROCESSING

We completed a series of steps to quality control our piezometric dataset and analyse 21st century groundwater level trends. The following subsections detail steps we applied to:

- i. exclude potentially erroneous well water level measurements (see Supplementary Note 2.1 entitled "Excluding potentially erroneous well water level measurements"); and,*
- ii. calculate annual median groundwater levels (see Supplementary Note 2.2 – "Calculating annual median groundwater levels"); and,*
- iii. identify monitoring wells with a time series spanning a sufficient range of calendar years to characterize groundwater level trends for the 21st century (see Supplementary Note 2.3 – "Identifying records capturing 21st-century groundwater level trends").*

2.1 Excluding potentially erroneous well water level measurements

As detailed in the main text methods, we excluded (i) extreme values of depth to groundwater (i.e., >1,000 m and <-1,000 m) and implausibly high groundwater elevations (i.e., >8,000 m above sea level), and (ii) groundwater-level measurements with values of '999', '-9999', or '0', because some databases used these values as a code for missing measurements (see figures on the following pages).

*Further, we also excluded outlier values detected by a machine-learning algorithm (see Taylor, S.J., & Letham, B. Forecasting at scale. *The American Statistician* 72, 37–45 (2018)). The algorithm is entitled*

“Prophet”, and is specifically designed to automate time series forecasting (for details: https://facebook.github.io/prophet/docs/quick_start.html#python-api; accessed July 25, 2023). It is based on an additive regression model (see Friedman, J.H., & Stuetzle, W. Projection pursuit regression. *Journal of the American statistical Association* 76, 817–823 (1981)), where non-linear trends are fit with yearly, weekly, and daily seasonality, plus holiday effects (Taylor, S.J., & Letham, B. Forecasting at scale. *The American Statistician* 72, 37–45 (2018)). For our purposes, we do not use Prophet to forecast time series but, rather, we use Prophet to determine the confidence interval of the model, which is then compared to the actual measurements in our time series. The main advantage of Prophet over other machine-learning tools for time series forecasting is that it automates the forecast performance evaluation when determining an optimal model. Prophet only requires the time series (e.g., measurement date and groundwater level) and the number of iteration steps to optimize the model. In our case, we used 300 Monte-Carlo steps and 1000 uncertainty iterations (for further details for these parameters, see the Prophet manual: https://facebook.github.io/prophet/docs/quick_start.html#python-api; accessed July 25, 2023). For each groundwater monitoring station, we first trained the machine-learning algorithm (i.e., Prophet) using the corresponding groundwater level (or groundwater elevation) time series, creating a prediction of the depth to groundwater time series (or groundwater elevation time series) over the same timespan. Supplementary Figs. 1 and 2 depict a groundwater level time series (dark blue colours in supplementary figures to follow) together with the model prediction (violet in supplementary figures to follow). Excluded groundwater level values (i.e., those removed from the time series before applying the machine learning) are depicted as green points in Supplementary Figs. 1 and 2.

In addition to outputting a model prediction (which is shown as a dark purple line in Supplementary Figs. 1 and 2) Prophet also provides lower and upper confidence bounds for each prediction. Specifically, we use the 99% confidence bounds to calculate a band around the model prediction that allows us to classify the groundwater measurement values. We defined outliers as well water level measurements that fall outside of the range specified as plus (or minus) 0.75 times the 99% confidence interval (offset from the machine learning prediction; that is, the predicted value from the machine learning algorithm (dark purple line) plus-or-minus 0.75 times the 99 percent confidence interval; see light purple shading in Supplementary Figs. 1 and 2). These points (which fall outside of the range specified by the prediction plus or minus 0.75 times the 99% confidence interval) are depicted as orange circles in Supplementary Figs. 1 and 2.

This approach (i.e., machine-learning-based outlier detection method) was applied to each monitoring well with more than 15 groundwater-level measurements, yielding a prediction for each time step and its 99% confidence interval; we defined points to be outliers and excluded them if they fell outside of the range defined by the predicted groundwater level ± 0.75 times this confidence interval (see figures on the following pages). Later in this Supplementary Note 2 we compare (A) the aquifer-scale trends in depth to groundwater presented in the main text (which are based on groundwater level time series after we excluded measurements with the application of the machine learning algorithm), versus (B) aquifer-scale trends in depth to groundwater calculated without applying the machine learning algorithm to detect and exclude potential outlier measurements; these two aquifer-scale trends in depth to groundwater (i.e., (A) and (B) described in this paragraph) are similar for most aquifer systems.

Supplementary Fig. 1. Schematic of a groundwater level time series and how outlier values were excluded (this figure differs from the one below as it plots a time series in units of groundwater elevation (not depth to groundwater)). The dark blue circles and line represent the groundwater level time series (note: points within this time series were generated by us for illustrative purposes). Green circles represent points that were excluded because their groundwater elevation value exceeded 8000m (label “>8000m”), was less than 1000m below sea level (see label “<-1000m”), or were exactly 999 or -9999 (see labels “999m” and “-9999m”). The thick purple line represents the predicted groundwater level time series (based on a machine learning algorithm: see Taylor, S.J., & Letham, B. *Forecasting at scale. The American Statistician* 72, 37–45 (2018)). The light purple shaded area represents an offset from the prediction (i.e., a constant offset along the y-axis away from the purple line) of a magnitude of the 99% confidence interval (CI) multiplied by a factor of 0.75 (i.e., we calculated the 99% confidence interval, multiplied by a factor of 0.75, and then plotted this offset above (top of light purple shading; see label “+0.75xCI(99)”, corresponding to an addition of 0.75 times the 99% confidence interval from the machine learning model prediction) and below (bottom of light purple shading; see label “-0.75xCI(99)”, corresponding to a subtraction of 0.75 times the 99% confidence interval from the machine learning model prediction). The red circles represent points that fall outside of the light purple shading (i.e., the measurement was flagged as an outlier by the machine learning algorithm and our threshold for outliers of 0.75 times the 99% confidence interval).

Supplementary Fig. 2. Schematic of a groundwater level time series and how outlier values were excluded (this figure differs from the one above as it plots a time series in units of depth to groundwater (not groundwater elevation)). The dark blue circles and line represent the groundwater level time series (note: points within this time series were generated by us for illustrative purposes). Green circles represent points that were excluded because their depth to groundwater values exceeded 1000m (label “>1000m”), or were exactly 999 or -9999 (see labels “999m” and “-9999m”). The thick purple line represents the predicted groundwater level time series (based on a machine learning algorithm: see Taylor, S.J., & Letham, B. *Forecasting at scale. The American Statistician* 72, 37–45 (2018)). The light purple shaded area represents an offset from the prediction (i.e., a constant offset along the y-axis away from the purple line) of a magnitude of the 99% confidence interval (CI) multiplied by a factor of 0.75 (i.e., we calculated the 99% confidence interval, multiplied by a factor of 0.75, and then plotted this offset above (top of light purple shading; see label “+0.75xCI(99)”, corresponding to an addition of 0.75 times the 99% confidence interval from the machine learning model prediction) and below (bottom of light purple shading; see label “-0.75xCI(99)”, corresponding to a subtraction of 0.75 times the 99% confidence interval from the machine learning model prediction). The red circle represents a point that falls outside of the light purple shading (i.e., the measurement was flagged as an outlier by the machine learning algorithm and our threshold for outliers of 0.75 times the 99% confidence interval).

Calculating annual median groundwater levels

We calculated annual median groundwater levels (i.e., the median of all groundwater level measurements made for a given station, for a given year). The following figure presents a walkthrough of the steps applied to calculate annual median groundwater levels.

Supplementary Fig. 3. Calculation of annual median groundwater levels from well water level time series. (a) Location of a monitoring well in northern Iran (monitoring well is located within the boundaries of the West Qazvin Plain aquifer system). (b) Time series of groundwater levels recorded at the monitoring well in northern Iran. Each grey circle represents one recorded well water level measurements (units on y-axis are meters below the land surface). X-axis values represent the measurement date. (c) Calculation of annual median groundwater levels. Each black circle represents the median well water level for a given calendar year. The original time series of well water level measurements are presented as grey circles displayed behind the black circles. The black circles are equally spaced (one point per calendar year) with the exception of missing values (for calendar years during which no well water level measurements were made).

2.3 Identifying records capturing 21st-century groundwater level trends

We identified monitoring wells that capture 21st century trends in groundwater levels following a suite of criteria.

First, we excluded monitoring wells with just one water level measurement (rendering these wells unhelpful for examining changes in groundwater levels over time, as they have just a single snapshot of the groundwater level at a single point in time), or reporting groundwater level measurements that were all made during the same calendar year (rendering these monitoring wells unhelpful for examining changes in groundwater levels over multiple calendar years). This step – i.e., excluding monitoring wells where the water level time series takes place entirely within just one calendar year – excluded ~672 thousand monitoring wells (see figure below).

Second, we excluded monitoring wells that lack water level measurements made in two different calendar years during the 21st century (rendering these monitoring wells unhelpful for examining changes in groundwater levels over multiple calendar years in the 21st century). This step excluded ~219 thousand monitoring wells.

Third, we excluded monitoring wells where the earliest and latest calendar year (in the 21st century) where the monitoring well time series reports at least one measurement are separated by at least 8 years. This step – i.e., excluding monitoring wells where the 21st century time series spans fewer than 8 calendar years – excluded a further ~170 thousand wells (Supplementary Fig. 4; note: we present alternate results based on less-stringent thresholds (minimum of 5 years or minimum of 3 years separating the earliest and most-recent calendar years in a time series) in Supplementary Note 6).

Supplementary Fig. 4. Monitoring wells in the compiled database that meet a suite of different criteria. (top row) There are 1.23 million monitoring wells in our dataset that report at least one well water level measurement. (second row). But, of these 1.23 million wells, only ~45% (559 thousand wells) report at least one water level measurement in two different calendar years. (third row) Furthermore, only 28% (340 thousand wells) report at least one water level measurement in two different calendar years during the 21st century. (fourth row) Of these monitoring wells, only 251 thousand report at least one well water level measurement in at least two different 21st-century calendar years that are separated by at least 3 years. (fifth row) Of these monitoring wells, only 222 thousand report at least one well water level measurement in at least two different 21st-century calendar years that are separated by at least 5 years. (bottom row) Of these monitoring wells, only 170 thousand report at least one well water level measurement in at least two different 21st-century calendar years that are separated by at least 8 years (results for these monitoring wells are presented in main text Fig. 1, and these 170 thousand monitoring wells form the foundation of the analyses presented in the main text). For a comparison of aquifer-scale trends in depth to groundwater presented in the main text versus results that are based on a less-stringent criteria (i.e., the earliest and most-recent 21st century calendar years with at least one groundwater level measurement must be separated by at least 3 years (or at least 5 years), see Supplementary Note 6.

Overall, the results are very interesting, but I think you need to be careful when discussing their implications. I think the results indicate a much more nuanced picture than your discussion suggests. In fact, I would argue that the 46% of aquifer systems that have seen groundwater level declines slowing, reversed, or continuing to rise in the 21st century, have as much importance as the 54% of aquifers where declines have accelerated or remained constant. I think this nuance needs to be recognised in your discussion and the implications considered. There are important management lessons, which have not really been drawn out very effectively in the text, from areas where declines have slowed or reversed. I am concerned that your analysis was conducted with a presupposition that groundwater level decline dominates globally (see for example the narrative in Supplementary Note 10, looking for evidence of groundwater deepening) and that this has skewed the discussion of the findings.

My view is that in its present form the paper is not suitable for publication in Nature. The real novelty and importance of the findings has perhaps been drowned out by a focus on groundwater depletion, when, as highlighted above, it in fact it appears that there is a much more nuanced story

globally. That is not to say that significant depletion is occurring and is not a serious issue, but that your results suggest that there is almost an equal number of aquifer systems where depletion has slowed, reversed, or not occurred at all in the last 40 years or so. Perhaps, if this nuance was recognised and discussed then the paper would be suitable for publication in Nature.

We thank Reviewer 2 for this wise and important comment. This comment motivated us to make major changes to our manuscript. The changes were designed specifically to highlight the much more nuanced picture than the discussion section within our original manuscript. Here we detail five examples of substantial changes (i.e., (i) to (v) below) we have made to our work to help it to better recognize nuance in our results:

(i) We added new text to our summary paragraph (abstract) highlighting the nuanced picture elucidated by the global piezometric data, which the reviewer highlights. Specifically, we added the following concluding sentence to our abstract, to emphasize Reviewer 2's point about nuance in a prominent way:

"However, our data also reveal specific cases in which depletion trends have reversed following policy changes and surface-water diversions, demonstrating the potential for depleted aquifer systems to recover."

(ii) We reorganized our manuscript by presenting our comparison of groundwater level trends in the late-20th versus early-21st centuries earlier in the manuscript (specifically, we moved this section up such that it comes before our discussion of climate and cultivated land areas). We added a specific heading to our revised manuscript to emphasize the Reviewer's comment; this new subheading reads as follows:

"Slowing and reversing groundwater-level declines"

(iii) We have added considerable new discussion surrounding nuance to our manuscript, motivated by Reviewer 2's comment. For example, we have added the following new sentences to our main text:

"Specifically, in half (49%) of the 542 aquifer systems in our analysis, groundwater level declines have decelerated (i.e., slowed; orange in Fig. 3; 20%) or reversed (blue in Fig. 3; 16%), or groundwater levels have continued to rise (purple in Fig. 3; 13%)."

(iv) We have completed an even more thorough review of available literature on groundwater level shallowing, adding to our compilation of groundwater shallowing events (presented in the Supplementary Information (Supplementary Note 15.2)). Our revised compilation of groundwater shallowing trends (as presented in the Supplement) is shown below (and these data are also presented in main text Fig. 2 for locations where we lack groundwater level data):

Locations where groundwater level shallowing has been observed in one or more monitoring wells. Each circle represents the location of a local- to regional-scale study area where one or more published works have recorded groundwater level shallowing in one or more monitoring wells (for details for each study area, see table on the following pages). The colour of each circle corresponds to the average of the minimum and maximum literature values (see table on the following pages). Groundwater level shallowing has been captured in study areas beyond the countries for which we have monitoring well data (i.e., groundwater level shallowing in some countries that are not part of the analyses presented in the main text figures), including areas in Burkina Faso and Niger.

(v) We have substantially revised the concluding section of our manuscript, after considering Reviewer 2’s comment. Specifically, we have revised the title of the subsection to “*Depleting and recovering groundwater resources*” (instead of our original manuscript’s heading, which read as: “Depleting groundwater resources”). Further, we have added the following content to better highlight cases of groundwater recovery (and cases where rates of groundwater level decline have slowed):

“Our analysis also documents cases where groundwater declines have slowed or reversed after (a) the implementation of groundwater policies, (b) the alleviation of groundwater demand via surface water transfers, or (c) the addition of groundwater storage following managed aquifer recharge projects. To address the growing problem of global groundwater depletion, these kinds of success stories would need to be replicated in dozens of aquifer systems with declining groundwater levels. Thus, our analysis illustrates the potential for depleted aquifers to recover, while demonstrating how much work remains to be done to protect groundwater resources. By documenting global hotspots of groundwater level decline and recovery, this analysis can inform efforts to address rapid and accelerating groundwater depletion.”

However, I think that even then the authors need to make a much stronger case for its suitability for publication in Nature given the dominant (and defensible) narrative of groundwater depletion already in the literature and in the areas where you have data (e.g., Bierkens and Wada, 2019; Konikow and Kendy, 2005; Wada et al, 2010, Aeschbach-Hertig and Gleeson, 2012; and other related literature much of which you have already cited).

We thank Reviewer 2 for their recommendation. We agree with their suggestion and have revised our manuscript to better highlight aspects of our work that move beyond the dominant narrative of groundwater depletion established via previous works.

1. We have added the following specific statement to simultaneously highlight important works that have been published in the past, while also highlighting how our work focuses on both groundwater losses and recovery; the newly added sentence reads as follows:

“Many previous studies^{1,2,3,4,5,6,7,8,9,10} have highlighted groundwater losses, but the potential for slowing or reversing these losses has received less attention. Our analysis of groundwater levels suggests that long-term groundwater losses are neither universal nor inevitable. Specifically, in half (49%) of the 542 aquifer systems in our analysis, groundwater level declines have decelerated (i.e., slowed; orange in Fig. 3; 20%) or reversed (blue in Fig. 3; 16%), or groundwater levels have continued to rise (purple in Fig. 3; 13%).”

2. We have made numerous revisions to our manuscript to better highlight nuance, as recommended by the Reviewer in their earlier comment. We thank Reviewer 2 for recommending that we better highlight nuance in our results and discussion, as this nuance is itself novel (given the dominant narrative of groundwater depletion that the Reviewer notes in this comment, which we agree with). Specifically, we made five substantial revisions to our work to better highlight the nuanced nature of groundwater level changes in the 21st century (these 6 revisions are detailed above in our reply to Reviewer 2’s comment: specifically, our replies “(i)” through “(vi)” above)

3. We also revised the order and changed the amount of text devoted to sections of our manuscript to intentionally emphasize our comparison of early-21st versus late-20th century trends in depth to groundwater, as this examination of four-decades of groundwater level variation forms a considerable portion of the novelty of our analyses. Specifically, we deleted one paragraph from our explanatory variable analysis (subsection entitled “Groundwater declines in cultivated drylands”) and moved this section down so that it comes later in our manuscript; these changes allow us to (a) move the sections devoted to early-21st versus late-20th century trends so that they are more prominent (i.e., are discussed earlier) in our manuscript, and (b) so that we have more space to discuss these changes in trends over the past four decades. We have added considerable new discussion of these multi-decadal trends to our manuscript as demonstrated in the tracked changes document appended to this reply letter and in the “(i)” through “(v)” changes specified earlier in this response to Reviewer 2 letter.

4. One aspect of our manuscript that carries some novelty is our presentation of manually delineated aquifer system boundaries (these geospatial data will be made available alongside this work, should it be published). The reviewer’s comment – along with recommendation from Reviewer 1 – motivated us to be clearer about this novel component by revising our manuscript; specifically, we added the following statement to our main text:

“To evaluate aquifer-scale groundwater level trends, we manually delineated the boundaries of 1,693 aquifer systems—areas underlain by one or more aquifers—using maps from 1,236 local and regional studies”

My co-authors and I also gave a considerable amount of thought to changing the title to reflect the more nuanced (balanced) picture of changing groundwater levels described in the revised discussion section compared to our previous manuscript. We elected to remain with our original title, “Rapid and accelerating groundwater level declines in global cultivated drylands”, for the following reasons: (a) a key outcome of the manuscript – not previously reported from remotely sensed (GRACE) data – is (as reported in the Abstract): “... groundwater level declines have accelerated over the past four decades in a disproportionate share of the world’s regional aquifers” and that “This widespread acceleration in groundwater level deepening highlights an urgent need for more effective measures to address groundwater depletion.”; (b) the communication of three fundamental components of our analysis – the acceleration of groundwater-level declines (in cultivated drylands), evidence of interventions arresting groundwater depletion in areas of intensive groundwater abstraction, and that both outcomes are revealed by a global-scale analysis of compiled groundwater-level measurements – became too challenging to integrate clearly and coherently into a title of <80 characters including spaces (Nature’s guidelines); and (c) each of the three fundamental components of our analysis is clearly explained in the manuscript’s

abstract, providing a prominent position for our newly added discussion of nuance in the context of groundwater level variations over time.

Specific comments:

Line 41 – 42: I would argue that the global prevalence of groundwater depletion is already well known in the areas where your data is concentrated (e.g. Scanlon et al, 2012 and Konikow 2014 in the United States; Chen et al., 2016 in Australia; Dangar et al., 2021 India; Feng et al., 2013 in China; and many other example in the literature) and that many of these studies also use in-situ data. Having said this, I recognise that there are no other studies (at least as far as I am aware) that have integrated data from all these regions into one global analysis, and I think this is a big achievement, but it doesn't mean that the phenomenon was previously unknown.

Thank you. We agree. We have revised the text that existed in our original manuscript on lines 41-42 (which, in our original submission, read as: “However, the global prevalence of local groundwater level deepening is unknown because in-situ groundwater level data have not been synthesised at global scale.”). Specifically, we revised our manuscript to (1) replace ‘unknown’ with ‘poorly constrained’ (revision is underlined in revised text shown below), and (2) specify “pace” (instead of solely “prevalence”, as our original manuscript stated; this revision is underlined in revised text shown below). Our revised manuscript now states the following:

“However, the global pace and prevalence of local groundwater declines are poorly constrained because in-situ groundwater levels have not been synthesised at global scale.”

Line 46 – 47: I am not sure this statement is true. What do you mean by a disproportionate share of the world's regional aquifers. Your results show that at least 46% of aquifers have experienced slowing depletion, reversals, or continued groundwater level rise.

Thank you for your comment. Our original manuscript read as follows on line 46-47 “We also show that groundwater level deepening has accelerated over the past four decades in a disproportionate share of the world's regional aquifers”. Our revised manuscript includes specific statements detailing the multiple lines of evidence suggesting that the proportion of global aquifer systems exhibiting accelerating groundwater level declines is more prevalent as one would expect from random fluctuations alone (specifically, in the absence of systematic trends during the late-20th versus early-21st centuries). We thank the reviewer for motivating us to provide this clearer definition of what we mean by a disproportionate share; our revised manuscript states the following:

“These cases of accelerating groundwater level declines are over twice as prevalent as one would expect from random fluctuations in the absence of any systematic trends in either time period (12.5%; P -value < 0.001 by the binomial test). Furthermore, among all cases where groundwater levels declined in both the late-20th and early-21st centuries, declines in the early 21st century outpaced those in the late 20th century much more often than one would expect by chance (163 red points vs. 107 orange points in Fig. 3a; P -value < 0.001 by the sign test). If we exclude cases in which groundwater level trends changed by less than 0.1 m/year between these two periods (i.e., considering only points lying outside the grey diagonal band in Fig. 3a), we find that accelerating declines (red points) outnumber decelerating declines (orange points) by a ratio of 5:2 (P -value < 0.001 by the sign test). Thus, groundwater-level declines have accelerated in a disproportionate fraction of our aquifer systems.”

We thank the reviewer for motivating us to make these revisions to our manuscript to be clearer about our use of the term ‘disproportionate’. In addition to this text clarifying our use of the term ‘disproportionate’ (above), we also made considerable updates to our work in response to Reviewer 2's comment about the proportion of aquifers that have experienced slowing depletion, reversals, or continued groundwater level

rise. Specifically, our revised manuscript contains a section devoted solely to these cases (subsection entitled “*Slowing and reversing groundwater depletion*”, in which we state:

“Specifically, in half (49%) of the 542 aquifer systems in our analysis, groundwater level declines have decelerated (i.e., slowed; orange in Fig. 3; 20%) or reversed (blue in Fig. 3; 16%), or groundwater levels have continued to rise (purple in Fig. 3; 13%).”

We also now make clear mention of these cases of recovery in our summary paragraph (abstract), where we have added a statement that brings in Reviewer 2’s point about the importance of highlighting aquifers that exhibit slowing depletion, reversals, or continued groundwater level rises; the newly added sentence (at the conclusion of our summary paragraph) reads as follows:

“However, our data also reveal specific cases in which depletion trends have reversed following policy changes and surface-water diversions, demonstrating the potential for depleted aquifer systems to recover.”

Line 48 – 50: I think a key missing message here is the success in reversing depletion in at least 16% of aquifers and slowing depletion in another 18% of aquifers.

Thank you. We are grateful for these recommendations for us to revise our manuscript to better highlight the nuance in global groundwater level trends. We added a sentence to the abstract at the location the reviewer has specified (i.e., at the end of the abstract) stating:

“However, our data also reveal specific cases in which depletion trends have reversed in the early 21st century, demonstrating the potential for depleted aquifer systems to recover.”

Line 77: What period does your study cover? Is it up to 2022? It would be helpful to summarise the length of the time series used in the analysis, you could perhaps use a heatmap (or several heatmaps by region for example) to do this for all sites.

This was a helpful suggestion. Thank you. We created a heatmap to highlight the length of the time series analysed; this heatmap is displayed below:

“We analysed 21st century changes over time in depth to groundwater via three approaches, all of which involve non-parametric regression techniques. Specifically, we characterized changes over time in annual median depths to groundwater – for each monitoring well with adequate data for analyses – by calculating: (i) Spearman rank correlation coefficients; and, (ii) Mann-Kendall correlation coefficients.

We cannot straightforwardly compare a correlation coefficient from a non-parametric regression (e.g., Spearman ρ) against the Theil-Sen slopes presented in our main text. Instead, we report on the proportion of monitoring wells with consistent signs in trends determined via Theil-Sen versus nonparametric regression (e.g., the sign (positive versus negative) of Spearman ρ values and the sign (positive versus negative) of Theil-Sen slopes). We also present a plot comparing Mann-Kendall and Spearman rank correlation coefficients. These results are presented in the following pages.

Supplementary Table 2. Comparison of signs of Theil-Sen slopes (21st century trends in depth to groundwater) versus Spearman rank correlation coefficients. The percentages in the table refer to the proportion of monitoring wells falling within a given category (e.g., 61% of monitoring wells have positive Theil-Sen slopes and *also* positive Spearman ρ values)

	Negative Theil-Sen slope	Positive Theil-Sen slope
Positive Spearman ρ	1%	61%
Negative Spearman ρ	37%	2%

* we excluded 1% of monitoring wells with undefined or zero slopes (and/or correlation coefficients)

Supplementary Table 3. Comparison of signs of Theil-Sen slopes (21st century trends in depth to groundwater) versus Mann-Kendall correlation coefficients (τ). The percentages in the table refer to the proportion of monitoring wells falling within a given category (e.g., 62% of monitoring wells have positive Theil-Sen slopes and *also* positive Kendall τ values)

	Negative Theil-Sen slope	Positive Theil-Sen slope
Positive Kendall τ **	0.02%	62%
Negative Kendall τ **	38%	0.03%

* we excluded 1.8% of monitoring wells with undefined or zero slopes (and/or correlation coefficients)

** these values based on regression analyses without bootstrapping

Supplementary Fig. 8. Comparison of correlation coefficients based on Mann-Kendall and Spearman rank regressions. Each data point represents one monitoring well and its 21st century trend in depth to groundwater. Positive values indicate deepening groundwater levels over time, whereas negative values indicate shallowing of groundwater levels over time. Outputs of the two regression techniques agree in the majority of cases.

We also calculated Mann-Kendall correlation coefficients following a block bootstrapping approach (see figure below). We only completed this bootstrapping approach for stations with more than 15 unique calendar years during the 21st century with at least one water level measurement. In the overwhelming majority of cases, the non-bootstrapped Mann-Kendall correlation coefficient matched the bootstrapped Mann-Kendall correlation coefficient within 0.1 (99.88% of monitoring wells; see figure below). For further details on the bootstrapping approach see Önoz, B., & Bayazit M. Block bootstrap for Mann-Kendall trend test of serially dependent data. *Hydrological Processes* 26, 3552-3560 (2012).

Supplementary Fig. 9. Comparison of Mann-Kendall correlation coefficients without bootstrapping (x-axis values) versus with bootstrapping (y-axis values). Each data point represents one monitoring well and its 21st century variability in depth to groundwater (expressed as non-parametric correlation coefficients). Positive values indicate deepening groundwater levels over time, whereas negative values indicate shallowing of groundwater levels over time. Mann-Kendall correlation coefficients determined with versus without bootstrapping closely match one another in the vast majority of cases.

Hard to read in its present form. Can you improve the resolution of the main and inset maps?

Thank you for this suggestion. We have now uploaded a high resolution .eps file to the journal's manuscript tracking system, so that a figure of adequate resolution is available.

Line 146 – 155: Could you not look at increases in well numbers as a proxy for this? This type of data is certainly available from a significant number of areas to make it possible for a selection of basins (e.g., IGB in South Asia).

Thank you for this suggestion. We completed a substantial new geospatial analysis and detailed within a new Supplementary Note, which reads as follows:

“Some previous studies have examined statistical relationships between time series of well completion or permitting events (i.e., the number of wells in a given year) and trends in depth to groundwater (see, for example, Fig. 9 within Babaei, S., Mousavi, Z., Masoumi, Z., Malekshah, A.H., Roostaei, M., Aflaki, M. Land subsidence from interferometric SAR and groundwater patterns in the Qazvin plain, Iran. International Journal of Remote Sensing 41, 4780-4798 (2020)). Here we present time series of the number of recorded well completion events over time for a subset of aquifer systems where (i) aquifer-scale trends in depth to groundwater exceeding 0.5 m/year (i.e., deepening faster than 0.5 m/year), and (ii) a well completion event was recorded in at least 15 unique years during the 21st century in a database of global well completion events (database of well completion data described within the extensive Supplementary Materials of the following publication: Jasechko, S., Perrone, D. Global groundwater wells at risk of running dry. Science 372, 418-421. doi.org/10.1126/science.abc2755 (2021)).

Twenty-four (n=24) aquifer systems meet both of these criteria (i.e., “(i)” and “(ii)” described in the preceding paragraph). The following figures present time series of the total number of recorded well completion events for each of these aquifer systems.

We stress that the absolute number of wells should not be compared directly across different aquifer systems presented in the plots presented in this supplementary note because the underlying databases have differing degrees of completeness (for example, a survey we completed of well construction databases among US state agencies suggests wide variability in the number of actual wells that are recorded in state databases, see responses from groundwater well completion database managers to the inquiry “How representative are the wells?” presented in Table S1 within the Supplementary Information of Perrone, D., Jasechko, S. Deeper well drilling an unsustainable stopgap to groundwater depletion. *Nature Sustainability* 2, 773-782 (2019)). Therefore, we caution that changes in the recorded number of well completion events may be, to some degree, impacted by changes over time in the strategies and thoroughness of well construction data compilation (e.g., in Nebraska (US) domestic wells were exempt from inclusion in the state well completion database prior to the year 1993, see section S2.2 within the Supplementary Data of Perrone, D., Jasechko, S. Dry groundwater wells in the western United States. *Environmental Research Letters* 12, 104002 (2017)). Nevertheless, these cumulative 21st-century well completion time series (supplementary figures on the following pages) highlight that there was demand for the construction of new wells designed to extract groundwater throughout the early-21st century in all n=24 of the aquifer systems exhibiting deepening aquifer-scale trends in depth to groundwater.

Supplementary Fig. 291. 21st recorded century well completion events for the Upper Aconcagua Basin. Y-axis values are the cumulative number of recorded well completion events (during the 21st century). X-axis values represent calendar year. The title of the aquifer system is displayed at the top of the figure.

Supplementary Fig. 292. 21st recorded century well completion events for the Calera Aquifer. Y-axis values are the cumulative number of recorded well completion events (during the 21st century). X-axis values represent calendar year. The title of the aquifer system is displayed at the top of the figure.

Supplementary Fig. 293. 21st recorded century well completion events for the Chacabuco-Polpaico Aquifer. Y-axis values are the cumulative number of recorded well completion events (during the 21st century). X-axis values represent calendar year. The title of the aquifer system is displayed at the top of the figure.

Supplementary Fig. 294. 21st recorded century well completion events for the Chowchilla Basin (California Central Valley). Y-axis values are the cumulative number of recorded well completion events (during the 21st century). X-axis values represent calendar year. The title of the aquifer system is displayed at the top of the figure.

Supplementary Fig. 295. 21st recorded century well completion events for the Chupaderos Aquifer. Y-axis values are the cumulative number of recorded well completion events (during the 21st century). X-axis values represent calendar year. The title of the aquifer system is displayed at the top of the figure.

Supplementary Fig. 296. 21st recorded century well completion events for the Codegua and Graneros-Rancagua Aquifers. Y-axis values are the cumulative number of recorded well completion events (during the 21st century). X-axis values represent calendar year. The title of the aquifer system is displayed at the top of the figure.

Supplementary Fig. 297. 21st recorded century well completion events for the Confined Claiborne Near Jackson. Y-axis values are the cumulative number of recorded well completion events (during the 21st century). X-axis values represent calendar year. The title of the aquifer system is displayed at the top of the figure.

Supplementary Fig. 298. 21st recorded century well completion events for the Cuyama Valley. Y-axis values are the cumulative number of recorded well completion events (during the 21st century). X-axis values represent calendar year. The title of the aquifer system is displayed at the top of the figure.

Supplementary Fig. 299. 21st recorded century well completion events for the Gila Bend Basin. Y-axis values are the cumulative number of recorded well completion events (during the 21st century). X-axis values represent calendar year. The title of the aquifer system is displayed at the top of the figure.

Supplementary Fig. 300. 21st recorded century well completion events for the Kaweah Basin. Y-axis values are the cumulative number of recorded well completion events (during the 21st century). X-axis values represent calendar year. The title of the aquifer system is displayed at the top of the figure.

Supplementary Fig. 301. 21st recorded century well completion events for the Northern Kern Basin. Y-axis values are the cumulative number of recorded well completion events (during the 21st century). X-axis values represent calendar year. The title of the aquifer system is displayed at the top of the figure.

Supplementary Fig. 302. 21st recorded century well completion events for the Southern Kern Basin. Y-axis values are the cumulative number of recorded well completion events (during the 21st century). X-axis values represent calendar year. The title of the aquifer system is displayed at the top of the figure.

Supplementary Fig. 303. 21st recorded century well completion events for the Little Chino Valley. Y-axis values are the cumulative number of recorded well completion events (during the 21st century). X-axis values represent calendar year. The title of the aquifer system is displayed at the top of the figure.

Supplementary Fig. 304. 21st recorded century well completion events for the Madera Basin. Y-axis values are the cumulative number of recorded well completion events (during the 21st century). X-axis values represent calendar year. The title of the aquifer system is displayed at the top of the figure.

Supplementary Fig. 305. 21st recorded century well completion events for the Mill Creek Aquifer. Y-axis values are the cumulative number of recorded well completion events (during the 21st century). X-axis values represent calendar year. The title of the aquifer system is displayed at the top of the figure.

Supplementary Fig. 306. 21st recorded century well completion events for the Monterrey Valley. Y-axis values are the cumulative number of recorded well completion events (during the 21st century). X-axis values represent calendar year. The title of the aquifer system is displayed at the top of the figure.

Supplementary Fig. 307. 21st recorded century well completion events for the Parowan Valley. Y-axis values are the cumulative number of recorded well completion events (during the 21st century). X-axis values represent calendar year. The title of the aquifer system is displayed at the top of the figure.

Supplementary Fig. 308. 21st recorded century well completion events for the Popeto Aquifer. Y-axis values are the cumulative number of recorded well completion events (during the 21st century). X-axis values represent calendar year. The title of the aquifer system is displayed at the top of the figure.

Supplementary Fig. 309. 21st recorded century well completion events for the Salisbury Plain and Hampshire Chalk Aquifer. Y-axis values are the cumulative number of recorded well completion events (during the 21st century). X-axis values represent calendar year. The title of the aquifer system is displayed at the top of the figure.

Supplementary Fig. 310. 21st recorded century well completion events for the Central Santiago Basin. Y-axis values are the cumulative number of recorded well completion events (during the 21st century). X-axis values represent calendar year. The title of the aquifer system is displayed at the top of the figure.

Supplementary Fig. 311. 21st recorded century well completion events for the Northern Santiago Basin. Y-axis values are the cumulative number of recorded well completion events (during the 21st century). X-axis values represent calendar year. The title of the aquifer system is displayed at the top of the figure.

Supplementary Fig. 312. 21st recorded century well completion events for the Southern Santiago Basin. Y-axis values are the cumulative number of recorded well completion events (during the 21st century). X-axis values represent calendar year. The title of the aquifer system is displayed at the top of the figure.

Supplementary Fig. 313. 21st recorded century well completion events for the Tule Basin. Y-axis values are the cumulative number of recorded well completion events (during the 21st century). X-axis values represent calendar year. The title of the aquifer system is displayed at the top of the figure.

Supplementary Fig. 314. 21st recorded century well completion events for the Westside Basin. Y-axis values are the cumulative number of recorded well completion events (during the 21st century). X-axis values represent calendar year. The title of the aquifer system is displayed at the top of the figure

We reference this new supplementary section in the main text methods, where we state the following within our revised manuscript:

“Many of the aquifer systems exhibiting rapid groundwater level declines are being accessed by wells, as evidenced by recorded well completion events throughout the early 21st century (Supplementary Note 22).”

Line 167 – 169: What do you mean by long time spans here? Recent research shows that groundwater levels increased significantly in the Indus basin over a period of 60 – 70 years, this is within one lifetime. While the development of surface water irrigation played a large part in that it does demonstrate that groundwater accumulation can happen quickly given the right conditions.

Thank you for your comment. We that substantial groundwater can accumulate within a lifetime where conditions support such replenishment. We also agree that the Indus Basin is an excellent example of such accumulation. At the lines referenced by Reviewer 2, our original manuscript referred to long timespans without providing adequate context; we thank Reviewer 2 for highlighting that our original manuscript was unclear and vague, and we agree (the statement from our original manuscript read as follows: “Aquifer recharge is typically slow in drylands³⁰, and thus long timespans may be required for depleted dryland aquifers to recover³¹.”). Our revised manuscript now (a) places the time interval we refer to into context by juxtaposing dryland recharge rates (typically relatively low) against humid areas (typically with relatively high recharge rates), and (b) making specific mention of the Indus Basin case as an example of a dryland aquifer system where recharge rates have been increased via artificially enhanced recharge. Specifically, our revised manuscript reads as follows:

“Aquifer recharge is typically slow in drylands⁴¹, meaning that depleted dryland aquifers will generally take longer to recover than aquifers in wetter climates⁴², except where recharge rates are artificially increased (e.g., seepage from unlined canals in the Indus Basin³³).”

Line 227 – 229: You say deepening groundwater levels followed by shallowing groundwater levels is rare, however the preceding sentence suggests that as many as 16% of sites have experienced this. What is your definition of rare?

Thank you for this comment. We agree that our use of the term ‘rare’ in our original manuscript was without context and was therefore, at best, unclear. Our original manuscript read as follows on lines 227-229: “In 16% of our aquifer systems, groundwater levels became deeper during the late 20th century but rose in the early 21st century (the blue colours in Fig. 4). Such reversals of groundwater declines are rare but have occurred in some aquifer systems.” We have since revised this sentence by deleting the term ‘rare’ and instead simply presenting the percentage of aquifer systems exhibiting declining levels in the late-20th century; our revised manuscript now reads as follows:

“In 16% of our aquifer systems, groundwater level declines reversed—defined as cases where groundwater levels declined in the late 20th century but rose in the early 21st century (the blue colours in Fig. 3). For example, in the Bangkok Basin (Thailand), groundwater levels deepened during the late 20th century but shallowed in the early 21st century (see labelled blue point in Fig. 3a); this reversal has been attributed²⁵ to regulatory measures (e.g., groundwater pumping fees and licensing of wells).”

Line 212 – 249: So, 16% of sites experienced increased groundwater levels and 18% experienced slowing declines. That equates to 34% of sites where the situation is better now than it was at the end of the 20th century. This is larger overall than the percentage of sites where groundwater level decline has accelerated in the 21st century. This is a significant finding.

Thank you for this positive comment, and others you have provided. They helped us rework our manuscript to provide a more appropriately balanced discussion.

Overall, 46% of sites have experienced, slowing decline or rising groundwater level. This is surprising and I think is cause for optimism overall. It also illustrates that if you get management right, it can have significant positive impacts (although your point about the challenges associated with rising groundwater levels is well taken). I think you should stress this more in the manuscript.

Thank you. We made considerable updates to our manuscript. For example, we made five major updates to our manuscript to better highlight nuance in the groundwater level trends; these five revisions are specified above in this reply letter (following the Reviewer 2 comment that concludes with “Perhaps, if this nuance was recognised and discussed then the paper would be suitable for publication in Nature.”).

Fig 4d – where are these time-series from? It would be good to provide a location perhaps on your inset world map adjacent to this plot.

Thank you for your recommendation. We have updated the figure to include the locations of these monitoring wells on the inset world map adjacent to the plot. Here is the revised figure:

Fig 4e – i – these are not all that clear. It took me a while to work out some of the locations, could you add an outline to each landmass?

Thank you for suggesting we revise our figure. We (i) added an outline to each landmass, and (ii) added a heading at the top of each panel stating the general region depicted in the map (e.g., “central Chile” and “Australia”). The revised figure - with outlined landmasses and titles for each panel - now appears as:

Line 273 – 274: I do not necessarily take this message from Fig 4 or the preceding section of the manuscript, as discussed above. They are accelerating in 54% of aquifers, which is the majority, but not necessarily a disproportionate number of aquifer systems. In fact, this is a better picture than I might have imagined, and it is very interesting.

Thank you for this comment. Our original manuscript read as follows on line 273-274: “rates of groundwater decline are accelerating in a disproportionate share of aquifer systems (Fig. 4).” We have revised our manuscript to be clearer about the way we define the term ‘disproportionate’ and, critically (to Reviewer 2’s point), we have substantially revised our manuscript to provide a more balanced (nuanced) discussion of groundwater level declines. Specific to this location in the manuscript (i.e., original line 273-274), we have revised our manuscript to highlight the positive cases; the added text in our revised manuscript reads as follows:

“Our analysis also identifies cases where late-20th-century groundwater declines have been reversed in the early 21st century (blue points in Fig. 3).”

Line 276: Equally they may mask regions of groundwater accumulation (for example south-western Indian Punjab, which when aggregated at the scale of your analysis would not show up and is perhaps visible in your Supplementary Fig. 38).

This is a fair and good point. As part of our revisions to the text to balance our discussion of accelerated depletion by providing more (and deserved) content pertaining to groundwater recovery, we deleted the sentence that the Reviewer has marked in this comment (which, previously in our original manuscript, read as follows “Groundwater-level declines are likely to be even more widespread than our results suggest (Figs. 2-4), because they aggregate groundwater-level trends at aquifer-scale and thus may mask local hotspots of groundwater depletion”) from the manuscript.

Line 277 – 279: They also indicate significant management success ins some key areas. I think this also needs to be discussed.

Thank you. Your comments—including this one—have led to substantial revisions that, in our view, substantially improved our manuscript. Our original manuscript read as follows on lines 277-279 “Our results suggest that 21st century realities—including climatic trends, hydrogeologic conditions, groundwater withdrawal rates, land uses, and management approaches—have resulted in widespread, rapid, and accelerating groundwater-level declines.”. We have now revised our manuscript to also discuss the positive cases; specifically, we added the following statement at the location in our manuscript that Reviewer 2 has marked in this comment:

“Nevertheless, our compiled in-situ observations also capture numerous cases where declines in groundwater levels have been slowed, stopped or reversed following intervention (e.g., implementation of regulatory measures²⁵).”

Line 297 – 299: As does the evidence of groundwater level rises in at least 28% of aquifers in the 21st century. This is an important part of the story, and if we don’t learn from success (particularly those 16% of aquifers that have seen declines reversed) then what can we learn from?

This comment has helped us improve our conclusion to the manuscript; thank you. The line numbers identified by Reviewer 2 in this comment point to the following statement in our original manuscript (which we have now revised): “By documenting global hotspots of groundwater level declines, this analysis can inform efforts to address rapid and accelerating groundwater depletion.”. We revised this statement after reflecting on Reviewer 2’s comment (and their other comments earlier on); specifically, we now highlight cases of recovery in this closing sentence of the main text. Our revised manuscript reads as follows:

“Our analysis also documents cases where groundwater declines have slowed or reversed after (a) the implementation of groundwater policies, (b) the alleviation of groundwater demand via surface water transfers, or (c) the addition of groundwater storage following managed aquifer recharge projects. To address the growing problem of global groundwater depletion, these kinds of success stories would need to be replicated in dozens of aquifer systems with declining groundwater levels. Thus, our analysis illustrates the potential for depleted aquifers to recover, while demonstrating how much work remains to be done to protect groundwater resources. By documenting global hotspots of groundwater level decline and recovery, this analysis can inform efforts to address rapid and accelerating groundwater depletion.”

Line 312 – 313: Could you add a heatmap to the supplementary materials that illustrates the time series for each well, and highlights data gaps, for each country/continent (or whatever scale you think might be most appropriate)?

This was a helpful suggestion. Thank you. We added a new figure to our supplementary information detailing the proportion of monitoring wells spanning a given calendar year (among the ~170 thousand wells meeting our criteria for analyses of 21st century groundwater level trends). The new Supplementary Figure improved our manuscript as it provides better clarity about the differing time intervals that some of our databases span. The new figure appears as follows:

Journal of the American statistical Association 76, 817–823 (1981)), where non-linear trends are fit with yearly, weekly, and daily seasonality, plus holiday effects (Taylor, S.J., & Letham, B. *Forecasting at scale. The American Statistician* 72, 37–45 (2018)). For our purposes, we do not use Prophet to forecast time series but, rather, we use Prophet to determine the confidence interval of the model, which is then compared to the actual measurements in our time series. The main advantage of Prophet over other machine-learning tools for time series forecasting is that it automates the forecast performance evaluation when determining an optimal model. Prophet only requires the time series (e.g., measurement date and groundwater level) and the number of iteration steps to optimize the model. In our case, we used 300 Monte-Carlo steps and 1000 uncertainty iterations (for further details for these parameters, see the Prophet manual: https://facebook.github.io/prophet/docs/quick_start.html#python-api; accessed July 25, 2023). For each groundwater monitoring station, we first trained the machine-learning algorithm (i.e., Prophet) using the corresponding groundwater level (or groundwater elevation) time series, creating a prediction of the depth to groundwater time series (or groundwater elevation time series) over the same timespan. Supplementary Figs. 1 and 2 depict a groundwater level time series (dark blue colours in supplementary figures to follow) together with the model prediction (violet in supplementary figures to follow). Excluded groundwater level values (i.e., those removed from the time series before applying the machine learning) are depicted as green points in Supplementary Figs. 1 and 2.

In addition to outputting a model prediction (which is shown as a dark purple line in Supplementary Figs. 1 and 2) Prophet also provides lower and upper confidence bounds for each prediction. Specifically, we use the 99% confidence bounds to calculate a band around the model prediction that allows us to classify the groundwater measurement values. We defined outliers as well water level measurements that fall outside of the range specified as plus (or minus) 0.75 times the 99% confidence interval (offset from the machine learning prediction; that is, the predicted value from the machine learning algorithm (dark purple line) plus-or-minus 0.75 times the 99 percent confidence interval; see light purple shading in Supplementary Figs. 1 and 2). These points (which fall outside of the range specified by the prediction plus or minus 0.75 times the 99% confidence interval) are depicted as orange circles in Supplementary Figs. 1 and 2.

This approach (i.e., machine-learning-based outlier detection method) was applied to each monitoring well with more than 15 groundwater-level measurements, yielding a prediction for each time step and its 99% confidence interval; we defined points to be outliers and excluded them if they fell outside of the range defined by the predicted groundwater level ± 0.75 times this confidence interval (see figures on the following pages). Later in this Supplementary Note 2 we compare (A) the aquifer-scale trends in depth to groundwater presented in the main text (which are based on groundwater level time series after we excluded measurements with the application of the machine learning algorithm), versus (B) aquifer-scale trends in depth to groundwater calculated without applying the machine learning algorithm to detect and exclude potential outlier measurements; these two aquifer-scale trends in depth to groundwater (i.e., (A) and (B) described in this paragraph) are similar for most aquifer systems.

Supplementary Fig. 1. Schematic of a groundwater level time series and how outlier values were excluded (this figure differs from the one below as it plots a time series in units of groundwater elevation (not depth to groundwater)). The dark blue circles and line represent the groundwater level time series (note: points within this time series were generated by us for illustrative purposes). Green circles represent points that were excluded because their groundwater elevation value exceeded 8000m (label “>8000m”), was less than 1000m below sea level (see label “<-1000m”), or were exactly 999 or -9999 (see labels “999m” and “-9999m”). The thick purple line represents the predicted groundwater level time series (based on a machine learning algorithm: see Taylor, S.J., & Letham, B. *Forecasting at scale. The American Statistician* 72, 37–45 (2018)). The light purple shaded area represents an offset from the prediction (i.e., a constant offset along the y-axis away from the purple line) of a magnitude of the 99% confidence interval (CI) multiplied by a factor of 0.75 (i.e., we calculated the 99% confidence interval, multiplied by a factor of 0.75, and then plotted this offset above (top of light purple shading; see label “+0.75xCI(99)”, corresponding to an addition of 0.75 times the 99% confidence interval from the machine learning model prediction) and below (bottom of light purple shading; see label “-0.75xCI(99)”, corresponding to a subtraction of 0.75 times the 99% confidence interval from the machine learning model prediction). The red circles represent points that fall outside of the light purple shading (i.e., the measurement was flagged as an outlier by the machine learning algorithm and our threshold for outliers of 0.75 times the 99% confidence interval).

Supplementary Fig. 2. Schematic of a groundwater level time series and how outlier values were excluded (this figure differs from the one above as it plots a time series in units of depth to groundwater (not groundwater elevation)). The dark blue circles and line represent the groundwater level time series (note: points within this time series were generated by us for illustrative purposes). Green circles represent points that were excluded because their depth to groundwater values exceeded 1000m (label “>1000m”), or were exactly 999 or -9999 (see labels “999m” and “-9999m”). The thick purple line represents the predicted groundwater level time series (based on a machine learning algorithm: see Taylor, S.J., & Letham, B. *Forecasting at scale. The American Statistician* 72, 37–45 (2018)). The light purple shaded area represents an offset from the prediction (i.e., a constant offset along the y-axis away from the purple line) of a magnitude of the 99% confidence interval (CI) multiplied by a factor of 0.75 (i.e., we calculated the 99% confidence interval, multiplied by a factor of 0.75, and then plotted this offset above (top of light purple shading; see label “+0.75xCI(99)”, corresponding to an addition of 0.75 times the 99% confidence interval from the machine learning model prediction) and below (bottom of light purple shading; see label “-0.75xCI(99)”, corresponding to a subtraction of 0.75 times the 99% confidence interval from the machine learning model prediction). The red circle represents a point that falls outside of the light purple shading (i.e., the measurement was flagged as an outlier by the machine learning algorithm and our threshold for outliers of 0.75 times the 99% confidence interval).

(ii) Our revised work includes a new Supplementary Note 13 that provides a comparison of aquifer-scale trends in depth to groundwater with and without the machine-learning algorithm outlier detection algorithm reads as follows:

“The results presented in the main text are based on the exclusion of some well water level measurements defined to be outliers because they fall outside of the range defined by the machine-learning-model-predicted groundwater level ± 0.75 times its confidence interval (for schematic, see Supplementary Note 2). The figure below compares the aquifer-scale median trends in depth to groundwater presented in the main text (i.e., after excluding points defined to be outliers on the basis of the machine-learning-algorithm as described in the main text methods) versus aquifer-scale median

trends in depth to groundwater as determined without applying the machine-learning-algorithm (i.e., *including* points and time series that were excluded from our main text results on the basis of the machine learning algorithm).

The plot below demonstrates a very close correspondence of aquifer-scale trends in depth to groundwater after excluding values on the basis of a machine-learning-algorithm and its predicted groundwater level time series (x-axis values below) and aquifer-scale trends in depth to groundwater if no such machine-learning-based exclusion approach is applied. Specifically, all but 3 of our 1,693 aquifer systems (i.e., >99% of aquifer systems) have aquifer-scale trends in depth to groundwater that differ by less than 0.1 m/year between the following two scenarios: (i) after excluding points defined to be outliers on the basis of the machine-learning-algorithm as described in the main text methods, versus (ii) without applying the machine-learning-algorithm (i.e., *including* points and time series that were excluded from our main text results on the basis of the machine learning algorithm).”

Supplementary Fig. 39. Aquifer-scale trends in depth to groundwater (median Theil-Sen slopes) when we either (i) exclude points defined to be outliers on the basis of the machine-learning-algorithm as described in the main text methods (x-axis values), versus (ii) do not apply a machine-learning-algorithm to exclude points (i.e., *including* points and time series that were excluded from our main text results on the basis of the machine learning algorithm). Each circle represents one aquifer system. Points within the light grey area shaded area have x-axis and the y-axis values that differ by less than 0.1 m/year. 99.8% of aquifer-scale trends in depth to groundwater differ by less than 0.1 m/year under scenario (i) versus scenario (ii).

Line 369 – 372: This approach gives the reader more confidence in the robustness of your analysis. I would suggest mentioning in the main text. Did you consider using a bootstrapped Mann-Kendall analysis as well? I think it would be valuable.

Thank you for this good recommendation to consider non-parametric regression analyses. We agree with the reviewer that doing so only serves to benefit our work by further demonstrating robustness in our results.

We wrote a new Supplementary Note (Supplementary Note 4 entitled ‘Non-parametric regression approaches applied to groundwater level time series’) that includes not only Mann-Kendall results but also Spearman rank regressions. We compared the sign of these correlation coefficients to those from our Theil-Sen slopes presented in main text Fig. 1 and found that the overwhelming majority have identical signs (e.g., negative Kendall tau *and* negative Theil-Sen slope, or, positive Kendall tau *and* positive Theil-Sen slope). These new results improved our manuscript, and we thank the reviewer for this recommendation. We reference this new supplementary note in the main text of our manuscript to draw attention to this new analysis; specifically, we write the following in our revised manuscript:

“...analyses based on alternate regression techniques and on different quality control thresholds yield similar results; see Supplementary Notes 3, 4, 5 and 6”

We also reference this supplementary note specifically in the main text methods as follows:

“...for non-parametric regression techniques see Supplementary Note 4”

The new Supplementary Note 4 that contains the Mann-Kendall results reads as follows:

“We analysed 21st century changes over time in depth to groundwater via three approaches, all of which involve non-parametric regression techniques. Specifically, we characterized changes over time in annual median depths to groundwater – for each monitoring well with adequate data for analyses – by calculating: (i) Spearman rank correlation coefficients; and, (ii) Mann-Kendall correlation coefficients.

We cannot straightforwardly compare a correlation coefficient from a non-parametric regression (e.g., Spearman ρ) against the Theil-Sen slopes presented in our main text. Instead, we report on the proportion of monitoring wells with consistent signs in trends determined via Theil-Sen versus nonparametric regression (e.g., the sign (positive versus negative) of Spearman ρ values and the sign (positive versus negative) of Theil-Sen slopes). We also present a plot comparing Mann-Kendall and Spearman rank correlation coefficients. These results are presented in the following pages.

Supplementary Table 2. Comparison of signs of Theil-Sen slopes (21st century trends in depth to groundwater) versus Spearman rank correlation coefficients. The percentages in the table refer to the proportion of monitoring wells falling within a given category (e.g., 61% of monitoring wells have positive Theil-Sen slopes and **also** positive Spearman ρ values)

	Negative Theil-Sen slope	Positive Theil-Sen slope
Positive Spearman ρ	1%	61%
Negative Spearman ρ	37%	2%

** we excluded 1% of monitoring wells with undefined or zero slopes (and/or correlation coefficients)*

Supplementary Table 3. Comparison of signs of Theil-Sen slopes (21st century trends in depth to groundwater) versus Mann-Kendall correlation coefficients (tau). The percentages in the table refer to the proportion of monitoring wells falling within a given category (e.g., 62% of monitoring wells have positive Theil-Sen slopes and **also** positive Kendall τ values)

	Negative Theil-Sen slope	Positive Theil-Sen slope
Positive Kendall τ **	0.02%	62%
Negative Kendall τ **	38%	0.03%

** we excluded 1.8% of monitoring wells with undefined or zero slopes (and/or correlation coefficients)*

*** these values based on regression analyses without bootstrapping*

Supplementary Fig. 8. Comparison of correlation coefficients based on Mann-Kendall and Spearman rank regressions. Each data point represents one monitoring well and its 21st century trend in depth to groundwater. Positive values indicate deepening groundwater levels over time, whereas negative values indicate shallowing of groundwater levels over time. Outputs of the two regression techniques agree in the majority of cases.

We also calculated Mann-Kendall correlation coefficients following a block bootstrapping approach (see figure below). We only completed this bootstrapping approach for stations with more than 15 unique calendar years during the 21st century with at least one water level measurement. In the overwhelming majority of cases, the non-bootstrapped Mann-Kendall correlation coefficient matched the bootstrapped Mann-Kendall correlation coefficient within 0.1 (99.88% of monitoring wells; see figure below). For further details on the bootstrapping approach see Önöz, B., & Bayazit M. Block bootstrap for Mann-Kendall trend test of serially dependent data. *Hydrological Processes* 26, 3552-3560 (2012).

Supplementary Fig. 9. Comparison of Mann-Kendall correlation coefficients without bootstrapping (x-axis values) versus with bootstrapping (y-axis values). Each data point represents one monitoring well and its 21st century variability in depth to groundwater (expressed as non-parametric correlation coefficients). Positive values indicate deepening groundwater levels over time, whereas negative values indicate shallowing of groundwater levels over time. Mann-Kendall correlation coefficients determined with versus without bootstrapping closely match one another in the vast majority of cases.

Line 439 – 442 and Supplementary Note 11: The regional analyses conducted are very interesting. I think they also illustrate the same point as I have tried to make above, mainly that the overall depletion vs accumulation narrative is much more nuanced than your discussion would suggest. Take for example, Supplementary Fig. 25, 26, 29, 30, 31, to my eye there are almost as many areas where accumulation is occurring as there is areas where depletion is occurring. Supplementary Fig 27 is a notable exception, depletion certainly appears to be the dominant trend here. The plots in Supplementary Note 13 leave me with similar impressions.

Thank you for pointing this out. We agree, and we have made major updates to our manuscript (five of which are detailed in our reply to an earlier comment that concluded with “Perhaps, if this nuance was recognised and discussed then the paper would be suitable for publication in Nature.”).

We also added statements to the figure captions of the supplementary figures identified by the Reviewer in this specific comment. Each of these added statements draw attention to groundwater shallowing trends. In this way we have revised our supplementary information to provide discussion of both depletion but also accumulation. For example, our revised figure caption linked to a regional analysis for Europe now includes the following closing statement:

“Groundwater level shallowing (blue areas) are evident in several areas in central and south-central Spain, whereas groundwater level deepening is evident in some areas in southeastern Spain (red areas).”

As another example, we have added the following closing statement to the caption of another one of the supplementary figures that Reviewer 2 highlights in this particular comment:

“Groundwater level shallowing trends are found in several regions in southern China (blue areas near, for example, the Oujiang River delta) and southern Bangkok (Thailand).”

Line 459 – 462: Could you not test this hypothesis? It would potentially be useful to try and get a hold on what percentage of wells this might have applied to. A graphical summary of the time-series might help but there are possibly other ways you could assess this, perhaps looking at trends prior to data gaps or looking at absolute values (e.g. if groundwater is very shallow just prior to a data gap, might it not suggest a cause other than drying for the termination of the time series)?

This was an especially helpful recommendation. Thank you. Specifically, the recommendation to examine groundwater level time series motivated us to subset our monitoring well database to examine sites with well depth data. Next, we were able to map monitoring wells where the vertical offset between the well bottom and the deepest groundwater level was relatively small (implying that only a few meters of further decline could lead to well drying). Next, subset these wells based on groundwater level trends to identify sites where both of the following conditions are met:

We made two changes to our work in response to this comment:

First, we added a new sentence to our main text (at the location specified by the Reviewer in their comment) that reads as follows:

“We analysed monitoring well depths and depth to groundwater data for 72 thousand wells, and conclude that it is possible that a small proportion of our groundwater level time series were truncated due to well desiccation (see Supplementary Note 20: ‘Potential for monitoring wells to have run dry’).”

Second, we added a new supplementary section devoted to analysing the potential for monitoring wells to run dry. It reads as follows:

“It is possible that some monitoring well water level time series may be truncated because groundwater levels decline below the bottom of the well (see Hora, T., Srinivasan, V., & Basu, N.B. The groundwater recovery paradox in South India. Geophysical Research Letters 46, 9602–9611 (2019)). To explore the potential that monitoring wells ran dry during the 21st century we (i) developed a map of monitoring well depths (72,403 of the 169,178 monitoring wells in Fig. 1 have a recorded well depth value), (ii) calculated the deepest 21st century annual median groundwater level in each of these wells, and (iii) mapped the vertical offset between well bottoms and the deepest annual median groundwater level (i.e., [monitoring well depth] minus [deepest annual depth to groundwater]; see maps on the following pages).

Supplementary Fig. 287. Recorded total depths of monitoring wells. Each circle represents one monitoring well (subset from the 170 thousand monitoring wells displayed in main text Fig. 1). Blue points represent monitoring wells that are shallower than 25 m, whereas orange and red points represent monitoring wells completed to deeper depths. (a) Global map of monitoring well depths. (b-g) Regional maps of monitoring well depths.

Supplementary Fig. 288. Vertical offset from the bottom of monitoring wells to the deepest annual median groundwater level observed during the 21st century. Each circle represents one monitoring well (subset from the 170 thousand monitoring wells displayed in main text Fig. 1). Dark purple points represent monitoring wells where the deepest annual median groundwater level fell to within 5 m of the well bottom for at least one year during the 21st century; the deepest annual median groundwater level fell to within 5 m of the well bottom for at least one year during the 21st century in just 11% of monitoring wells (among the 72,403 monitoring wells with depth data depicted here). Lighter purple and yellow points represent monitoring wells where even the deepest annual median groundwater level did not come within 5 m of the well bottom during the 21st century (implying a smaller likelihood that the monitoring well went dry during the 21st century). (a) Global map of monitoring wells and their vertical offsets between well bottoms and the deepest annual median groundwater levels. (b-g) Regional maps of monitoring wells and their vertical offsets between well bottoms and the deepest annual median groundwater levels.

The annual median groundwater level fell to within 5 m of the well bottom for at least one year during the 21st century in just 11% of monitoring wells (7,995 wells among the 72,403 monitoring wells – with depth data – depicted in the map on the preceding page are shaded dark purple). Therefore, it is unlikely that the great majority of the monitoring wells presented in main text Fig. 1 were at risk of experiencing an

event where the groundwater fell below the well bottom (though we highlight that automated water level sensors reside above the bottom of the well, and thus we cannot rule out the possibility that these sensors ran dry for some of the monitoring wells we studied here). Furthermore, among these 7,995 wells, just 924 wells (i.e., 1.3% of the 72,403 monitoring wells displayed above) also have a 21st century trend in depth to groundwater exceeding 0.1 m/year (i.e., 924 wells have deepening groundwater levels and at least one year in the 21st century where the annual median groundwater level fell to within 5 m of the well bottom during the 21st century). One example of a monitoring well that likely ran dry is depicted below (see time series plot below).

Supplementary Fig. 289. Groundwater level time series for a monitoring well located ~15 km west of Levelland, Texas (USA; see inset maps in top right (global) and top right (regional) for the location of the well, marked by a white circle). Each circle represents one groundwater level measurement. The grey horizontal dashed line near the bottom of the plot represents the recorded bottom of the monitoring well. Groundwater levels were recorded to be 37 m below the land surface in the early 1960s. However, groundwater levels declined from 1961-2012 at a rate of ~0.3 m/year. The final measurement of depth to groundwater in the time series is 56.48 m and, and the total depth of the monitoring well is just marginally deeper (56.69 m); given the chronic groundwater level deepening observed over this 50-year time series, there is a high likelihood that this monitoring well ran dry following the year 2012.

The time series presented in the figure above exemplifies the potential for monitoring wells to run dry. However, we stress that cases such as the time series depicted above (where the monitoring well may have run dry) are rare in our dataset. We find that there are only 924 wells (out of 72,403 wells with depth data) where (a) the deepest annual median groundwater level fell to within 5 m of the well bottom for at least one year during the 21st century, and (b) the 21st century trend in depth to groundwater exceeded 0.5 m/year. Given that such cases are rare, it is unlikely that a substantial share of monitoring wells were perennially desiccated at a point during the 21st century. Thus, although we lack data describing the exact proportion of monitoring wells that may have run dry, the groundwater level time series and well depth data available to us do not immediately suggest that monitoring well desiccation events are so widespread such that they may substantially alter our main conclusions.

Nevertheless, monitoring well desiccation events will expectedly mean that our results will, if anything, tend to under-represent groundwater level deepening (because monitoring wells that run dry are likely to be disproportionately common in areas where groundwater levels are declining). Therefore, our main conclusions surrounding the prevalence of rapid and accelerating groundwater level declines are likely to be, if anything, conservative with respect to monitoring well desiccation events.

Line 466 – 467: the evidence from South Asia in the 20th century is that a period of substantial accumulation took place and that at the start of the 20th century groundwater levels were much deeper than they are now (despite the significant levels of groundwater depletion in the last 30 years or so). It would be good to be explicit about that here.

We agree. We revised on manuscript at the location identified by the reviewer; our original manuscript (which we have since revised) read as follows: “In some areas, substantial groundwater-level changes took place long before the four decades that we focus on here (e.g., there is evidence^{95,96} that substantial groundwater-level changes occurred during the mid-20th century in parts of South Asia).” Our revised manuscript now states, explicitly, that groundwater levels were much deeper than they are now in this part of South Asia. Specifically, our revised manuscript now reads as follows:

“For example, there is evidence^{104,105} that substantial accumulation occurred during the 20th century in parts of South Asia, and that groundwater levels were much deeper at the start of the 20th century than they are today (see, specifically, figure 3b within ref.¹⁰⁴).”

References

- Aeschbach-Hertig, Werner, and Tom Gleeson. "Regional strategies for the accelerating global problem of groundwater depletion." *Nature Geoscience* 5.12 (2012): 853-861.**
- Bierkens, Marc FP, and Yoshihide Wada. "Non-renewable groundwater use and groundwater depletion: a review." *Environmental Research Letters* 14.6 (2019): 063002.**
- Chen, J. L., Wilson, C. R., Tapley, B. D., Scanlon, B., & Güntner, A. (2016). Long-term groundwater storage change in Victoria, Australia from satellite gravity and in situ observations. *Global and Planetary change*, 139, 56-65.**
- Dangar, Swarup, Akarsh Asoka, and Vimal Mishra. "Causes and implications of groundwater depletion in India: A review." *Journal of Hydrology* 596 (2021): 126103.**
- Feng, W., Zhong, M., Lemoine, J. M., Biancale, R., Hsu, H. T., & Xia, J. (2013). Evaluation of groundwater depletion in North China using the Gravity Recovery and Climate Experiment (GRACE) data and ground-based measurements. *Water Resources Research*, 49(4), 2110-2118.**
- Konikow, L. F. (2015). Long-term groundwater depletion in the United States. *Groundwater*, 53(1), 2-9.**
- Konikow, L. F., & Kendy, E. (2005). Groundwater depletion: A global problem. *Hydrogeology Journal*, 13, 317-320.**
- Scanlon, B. R., Faunt, C. C., Longuevergne, L., Reedy, R. C., Alley, W. M., McGuire, V. L., & McMahon, P. B. (2012). Groundwater depletion and sustainability of irrigation in the US High Plains and Central Valley. *Proceedings of the national academy of sciences*, 109(24), 9320-9325.**
- Wada, Y., Van Beek, L. P., Van Kempen, C. M., Reckman, J. W., Vasak, S., & Bierkens, M. F. (2010). Global depletion of groundwater resources. *Geophysical research letters*, 37(20).**

Referee #3 (Remarks to the Author):

Review of Nature manuscript 2023-04-05873, “Rapid and accelerating groundwater level declines in global cultivated drylands”, Scott et al.

This manuscript details a compilation and analysis of in-situ measurements of groundwater-level trends in 178,000 globally distributed wells and provide new constraints on the prevalence of rapid and accelerating groundwater level declines and their correlation with land use and climatic drivers.

The work’s novelty lies in its impressive groundwater datasets ~1.5 million monitoring wells spanning more than 30 countries and calculated groundwater level trends for the remaining ~178,000 wells via robust regression of annual median groundwater levels (these Theil-Sen slopes, for each monitoring well). The dataset itself is of clear scientific merit and will be an asset to researchers and regional water resource managers, and the corresponding analysis and discussion implicate contextualization as vital for future groundwater sustainability and management.

My comments and minor points are outlined below. I recommend the manuscript may be published with some clarifications or corrections.

We thank Reviewer #3 for their helpful comments. Our manuscript improved considerably after making revisions and adding new analyses in response to each comment; the specific new analyses and revisions made to our work are detailed below.

Concerns :

1. Authors studied 207 aquifer systems/ trends in 1525 globally distributed aquifer systems and mention that the groundwater level trend show that the levels are becoming deeper over time but in many regions there are multiple aquifers and in most of the cases it has been found that the shallow aquifers are becoming dry in one season but yielding water in the rainy or after rainy season. Have the authors thought in this direction? (However, authors have mentioned in the limitations section about this, but still the location details are not clear and should be mentioned in the text)

Thank you for encouraging us to better highlight the importance of considering seasonal cyclicity. We agree entirely with Reviewer #3 that some shallow aquifers exhibit substantial seasonal variations, especially in climates with large intra-annual variability in precipitation (e.g., monsoonal climates). We also agree that our original manuscript was unclear about the importance of seasonality, and we have added a new supplementary section and text in an effort to make the importance of seasonality clearer in this reviewed manuscript.

Specifically, we have added a new supplementary section to our manuscript to better bring out the importance of seasonality and also to stress that the trends presented in the main text of our manuscript may differ if we were to examine time series of the deepest (or shallowest) annual groundwater levels. We specifically draw attention to a monitoring well in the Bengal Basin, where intra-annual groundwater level variability (i.e., the range of groundwater levels) has increased substantially over the past four decades, reflecting an increase in groundwater reliance but also the potential for dry-season drawdown to be partly recovered during the wet season due to local climate and land use conditions. The new supplementary text and figure read as follows:

“Our main results are based on annual median groundwater levels. However, we stress that some aquifers—particularly shallow aquifers that have the potential to receive considerable seasonal recharge—may have water level time series that exhibit extensive drawdown during one season, only to

be refilled (or partly refilled) during another season. To exemplify such a case, we present a groundwater level time series for a monitoring well located in the Bengal Basin (South Asia; figure below; data from Shamsudduha, M. et. al. *The Bengal Water Machine: Quantified freshwater capture in Bangladesh. Science* **377**, 1315-1319 (2022)). The groundwater level time series (figure below) displays substantial intra-annual variability, reflecting seasonal (i.e., dry season vs. wet season) differences in groundwater withdrawals and recharge. From 1980-1990, intra-annual groundwater variations averaged 4.6 m (i.e., as determined by the difference between the shallowest and the deepest groundwater level measured for any given calendar year). After the year 2000, intra-annual groundwater variations averaged 10.7 m. The larger intra-annual variations after the year 2000 suggest (i) that it is likely that annual groundwater extractions have increased in more recent years, and (ii) that groundwater recharge rates have also likely increased over time, as evidenced by the high-magnitude groundwater rises that occur each year, leading groundwater levels to approach levels observed prior to the year 2000. Further, groundwater level trend analyses for the deepest groundwater levels observed in a given calendar year yield a steeper slope (light pink circles; 0.44 m/year) than a trend based on the shallowest groundwater levels observed in a given calendar year (light blue circles; 0.17 m/year), further highlighting the increasing seasonal variability over time and how groundwater levels partly recover on a seasonal time step.”

Supplementary Fig. 290. Groundwater level time series for a monitoring well in the Bengal Basin. (a) Observed depth to groundwater levels for a monitoring well from 1980-2015. White circles represent individual measurements. Circles surrounded by a blue halo are the annual minimum depth to water (i.e., the shallowest groundwater level for a given calendar year). Circles surrounded by a red halo are the annual maximum depth to water (i.e., the deepest groundwater level for a given calendar year). Two lines represent the trends in depth to groundwater; the red line represents the trend in the annual maximum depth to water, and the blue line represents the trend in the annual minimum depth to water. Overall, intra-annual groundwater level variability has increased over time. (b) The location of the monitoring well (white circle)

in the Bengal Basin. (c) Inset map displaying the extent of the map in panel b (see map in lower right corner)."

2. In other case, the shallow aquifers are dried up and abandoned and the deeper aquifers are tapped for getting water. So in either case whether observations derived mostly from shallow aquifers are sufficiently representative, in addition to whether the length of time over which the observations are taken is sufficient, to draw robust conclusions based on this analysis. If most of the wells that authors used are shallow and those do not provide the clear picture of the recent groundwater pumping, I wonder if authors can provide separate analysis from the shallow and deep wells to show that there are considerable differences in the trends or not.

This was a very helpful suggestion; thank you. Specifically, this comment led to a new analysis and comparison of aquifer-scale trends based on either (i) wells shallower than 30 m, versus (ii) wells deeper than 100 m. Specifically, we identified 232 aquifer systems with at least 4 monitoring wells meeting our criteria for analyses that are shallower than 30 m, and also at least 4 monitoring wells meeting our criteria for analyses that are deeper than 100 m. We then compared (i) aquifer-scale trends in depth to groundwater based on shallow monitoring wells (i.e., median trend in depth to groundwater among all shallow monitoring wells), versus (ii) aquifer-scale trends in depth to groundwater based on deep monitoring wells. This analysis helped identify aquifer systems where groundwater levels are falling relatively rapidly in deep aquifers (compared to trends in shallower portions of these same aquifer systems; e.g., basins in the southern portion of California's Central Valley). In sum, this comment motivated us to complete a new analysis that better details differences between shallow and deep monitoring wells; the newly added supplementary information content reads as follows:

"To further compare groundwater level trends in shallow versus deep wells, we identified aquifer systems with both deep (>100 m) and shallow (<30 m) monitoring wells.

Specifically, we identified n=232 aquifer systems with at least 4 shallow (<30 m deep) monitoring wells and at least 4 deep (>100 m) monitoring wells with sufficient data to calculate a 21st century groundwater level trend (median Theil-Sen slope; see main text methods). For each of these n=232 aquifer systems, we calculated two aquifer-scale groundwater level trends: (i) an aquifer-scale groundwater level trend based solely on shallow (<30 m) monitoring wells (median Theil-Sen slope of all shallow monitoring wells), and (ii) an aquifer-scale groundwater level trend based solely on deep (>100 m) monitoring wells (median Theil-Sen slope of all shallow monitoring wells). We then compared these aquifer-scale groundwater level trends across all n=232 aquifer systems (figure below).

Among these n=232 aquifer systems, we show that aquifer-scale groundwater level trends based on deeper monitoring wells tend to have higher-magnitude values than trends based on shallow monitoring wells. Further, aquifer-scale groundwater level trends based on deep monitoring wells exceed trends based on shallow monitoring wells in a large number of aquifer systems (152 of 232 aquifer systems), implying that deepening groundwater level trends (or slower groundwater level shallowing trends) are more prevalent among deep monitoring wells compared to shallow monitoring wells (in these n=232 aquifer systems). Some of the aquifer systems where groundwater level deepening trends based on deep monitoring wells are outpacing deepening trends observed in shallow monitoring wells are in California's Central Valley (e.g., Tulare Lake Basin, Kaweah Basin, Kings Basin, Chowchilla Basin; see figure below). That groundwater levels are falling faster in deeper monitoring wells (relative to shallower monitoring wells) may imply a strengthening of downward-oriented hydraulic gradients in these areas, which can speed the movement of shallower groundwater to deeper aquifers, potentially impacting the quality of

these deep groundwaters if the shallow groundwater contains mobile contaminants.”

- Aquifer-scale trend based on shallow (<30 m) monitoring wells exceeds aquifer-scale trend based on deep (>100 m) wells by >0.1 m/year (n=19)
- Aquifer-scale trend based on shallow (<30 m) monitoring wells exceeds aquifer-scale trend based on deep (>100 m) wells by <0.1 m/year (n=58)
- Aquifer-scale trend based on deep (>100 m) monitoring wells exceeds aquifer-scale trend based on shallow (>100 m) wells by <0.1 m/year (n=89)
- Aquifer-scale trend based on deep (>100 m) monitoring wells exceeds aquifer-scale trend based on shallow (>100 m) wells by >0.1 m/year (n=66)

Supplementary Fig. 282. 21st-century aquifer-scale trends in depth to groundwater based on shallow versus deep monitoring wells. Each point represents one aquifer system. The x-axis values are the aquifer-scale groundwater level trend based solely on monitoring wells that have total depths of no more than 30 m (“shallow” monitoring wells; aquifer-scale trend determined by the median of all shallow monitoring wells with sufficient data to determine a 21st century groundwater level trend – see main text methods for details). The y-axis values represent aquifer-scale groundwater level trend based solely on monitoring wells that have total depths of greater than 100 m (“deep” monitoring wells; aquifer-scale trend determined by the median of all deep monitoring wells with sufficient data to determine a 21st century groundwater level trend). Larger-sized points represent aquifer systems where the aquifer-scale groundwater level trend based on shallow (<30 m) monitoring wells differs from the aquifer-scale groundwater level trend based on deep (>100 m) monitoring wells by more than 0.1 m/year; among these points (i.e., large purple points and large green points), aquifer-scale trends based on deep wells tend to be greater than aquifer-scale trends based on shallow monitoring wells (i.e., the large purple circles outnumber the large green circles by a ratio of about 3:1). This analysis emphasizes that groundwater level trends in deep and shallow monitoring wells can differ even when these wells are located in the same aquifer system. However, we also highlight that in most of the aquifer systems shown here, the aquifer-scale groundwater level trend based on shallow monitoring wells differs from the aquifer-scale

groundwater level trend based on deep monitoring wells by less than 0.1 m/year (147 of 232 aquifer systems)."

3. Authors have taken a good number of wells for their study but the local variations are too high in some regions, given these uncertainties in observations, it is not clear that how many wells truly represent the selected regions for the estimation of water level changes.

We thank Reviewer 3 for their comment. We agree that there are locations with high variability in some regions, both in terms of well density and also trends in well water levels. We have added a new sentence to our manuscript to highlight both the uncertainty in our results due to monitoring well density, and also the high variability in groundwater level trends (sometimes over short horizontal distances); specifically, we added the following new sentence to our main text 'Limitations' section:

"The high variability in monitoring well densities within aquifer systems, as well as the substantial variability in groundwater level trends even among co-located monitoring wells, are presented in a suite of maps for individual aquifer systems within Supplementary Notes 16 and 17."

The Reviewer's comment also motivated us to consider potential factors that may explain some of the cases where co-located monitoring wells exhibit differences in groundwater level trends. One of the ways this can arise is if nearby monitoring wells (i.e., multiple wells with similar x,y locations) have different depths (e.g., one monitoring well monitoring the shallow aquifer, another monitoring deeper aquifers). We have added a new statement in our manuscript to highlight the potential that monitoring well depths explain some of the spatial variability (x,y) in trends in depth to groundwater; the new statement reads as follows:

"Some of the variability in groundwater level trends among co-located wells may be partly explained by differences in the depths of nearby monitoring wells, as shallow and deep aquifers can have different groundwater level trends (see Supplementary Note 19)."

4. Although there is spatial variability in the wells of a region in 21st century, Is their any spatial correlation in wells for a selected region?

Thank you for this suggestion. We agree that providing information about not only the median Theil-Sen slope but also the prevalence of monitoring wells exhibiting declining (or shallowing) groundwater levels can add value to our work. We have completed a new geospatial analysis to explore clustering of groundwater-level trends among monitoring wells with the boundaries of individual aquifer systems. Specifically, we calculated the Moran's I value for all aquifer systems containing at least 30 monitoring wells (which is the minimum number of features recommended for use in such a geospatial analysis: <https://pro.arcgis.com/en/pro-app/latest/tool-reference/spatial-statistics/h-how-spatial-autocorrelation-moran-s-i-spatial-st.htm>). Our results highlight that many aquifers exhibit a tendency towards clustering among the groundwater-level trends of monitoring wells within the aquifer system's perimeter. However, we also showcase aquifer systems where we do not find strong evidence for clustering in groundwater level trends (Supplementary Note 23.2 presents maps for several of these aquifer systems). The new supplementary note reads as follows:

"We examined calculated Moran's I to characterize the degree to which similar groundwater-level trends cluster in 2D space (within the boundaries of individual aquifer systems). First, we identified aquifer systems with at least 30 monitoring wells, which is the recommended minimum number of features (see "Best practice guidelines" available at <https://pro.arcgis.com/en/pro-app/latest/tool-reference/spatial-statistics/h-how-spatial-autocorrelation-moran-s-i-spatial-st.htm> (accessed August 3, 2023)). Next, we calculated Moran's I values for each aquifer and their associated P-values (see table below).

Statistical variability in Moran's I values among the 594 aquifer systems that contain at least 30 monitoring wells meeting our criteria for analysis

	Moran's I < 0	Moran's I > 0
All aquifer systems with at least 30 wells (N=594 aquifer systems)	N=101 of N=594 (17% of aquifer systems)	N=493 of N=594 (83% of aquifer systems)
All aquifer systems with at least 30 wells and Moran's I P-value of less than 0.01 (N=336 aquifer systems)	N=2 of N=336 (0.6% of aquifer systems)	N=334 of N=336 (99.4% of aquifer systems)

Our calculations suggest that groundwater-level trends exhibit some degree of clustering in many of our aquifer systems (e.g., Moran's I P value of < 0.01 and Moran's I value of > 0 in 334 of 594 aquifer systems; see table above). In cases where the Moran's I P value is less than zero (i.e., 334 of 594 aquifer systems), nearly all (99.4%) Moran's I values are greater than zero (consistent with a tendency towards clustering, rather than dispersion, of groundwater-level trends).

23.1 EXAMPLES OF AQUIFER SYSTEMS EXHIBITING CLUSTERING OF TRENDS IN DEPTH TO GROUNDWATER

The following maps present a series of aquifer systems exhibiting some clustering of groundwater-level trends (i.e., aquifer systems with significant (i.e., P-value < 0.01) Moran's I values that exceed zero).

Supplementary Fig. 315. 21st century trends in depth to groundwater in the Central Cambay Basin (South Asia). The groundwater-level trends within this aquifer system (i.e., center of the map above) exhibit some degree of spatial clustering (e.g., red-shaded points tend to be co-located). The largest map

(on the left-hand side of the figure) depicts 21st century trends in depth to groundwater. Each point (coloured circle) represents a monitoring well; the colour of the point corresponds to the 21st century trend in depth to groundwater for the monitoring well (Theil-Sen slope). Orange and red points represent monitoring wells where groundwater levels became deeper over time (Theil-Sen slope >0.25 m/year). Blue points represent monitoring wells where groundwater levels became shallower over time (Theil-Sen slope <0.25 m/year; see legend in upper-right corner of figure). The boundaries of the aquifer system are outlined in black. The map in the lower-right corner of the figure presents spatial variations in near-surface geology (data from Hartmann, J., & Moosdorf, N. (2012). The new global lithological map database GLiM: A representation of rock properties at the Earth surface. *Geochemistry, Geophysics, Geosystems* 13, doi.org/10.1029/2012GC004370).

Supplementary Fig. 316. 21st century trends in depth to groundwater in the Kaweah Basin (western US, within the broader Tulare Basin that is, itself, within the broader California Central Valley aquifer system). The groundwater-level trends within this aquifer system (i.e., center of the map above) exhibit some degree of spatial clustering (e.g., red-shaded points tend to be co-located). The largest map (on the left-hand side of the figure) depicts 21st century trends in depth to groundwater. Each point (coloured circle) represents a monitoring well; the colour of the point corresponds to the 21st century trend in depth to groundwater for the monitoring well (Theil-Sen slope). Orange and red points represent monitoring wells where groundwater levels became deeper over time (Theil-Sen slope >0.25 m/year). Blue points represent monitoring wells where groundwater levels became shallower over time (Theil-Sen slope <0.25 m/year; see legend in upper-right corner of figure). The boundaries of the aquifer system are outlined in black. The map in the lower-right corner of the figure presents spatial variations in near-surface geology (data from Hartmann, J., & Moosdorf, N. (2012). The new global lithological map database GLiM: A representation of rock properties at the Earth surface. *Geochemistry, Geophysics, Geosystems* 13, doi.org/10.1029/2012GC004370).

Supplementary Fig. 317. 21st century trends in depth to groundwater in the Northern Eromanga Basin (Australia; within the broader Great Artesian Basin). The groundwater-level trends within this aquifer system (i.e., center of the map above) exhibit some degree of spatial clustering (e.g., red-shaded points tend to be co-located). The largest map (on the left-hand side of the figure) depicts 21st century trends in depth to groundwater. Each point (coloured circle) represents a monitoring well; the colour of the point corresponds to the 21st century trend in depth to groundwater for the monitoring well (Theil-Sen slope). Orange and red points represent monitoring wells where groundwater levels became deeper over time (Theil-Sen slope >0.25 m/year). Blue points represent monitoring wells where groundwater levels became shallower over time (Theil-Sen slope <0.25 m/year; see legend in upper-right corner of figure). The boundaries of the aquifer system are outlined in black. The map in the lower-right corner of the figure presents spatial variations in near-surface geology (data from Hartmann, J., & Moosdorf, N. (2012). The new global lithological map database GLiM: A representation of rock properties at the Earth surface. *Geochemistry, Geophysics, Geosystems* 13, doi.org/10.1029/2012GC004370).

Supplementary Fig. 318. 21st century trends in depth to groundwater in the Shush Plain (Iran; within the broader Dezful-Andimeshk Plain). The groundwater-level trends within this aquifer system (i.e., center of the map above) exhibit some degree of spatial clustering (e.g., red-shaded points tend to be co-located). The largest map (on the left-hand side of the figure) depicts 21st century trends in depth to groundwater. Each point (coloured circle) represents a monitoring well; the colour of the point corresponds to the 21st century trend in depth to groundwater for the monitoring well (Theil-Sen slope). Orange and red points represent monitoring wells where groundwater levels became deeper over time (Theil-Sen slope >0.25 m/year). Blue points represent monitoring wells where groundwater levels became shallower over time (Theil-Sen slope <0.25 m/year; see legend in upper-right corner of figure). The boundaries of the aquifer system are outlined in black. The map in the lower-right corner of the figure presents spatial variations in near-surface geology (data from Hartmann, J., & Moosdorf, N. (2012). The new global lithological map database GLiM: A representation of rock properties at the Earth surface. *Geochemistry, Geophysics, Geosystems* 13, doi.org/10.1029/2012GC004370).

23.2 EXAMPLES OF AQUIFER SYSTEMS THAT DO NOT EXHIBIT CLUSTERING OF TRENDS IN DEPTH TO GROUNDWATER

The following maps present a series of aquifer systems where the Mohan's I values do not provide strong and significant evidence for clustering in trends in depth to groundwater (i.e., P -values > 0.5).

Supplementary Fig. 319. 21st century trends in depth to groundwater in the Northern Duero Basin (Spain). The groundwater-level trends within this aquifer system (i.e., center of the map above) exhibit some degree of spatial clustering (e.g., red-shaded points tend to be co-located). The largest map (on the left-hand side of the figure) depicts 21st century trends in depth to groundwater. Each point (coloured circle) represents a monitoring well; the colour of the point corresponds to the 21st century trend in depth to groundwater for the monitoring well (Theil-Sen slope). Orange and red points represent monitoring wells where groundwater levels became deeper over time (Theil-Sen slope >0.25 m/year). Blue points represent monitoring wells where groundwater levels became shallower over time (Theil-Sen slope <0.25 m/year; see legend in upper-right corner of figure). The boundaries of the aquifer system are outlined in black. The map in the lower-right corner of the figure presents spatial variations in near-surface geology (data from Hartmann, J., & Moosdorf, N. (2012). The new global lithological map database GLiM: A representation of rock properties at the Earth surface. *Geochemistry, Geophysics, Geosystems* 13, doi.org/10.1029/2012GC004370).

Supplementary Fig. 320. 21st century trends in depth to groundwater in the Northern Piedmont Plain (North China Plain). The groundwater-level trends within this aquifer system (i.e., center of the map above) exhibit some degree of spatial clustering (e.g., red-shaded points tend to be co-located). The largest map (on the left-hand side of the figure) depicts 21st century trends in depth to groundwater. Each point (coloured circle) represents a monitoring well; the colour of the point corresponds to the 21st century trend in depth to groundwater for the monitoring well (Theil-Sen slope). Orange and red points represent monitoring wells where groundwater levels became deeper over time (Theil-Sen slope >0.25 m/year). Blue points represent monitoring wells where groundwater levels became shallower over time (Theil-Sen slope <0.25 m/year; see legend in upper-right corner of figure). The boundaries of the aquifer system are outlined in black. The map in the lower-right corner of the figure presents spatial variations in near-surface geology (data from Hartmann, J., & Moosdorf, N. (2012). The new global lithological map database GLiM: A representation of rock properties at the Earth surface. *Geochemistry, Geophysics, Geosystems* 13, doi.org/10.1029/2012GC004370).

Supplementary Fig. 321. 21st century trends in depth to groundwater in the Osaka Plain (Japan). The groundwater-level trends within this aquifer system (i.e., center of the map above) exhibit some degree of spatial clustering (e.g., red-shaded points tend to be co-located). The largest map (on the left-hand side of the figure) depicts 21st century trends in depth to groundwater. Each point (coloured circle) represents a monitoring well; the colour of the point corresponds to the 21st century trend in depth to groundwater for the monitoring well (Theil-Sen slope). Orange and red points represent monitoring wells where groundwater levels became deeper over time (Theil-Sen slope >0.25 m/year). Blue points represent monitoring wells where groundwater levels became shallower over time (Theil-Sen slope <0.25 m/year; see legend in upper-right corner of figure). The boundaries of the aquifer system are outlined in black. The map in the lower-right corner of the figure presents spatial variations in near-surface geology (data from Hartmann, J., & Moosdorf, N. (2012). The new global lithological map database GLiM: A representation of rock properties at the Earth surface. *Geochemistry, Geophysics, Geosystems* 13, doi.org/10.1029/2012GC004370).

5. Authors mention that even management strategies are insufficient for check the decline in groundwater levels, authors should provide a detailed examination in that. It has been found that in some of the aquifer systems the management strategies are fruitful in augmenting groundwater levels.

We thank Reviewer 3 for their comment. Their comment, along with a similar suggestion from Reviewer 2, motivated us to make major changes to our manuscript to better highlight the much more nuanced picture than the discussion section of our original manuscript suggested. Here we detail five examples of substantial changes (i.e., (i) to (v) below) we have made to our work to help it to better recognize nuance in our results:

(i) We added new text to our summary paragraph (abstract) highlighting the nuanced picture elucidated by the global piezometric data, which the reviewer highlights. Specifically, we added the following concluding sentence to our abstract, to emphasize Reviewer 2’s point about nuance in a prominent way:

“However, our data also reveal specific cases in which depletion trends have reversed following policy changes and surface-water diversions, demonstrating the potential for depleted aquifer systems to recover.”

(ii) We reorganized our manuscript by presenting our comparison of groundwater level trends in the late-20th versus early-21st centuries earlier in the manuscript (specifically, we moved this section up such that it comes before our discussion of climate and cultivated land areas). We added a specific heading to our revised manuscript to emphasize the Reviewer’s comment; this new subheading reads as follows:

“Slowing and reversing groundwater-level declines”

(iii) We have added considerable new discussion surrounding nuance to our manuscript, motivated by Reviewer 2’s comment. For example, we have added the following new sentences to our main text:

“Specifically, in half (49%) of the 542 aquifer systems in our analysis, groundwater level declines have decelerated (i.e., slowed; orange in Fig. 3; 20%) or reversed (blue in Fig. 3; 16%), or groundwater levels have continued to rise (purple in Fig. 3; 13%).”

(iv) We have completed an even more thorough review of available literature on groundwater level shallowing, adding to our compilation of groundwater shallowing events (presented in the Supplementary Information (Supplementary Note 15.2). Our revised compilation of groundwater shallowing trends (as presented in the Supplement) is shown below (and these data are also presented in main text Fig. 2 for locations where we lack groundwater level data):

Supplementary Fig. 42. Locations where groundwater level shallowing has been observed in one or more monitoring wells. Each circle represents the location of a local- to regional-scale study area where one or more published works have recorded groundwater level shallowing in one or more monitoring wells (for details for each study area, see table on the following pages). The colour of each circle corresponds to the average of the minimum and maximum literature values (see table on the following pages). Groundwater level shallowing has been captured in study areas beyond the countries for which we have monitoring well data (i.e., groundwater level shallowing in some countries that are not part of the analyses presented in the main text figures), including areas in Burkina Faso and Niger.

(v) We have substantially revised the concluding section of our manuscript, after considering Reviewer 2’s comment. Specifically, we have revised the title of the subsection to *“Depleting and recovering groundwater resources”* (instead of our original manuscript’s heading, which read as: *“Depleting*

groundwater resources”). Further, we have added the following content to better highlight cases of groundwater recovery (and cases where rates of groundwater level decline have slowed):

“Our analysis also documents cases where groundwater declines have slowed or reversed after (a) the implementation of groundwater policies, (b) the alleviation of groundwater demand via surface water transfers, or (c) the addition of groundwater storage following managed aquifer recharge projects. To address the growing problem of global groundwater depletion, these kinds of success stories would need to be replicated in dozens of aquifer systems with declining groundwater levels. Thus, our analysis illustrates the potential for depleted aquifers to recover, while demonstrating how much work remains to be done to protect groundwater resources. By documenting global hotspots of groundwater level decline and recovery, this analysis can inform efforts to address rapid and accelerating groundwater depletion.”

6. Authors have mentioned about the groundwater level trends but not provided any details about the precipitation variability (interannual, decadal and that from climate change)

This was an exceptionally helpful recommendation. Thank you.

Specifically, Reviewer 3’s comment motivated us to complete a substantial new analysis of precipitation variability. Specifically, we analysed daily gridded precipitation data from two different climate data products (TerraClimate and CHELSA); we then calculated annual precipitation rates for all of our study aquifers for every year from 1980-present. Next, we examined how different categories of groundwater level trends (i.e., the five colours in main text Fig. 3 within the revised manuscript (previously main text Fig. 4 in our original manuscript) compare with different precipitation rates over the late-20th versus early-21st centuries. We document this new analysis in a lengthy new Supplementary Note 10, which reads as follows:

“Groundwater quantity can be influenced by climate variability via direct impacts (e.g., changes in recharge rates) or via indirect impacts (e.g., changes in groundwater demands; Taylor, R.G. et al. (2013). Ground water and climate change. Nature Climate Change 3, 322-329). We analyse precipitation time series for our 1,693 aquifer systems. Specifically, we analysed gridded precipitation data from:

- (A) ‘TerraClimate’ (Abatzoglou, J.T., Dobrowski, S.Z., Parks, S.A., & Hegewisch, K.C. TerraClimate, a high-resolution global dataset of monthly climate and climatic water balance from 1958–2015. Scientific Data 5 (2018) data downloaded July 21, 2023 via <https://www.climatologylab.org/wget-terraclimate.html>); and,*
- (B) ‘CHELSA’ (Karger, D.N., Wilson, A.M., Mahony, C., Zimmermann, N.E., Jetz, W. (2021): Global daily 1km land surface precipitation based on cloud cover-informed downscaling. Scientific Data. doi.org/10.1038/s41597-021-01084-6. see: <https://chelsa-climate.org>)*

We calculated annual precipitation rates for each of our aquifers by aggregating all grid cells within the boundaries of each aquifer system, and summing up values on an annual time step (to calculate an aquifer-scale annual precipitation rate for each calendar year for each aquifer system; units of mm/year). Next, we calculated a four-decade mean precipitation rate for each aquifer system (see maps to follow; ‘CHELSA’ four-decade average precipitation rates are based on calendar years 1980-2018, whereas ‘TerraClimate’ data are based on calendar years 1980-2022).

Supplementary Fig. 23. Annual precipitation (mm/year) for each of our study aquifers over the years 1980-2022 (data from 'TerraClimate'; Abatzoglou, J.T., Dobrowski, S.Z., Parks, S.A., & Hegewisch, K.C. TerraClimate, a high-resolution global dataset of monthly climate and climatic water balance from 1958–2015. *Scientific Data* 5 (2018). data downloaded July 21, 2023 via <https://www.climatologylab.org/wget-terraclimate.html>). Annual average precipitation rates vary substantially among the 1,693 aquifer systems that we study.

Supplementary Fig. 24. Annual precipitation (mm/year) for each of our study aquifers over the years 1980-2022 (data from 'CHELSA'; Karger, D.N., Wilson, A.M., Mahony, C., Zimmermann, N.E., Jetz, W. (2021): Global daily 1km land surface precipitation based on cloud cover-informed downscaling. *Scientific Data*. doi.org/10.1038/s41597-021-01084-6. see: <https://chelsa-climate.org>). Annual average precipitation rates vary substantially among the 1,693 aquifer systems that we study.

Supplementary Fig. 25. Comparison of aquifer-scale average annual precipitation (mm/year over the past 4 decades) as reported by (i) the ‘TerraClimate’ dataset, versus (ii) the ‘CHELSA’ dataset. (‘CHELSA’ dataset: Karger, D.N., Wilson, A.M., Mahony, C., Zimmermann, N.E., Jetz, W. (2021): Global daily 1km land surface precipitation based on cloud cover-informed downscaling. *Scientific Data*. doi.org/10.1038/s41597-021-01084-6. see: <https://chelsa-climate.org>; ‘TerraClimate’ dataset: Abatzoglou, J.T., Dobrowski, S.Z., Parks, S.A., & Hegewisch, K.C. TerraClimate, a high-resolution global dataset of monthly climate and climatic water balance from 1958–2015. *Scientific Data* 5 (2018). data downloaded July 21, 2023 via <https://www.climatologylab.org/wget-terraclimate.html>). Precipitation rates from the two data products are similar for most aquifer-systems.

We also calculated average annual precipitation rates over the late-20th century and over the early-21st-century. We then compared these two ~20-year precipitation records by dividing the early-21st-century mean annual precipitation rate by the late-20th-century mean annual precipitation rate. The results are presented in the maps to follow (the early-21st-century time interval, over which we calculated an average annual precipitation rate, is 2000-2018 for the ‘CHELSA’ dataset and 2000-2022 for the ‘TerraClimate’ dataset; the late-20th-century time interval, over which we calculated an average annual precipitation rate, is 1980-2000 for both ‘TerraClimate’ and ‘CHELSA’).

Our comparison of late-20th-century and early-21st-century average annual precipitation rates demonstrates that both the TerraClimate and CHELSA datasets suggest:

- Early-21st-century precipitation rates were lower than late-20th-century precipitation rates for many aquifer systems in (a) the western US, (b) eastern Iran, (c) central Saudi Arabia, (d) central Chile, and (e) southeastern Australia;

- Early-21st-century precipitation rates were higher than late-20th-century precipitation rates in (a) the northeastern US, (b) western Taiwan, (c) southeastern India, and (d) some parts of northern Australia.

Supplementary Fig. 26. Comparison of aquifer-scale average annual precipitation during the early-21st-century (2000-2022) versus the late-20th-century (1980-2000) as reported by the ‘TerraClimate’ dataset (‘TerraClimate’ dataset: Abatzoglou, J.T., Dobrowski, S.Z., Parks, S.A., & Hegewisch, K.C. TerraClimate, a high-resolution global dataset of monthly climate and climatic water balance from 1958–2015. *Scientific Data* 5 (2018). data downloaded July 21, 2023 via <https://www.climatologylab.org/wget-terraclimate.html>). Aquifer systems where average annual precipitation during the early-21st-century was lower than average annual precipitation during the late-20th-century are shaded orange or red. Aquifer systems where average annual precipitation during the early-21st-century was higher than average annual precipitation during the late-20th-century are shaded blue. ‘TerraClimate’ data suggests that early-21st-century annual precipitation was lower than late-20th-century annual precipitation in much of the western US, Iran, central Chile and east-central Australia.

Supplementary Fig. 27. Comparison of aquifer-scale average annual precipitation during the early-21st-century (2000-2022) versus the late-20th-century (1980-2000) as reported by the 'CHELSA' dataset ('CHELSA' dataset: Karger, D.N., Wilson, A.M., Mahony, C., Zimmermann, N.E., Jetz, W. (2021): Global daily 1km land surface precipitation based on cloud cover-informed downscaling. *Scientific Data*. doi.org/10.1038/s41597-021-01084-6. see: <https://chelsa-climate.org>). Aquifer systems where average annual precipitation during the early-21st-century was lower than average annual precipitation during the late-20th-century are shaded orange or red. Aquifer systems where average annual precipitation during the early-21st-century was higher than average annual precipitation during the late-20th-century are shaded blue. 'CHELSA' data suggests that early-21st-century annual precipitation was lower than late-20th-century annual precipitation in much of the western US, Iran, central Chile and east-central Australia.

To better understand potential spatiotemporal statistical relationships between precipitation variability, we plotted the 542 aquifer systems for which we had adequate data to meet our criteria for juxtaposition of late-20th versus early-21st century aquifer-scales in depth to groundwater (i.e., the aquifer systems represented in main text Fig. 3). Specifically, we compare average annual precipitation rates for the early-21st and late-20th centuries (i.e., early-21st-century average annual precipitation divided by late-20th-century average annual precipitation; values on x-axes of supplementary figures to follow) and 21st-century groundwater level trends (aquifer-scale trends calculated as the median of all monitoring well water level trends within the boundaries of an aquifer system; y-axis values in plots to follow). Each plot represents one of the categories presented in the main text (Fig. 3); specifically, these categories are: (i)

groundwater levels became shallower during 1980-2000, and continued to become shallower (purple points), (ii) groundwater levels became shallower during 1980-2000, but have since become deeper (yellow points), (iii) groundwater levels became deeper during 1980-2000, but have since become shallower (blue points), (iv) groundwater levels became deeper during 1980-2000, and continued to become deeper but at a slower rate (i.e., decelerated deepening; orange points), and, (v) groundwater levels became deeper during 1980-2000, and continued to become deeper at a faster rate (i.e., accelerated deepening; red points).

Our analysis demonstrates that: (a) the majority (83-90%) of aquifer systems categorized as exhibiting ‘accelerated deepening’ (i.e., groundwater levels became deeper during 1980-2000, and continued to become deeper at a faster rate) also received less precipitation in the early-21st century than in the late-20th century, and (b) a slight majority (56-62%) of aquifer systems categorized as exhibiting ‘recovering’ groundwater levels (i.e., groundwater levels became deeper during 1980-2000, but have since become shallower) received more precipitation in the early-21st century than in the late-20th century (see table below and supplementary figures to follow).

Average annual precipitation in the early-21st versus the late-20th centuries for aquifer systems with adequate data to assess groundwater level trends over the past 4 decades (i.e., those depicted in main text Fig. 3)

Category*	Number of aquifer systems	Percent of aquifer systems where there was *less* precipitation in the early-21st century than in the late-20th century	Percent of aquifer systems where there was *more* precipitation in the early-21st century than in the late-20th century
groundwater levels became shallower during 1980-2000, but have since become deeper (yellow points),	117	79% - 85%	15% - 21%
groundwater levels became deeper during 1980-2000, and continued to become deeper at a faster rate (i.e., accelerated deepening ; red points)	163	83% - 90%	10% - 17%
groundwater levels became deeper during 1980-2000, and continued to become deeper but at a slower rate (i.e., decelerated deepening ; orange points)	107	70% - 79%	21% - 30%
groundwater levels became deeper during 1980-2000, but have since become shallower (blue points)	86	38% - 44%	56% - 62%
groundwater levels became shallower during 1980-2000, and continued to become shallower (purple points)	69	55% - 58%	42% - 45%

* each row corresponds to one of the categories of aquifer-scale groundwater level trends depicted in main text Fig. 3 and in the supplementary figures on the pages to follow. The range of percentages correspond to range of results obtained from two different precipitation time series datasets: (a) ‘CHELSA’ dataset: Karger, D.N., Wilson, A.M., Mahony, C., Zimmermann, N.E., Jetz, W. (2021): Global daily 1km

land surface precipitation based on cloud cover-informed downscaling. Scientific Data. doi.org/10.1038/s41597-021-01084-6. see: <https://chelsa-climate.org>; and (b) ‘TerraClimate’ dataset: Abatzoglou, J.T., Dobrowski, S.Z., Parks, S.A., & Hegewisch, K.C. TerraClimate, a high-resolution global dataset of monthly climate and climatic water balance from 1958–2015. Scientific Data 5 (2018). data downloaded July 21, 2023 via <https://www.climatologylab.org/wget-terraclimate.html>).

Supplementary Fig. 28. Comparison of trends in depth to groundwater and annual precipitation rates for ‘red points’. (a) Comparison of trends in depth to groundwater during two different time intervals (early-21st-century and late-20th-century). This panel presents identical data and categories as that displayed in main text Fig. 3 (see that figure caption for details). (b) An example of a time series in depth to groundwater exhibiting accelerated groundwater level declines (i.e., red points in panel a). (c-d) Comparison of average annual precipitation rates for the early-21st and late-20th centuries (i.e., early-21st-century average annual precipitation divided by late-20th-century average annual precipitation; values on x-axes) and 21st-century groundwater level trends (aquifer-scale trends calculated as the median of all monitoring well water level trends within the boundaries of an aquifer system; y-axis values). Each point represents one of our 542 aquifer systems. X-axis values of less than 1 indicate that average annual precipitation during the early-21st-century was lower than average annual precipitation during the late-20th-century. Panels c and d only display aquifer systems categorized as exhibiting accelerated groundwater level declines (i.e., only the red points in panel a are displayed in panels c and d). The x-axis values in panel c are based on the ‘TerraClimate’ dataset, whereas the x-axis values in panel d are based on the

'CHELSA' dataset (see text above for full references to these two precipitation data sources). Our analysis of precipitation time series demonstrates that the majority (83-90%) of aquifer systems categorized as exhibiting 'accelerated deepening' (i.e., groundwater levels became deeper during 1980-2000, and continued to become deeper at a faster rate) also received less precipitation in the early-21st-century than in the late-20th-century.

Supplementary Fig. 29. Comparison of trends in depth to groundwater and annual precipitation rates for 'yellow points'. (a) Comparison of trends in depth to groundwater during two different time intervals (early-21st-century and late-20th-century). This panel presents identical data and categories as that displayed in main text Fig. 3 (see that figure caption for details). (b) An example of a time series in depth to groundwater exhibiting shallowing groundwater levels followed by declining groundwater levels (i.e., yellow points in panel a). (c-d) Comparison of average annual precipitation rates for the early-21st and late-20th centuries (i.e., early-21st-century average annual precipitation divided by late-20th-century average annual precipitation; values on x-axes) and 21st-century groundwater level trends (aquifer-scale trends calculated as the median of all monitoring well water level trends within the boundaries of an aquifer system; y-axis values). Each point represents one of our 542 aquifer systems. X-axis values of less than 1 indicate that average annual precipitation during the early-21st-century was lower than average annual precipitation during the late-20th-century. Panels c and d only display aquifer systems categorized as exhibiting shallowing followed by deepening groundwater level trends (i.e., only the yellow points in panel a are displayed in panels c and d). The x-axis values in panel c are based on the 'TerraClimate' dataset, whereas the x-axis values in panel d are based on the 'CHELSA' dataset (see text above for full references to these two precipitation data sources). Our analysis of precipitation time series demonstrates that the majority (79-85%) of aquifer systems shown here (i.e., groundwater levels became shallower during 1980-2000, but

have since become deeper) also received less precipitation in the early-21st-century than in the late-20th-century.

Supplementary Fig. 30. Comparison of trends in depth to groundwater and annual precipitation rates for 'orange points'. (a) Comparison of trends in depth to groundwater during two different time intervals (early-21st-century and late-20th-century). This panel presents identical data and categories as that displayed in main text Fig. 3 (see that figure caption for details). (b) An example of a time series in depth to groundwater exhibiting decelerated groundwater level declines (i.e., orange points in panel a). (c-d) Comparison of average annual precipitation rates for the early-21st and late-20th centuries (i.e., early-21st-century average annual precipitation divided by late-20th-century average annual precipitation; values on x-axes) and 21st-century groundwater level trends (aquifer-scale trends calculated as the median of all monitoring well water level trends within the boundaries of an aquifer system; y-axis values). Each point represents one of our 542 aquifer systems. X-axis values of less than 1 indicate that average annual precipitation during the early-21st-century was lower than average annual precipitation during the late-20th-century. Panels c and d only display aquifer systems categorized as exhibiting decelerated groundwater level declines (i.e., only the orange points in panel a are displayed in panels c and d). The x-axis values in panel c are based on the 'TerraClimate' dataset, whereas the x-axis values in panel d are based on the 'CHELSA' dataset (see text above for full references to these two precipitation data sources). Our analysis of precipitation time series demonstrates that the majority (70-79%) of aquifer systems categorized as exhibiting decelerated groundwater level deepening (i.e., groundwater levels became deeper during 1980-2000, and continued to become deeper but at a slower rate) also received less precipitation in the early-21st-century than in the late-20th-century.

Supplementary Fig. 31. Comparison of trends in depth to groundwater and annual precipitation rates for 'blue points'. (a) Comparison of trends in depth to groundwater during two different time intervals (early-21st-century and late-20th-century). This panel presents identical data and categories as that displayed in main text Fig. 3 (see that figure caption for details). (b) An example of a time series in depth to groundwater exhibiting recovering groundwater levels (i.e., blue points in panel a). (c-d) Comparison of average annual precipitation rates for the early-21st and late-20th centuries (i.e., early-21st-century average annual precipitation divided by late-20th-century average annual precipitation; values on x-axes) and 21st-century groundwater level trends (aquifer-scale trends calculated as the median of all monitoring well water level trends within the boundaries of an aquifer system; y-axis values). Each point represents one of our 542 aquifer systems. X-axis values of less than 1 indicate that average annual precipitation during the early-21st-century was lower than average annual precipitation during the late-20th-century. Panels c and d only display aquifer systems categorized as exhibiting recovering groundwater level trends (i.e., only the blue points in panel a are displayed in panels c and d). The x-axis values in panel c are based on the 'TerraClimate' dataset, whereas the x-axis values in panel d are based on the 'CHELSA' dataset (see text above for full references to these two precipitation data sources). Our analysis of precipitation time series demonstrates that the slight majority (56-62%) of aquifer systems categorized as exhibiting recovering groundwater levels (i.e., groundwater levels became deeper during 1980-2000, but have since become shallower) also received more precipitation in the early-21st-century than in the late-20th-century.

Supplementary Fig. 32. Comparison of trends in depth to groundwater and annual precipitation rates for ‘purple points’. (a) Comparison of trends in depth to groundwater during two different time intervals (early-21st-century and late-20th-century). This panel presents identical data and categories as that displayed in main text Fig. 3 (see that figure caption for details). (b) An example of a time series in depth to groundwater exhibiting continually shallowing groundwater levels (i.e., exemplifying purple points in panel a). (c-d) Comparison of average annual precipitation rates for the early-21st and late-20th centuries (i.e., early-21st-century average annual precipitation divided by late-20th-century average annual precipitation; values on x-axes) and 21st-century groundwater level trends (aquifer-scale trends calculated as the median of all monitoring well water level trends within the boundaries of an aquifer system; y-axis values). Each point represents one of our 542 aquifer systems. X-axis values of less than 1 indicate that average annual precipitation during the early-21st-century was lower than average annual precipitation during the late-20th-century. Panels c and d only display aquifer systems categorized as exhibiting continually shallowing groundwater levels (i.e., only the purple points in panel a are displayed in panels c and d). The x-axis values in panel c are based on the ‘TerraClimate’ dataset, whereas the x-axis values in panel d are based on the ‘CHELSA’ dataset (see text above for full references to these two precipitation data sources). Our analysis of precipitation time series demonstrates that the slight majority (55-58%) of aquifer systems categorized as exhibiting continually shallowing trends (i.e., groundwater levels became shallower during 1980-2000, and continued to become shallower) also received less precipitation in the early-21st-century than in the late-20th-century.

This particular comment motivated an analysis that led to new findings. We thank Reviewer 3 for recommending this to us. We have also added a new paragraph within the body of the main text to

highlight the outcomes of this new supplementary analysis; this new paragraph in the main text reads as follows:

“To test for a potential statistical relationship between accelerating groundwater-level declines and climate variability, we analysed precipitation rates over the past four decades (Supplementary Note 10). We show that the majority (>80%) of aquifer systems exhibiting accelerating groundwater level declines also experienced a decline in precipitation over time (i.e., lower average annual precipitation during the early-21st-century than in the late-20th-century). Declines in precipitation can cause groundwater levels to fall both via indirect impacts (e.g., increased groundwater abstractions during droughts) and also via direct impacts (e.g., reduced recharge rates during droughts; see ref.²⁷). Our finding—that early-21st-century precipitation rates were lower than in the late-20th-century in most aquifer systems exhibiting accelerating groundwater-level declines— highlights a potential link between decadal-scale climate variability and accelerating groundwater level declines. Accelerating groundwater-level declines, regardless of their potential drivers, are likely to also accelerate the consequences of those declines, including land subsidence^{12,13} and wells running dry¹⁷.”

7.Please make sure color scales used in figures are colorblind friendly.

Thank you; this is an important suggestion. We uploaded each of our figures into a Color Blindness Simulator to ensure the distinct colours can indeed be distinguished when a “Green-Weak/Deuteranomaly” color blindness simulator is run. Below we present the image before and after running the color blindness simulator (i.e., the image on the left is identical to the one we post in our manuscript, and the image on the right is the version after we run a “Green-Weak/Deuteranomaly” color blindness simulator (*the National Institutes of Health states that “Deuteranomaly is the most common type of red-green color blindness” (quoting <https://www.nei.nih.gov/learn-about-eye-health/eye-conditions-and-diseases/color-blindness/types-color-blindness>, accessed July 25, 2023)).

Main text Fig. 2 above

Main text Fig. 2 above (after applying the “Color Blindness Simulator”)

Main text Fig. 3 above

Main text Fig. 3 above (after applying the “Color Blindness Simulator”)

Main text Fig. 4 above

Main text Fig. 4 above (after applying the “Color Blindness Simulator”)

Reviewer Reports on the First Revision:

Referees' comments:

Referee #1 (Remarks to the Author):

I would like to thank the authors for their efforts in revising the paper. They have thoroughly addressed my initial concerns in both the response letter and the revised paper. In particular, their efforts to perform statistical time series analysis over scenarios of time intervals has greatly strengthened the study (e.g., 3, 5, and 8-year intervals), as well as the authors' explanation that only a small proportion of monitoring wells have multiple years of data. I also appreciate that the authors have further clarified the novelty of their study. They now explain that a novelty of their study is that it enables us to evaluate the locations where level declines have either reversed or slowed, in addition to continued rapid lowering (Is a more neutral title warranted in light of this?). It is also great to see that the well level data will be released with the paper, so that future research and decision-making can build on this work.

A few remaining minor points:

- I appreciate that the authors have clarified the number of wells with different time intervals of data available (e.g., the table provided in Update One of Two, included as the caption to Figure S11). However, the table in the response letter lists "2000-2022" as the time-period of consideration. But the abstract says "four decades" are analyzed. Please make sure the time-period of the analysis is consistent and clear throughout the paper.

- The authors have done a great job in curating and analyzing the available well data, but there is still selection bias in the data. The data that is analyzed in this study is not a random selection of wells around the world. This is a selected sample of monitored wells, further stratified for wells with multiple time periods of data. There is information missing for locations that have not yet drilled wells, or for location with wells that are not monitored. Presumably the wells that are being monitored are systematically different to the wells that are not monitored. I understand that we are not able to quantify these differences because this is a data selection problem. However, I think it is important for the authors to address this in the main body of the text and explain that sample selection means that we are only able to draw conclusions about groundwater levels from the locations with available well data. This means that we are unable to draw conclusions about overall global groundwater levels from these results, since they are not based on a randomly stratified sample.

Referee #2 (Remarks to the Author):

General comments:

I thank the author for their comprehensive response to all three reviewers. I am happy that most of my comments, and those of the other reviewers, have been taken on board and I think the result is a considerably improved manuscript. In particular, I appreciate the much clearer description of the methodology used to arrive at the final dataset and the useful sensitivity analysis of different filtering criteria. Also, the addition of a section focusing specifically on slowing and reversing groundwater-level decline is very welcome. I remain hugely impressed by the scale and quantity of work that the study represents. However, I still have some concerns outlined below.

My main concern is the title, I question how well your current title reflects the reality of your results? I realise you have provided some justification for the existing title but I remain unconvinced that it accurately reflects your findings. I would urge the authors to try and recognise the nuance in the title of the manuscript (as they have now done very effectively elsewhere), as it could be construed as misleading at present.

While I accept that decline is the dominant story (i.e. 69% of aquifer systems are still experiencing groundwater level decline), my main issue is with the use of the term “accelerating” in the title. I would urge the authors to consider dropping this term. In what areas is groundwater level decline actually accelerating? According to the table in Supplementary Note 9 only 30.1% of aquifer systems are seeing groundwater level declines at a faster rate. It would be good to see the numbers in the abstract for rates of decline based on aridity and land use which would link the abstract better to your title, e.g. a more concise description of the results presented in the section entitled, “Groundwater declines in cultivated drylands” and Supplementary note 11. I am also not clear on what “rapid” means here, I think it is superfluous. Furthermore, you state in the main text that, “Rapidly deepening groundwater levels (faster than 0.5 m/year) are found in 11%, 24% and 8% of aquifers in climate zones classified as hyper-arid, arid, and semi-arid, respectively”, which represent a relatively small proportion overall.

I also have some more minor comments in response to the reviewer rebuttals which I outline below.

Response to reviewer 1:

Supplementary Fig. 11: Large parts of South Asia (e.g. west central, eastern and western areas) appear to fall out of your analysis when the most stringent criteria apply. What impact might this have, given the importance of this region from a groundwater perspective?

Supplementary Fig. 12 and 13: It would be interesting to label the outliers on these plots, so we can get a feeling for areas where there are significant differences between different filtering criteria.

Supplementary Fig. 4 and Supplementary Fig. 35: It would also be helpful to present a figure showing how many of the wells have complete time series for each period, what percentage of wells have missing data in each year for your 8-, 5- and 3-year time series. Perhaps I misunderstand Supplementary Fig. 35, but while it goes some way towards this, I don't think it quite addressed the specific point about missing data in each year.

Supplementary page 915: Previous studies have looked at depth dependant groundwater level trends, the most notable to my knowledge is:

Malakar, P., Mukherjee, A., Bhanja, S.N., Ganguly, A.R., Ray, R.K., Zahid, A., Sarkar, S., Saha, D. and Chattopadhyay, S., 2021. Three decades of depth-dependent groundwater response to climate variability and human regime in the transboundary Indus-Ganges-Brahmaputra-Meghna mega river basin aquifers. *Advances in Water Resources*, 149, p.103856.

Supplementary Fig. 282: It would be useful to provide a map of the global distribution of these points, particularly given that previous results are available for India and it would be good to be able to see how your results from the Indian data within your dataset compare to the paper above.

Response to reviewer 2:

In the abstract the authors state: “...groundwater level declines have accelerated over the past four decades in a disproportionate share of the world's regional aquifers”. I would suggest providing the actual numbers here. I appreciate your definition of the term “disproportionate” later in the text but I think the use of the term here could still be misleading (particularly given my comments about the title above).

I appreciate the definition that you have added on lines 132 – 147, however when it is followed so closely by the following text it somewhat muddies the water:

“Specifically, in half (49%) of the 542 aquifer systems in our analysis, groundwater level declines have decelerated (i.e., slowed; orange in Fig. 3; 20%) or reversed (blue in Fig. 3; 16%), or groundwater levels have continued to rise (purple in Fig. 3; 13%).”

Even with your improved definition of terminology, it's hard to make the case that 51% of aquifers represents a “disproportionate” amount.

Supplementary note 22: Similar data are available for Pakistan and India. It's surprising these data are not included (even though they might not represent “completion events” in the same way as you have defined here) given their prominence as areas of large-scale groundwater extraction for irrigation.

Referee #3 (Remarks to the Author):

Authors have addressed all the queries raised and the revised manuscript looks significantly improved and recommend for acceptance

Author Rebuttals to First Revision:

Reply to Reviewers (October 17, 2023)

(reviewer comments are presented in **black text**; our replies are presented in blue text)

Referee #1 (Remarks to the Author):

I would like to thank the authors for their efforts in revising the paper. They have thoroughly addressed my initial concerns in both the response letter and the revised paper. In particular, their efforts to perform statistical time series analysis over scenarios of time intervals has greatly strengthened the study (e.g., 3, 5, and 8-year intervals), as well as the authors' explanation that only a small proportion of monitoring wells have multiple years of data. I also appreciate that the authors have further clarified the novelty of their study.

We thank Referee #1 for their helpful recommendations. Thanks to their comments, our work is much improved.

They now explain that a novelty of their study is that it enables us to evaluate the locations where level declines have either reversed or slowed, in addition to continued rapid lowering (Is a more neutral title warranted in light of this?).

Thank you. Referee #2 has provided a similar recommendation to revise our title; we agree with both Referee #1 and #2 and we have revised our manuscript title accordingly to: *"Rapid groundwater declines in many aquifers globally but cases of recovery"*

It is also great to see that the well level data will be released with the paper, so that future research and decision-making can build on this work.

Thank you. We agree, and have made every effort to secure written permission from database managers to post these annual groundwater level data.

A few remaining minor points:

- I appreciate that the authors have clarified the number of wells with different time intervals of data available (e.g., the table provided in Update One of Two, included as the caption to Figure S11). However, the table in the response letter lists "2000-2022" as the time-period of consideration. But the abstract says "four decades" are analyzed. Please make sure the time-period of the analysis is consistent and clear throughout the paper.

Thank you for recommending we ensure clarity with respect to the time intervals examined for different analyses. We have, for example, revised our manuscript's abstract to include a new sentence that specifies the 21st century time interval as the time period of that analysis; this new

sentence reads as: *“We show that rapid groundwater level declines (>0.5 m/year) are widespread in the 21st-century, especially in dry regions with extensive croplands.”*

- The authors have done a great job in curating and analyzing the available well data, but there is still selection bias in the data. The data that is analyzed in this study is not a random selection of wells around the world. This is a selected sample of monitored wells, further stratified for wells with multiple time periods of data. There is information missing for locations that have not yet drilled wells, or for location with wells that are not monitored. Presumably the wells that are being monitored are systematically different to the wells that are not monitored. I understand that we are not able to quantify these differences because this is a data selection problem. However, I think it is important for the authors to address this in the main body of the text and explain that sample selection means that we are only able to draw conclusions about groundwater levels from the locations with available well data. This means that we are unable to draw conclusions about overall global groundwater levels from these results, since they are not based on a randomly stratified sample.

Thank you. We agree that this is an important point. We have added the following sentence within the main text of our manuscript: *“Further, analysed monitoring wells do not represent a randomized sample of global wells, and we are only able to analyse groundwater level trends where monitoring data are available.”*

Referee #2 (Remarks to the Author):

General comments:

I thank the author for their comprehensive response to all three reviewers. I am happy that most of my comments, and those of the other reviewers, have been taken on board and I think the result is a considerably improved manuscript. In particular, I appreciate the much clearer description of the methodology used to arrive at the final dataset and the useful sensitivity analysis of different filtering criteria. Also, the addition of a section focusing specifically on slowing and reversing groundwater-level decline is very welcome. I remain hugely impressed by the scale and quantity of work that the study represents.

We are most grateful to Referee #2. Our paper improved substantially from our original submission due in no small part to the wise recommendations provided by Referee #2. Thank you.

However, I still have some concerns outlined below.

My main concern is the title, I question how well your current title reflects the reality of your results? I realise you have provided some justification for the existing title but I remain unconvinced that it accurately reflects your findings. I would urge the authors to try and recognise the nuance in the title of the manuscript (as they have now done very effectively elsewhere), as it could be construed as misleading at present.

While I accept that decline is the dominant story (i.e. 69% of aquifer systems are still experiencing groundwater level decline), my main issue is with the use of the term “accelerating” in the title. I would urge the authors to consider dropping this term. In what areas is groundwater level decline actually accelerating? According to the table in Supplementary Note 9 only 30.1% of aquifer systems are seeing groundwater level declines at a faster rate.

Thank you. We agree; as recommended we have dropped “accelerating” from the title. Referee #1 also recommends we revise our title, and we have done so. The revised title now embeds the nuance Referee #2 highlighted in their first set of comments; the revised title is: “*Rapid groundwater declines in many aquifers globally but cases of recovery*”

It would be good to see the numbers in the abstract for rates of decline based on aridity and land use which would link the abstract better to your title, e.g. a more concise description of the results presented in the section entitled, “Groundwater declines in cultivated drylands” and Supplementary note 11. I am also not clear on what “rapid” means here, I think it is superfluous. Furthermore, you state in the main text that, “Rapidly deepening groundwater levels (faster than 0.5 m/year) are found in 11%, 24% and 8% of aquifers in climate zones classified as hyper-arid, arid, and semi-arid, respectively”, which represent a relatively small proportion overall.

Thank you for this helpful comment. We agree that adding a clear and quantitative definition of 'rapid' to the abstract is helpful; we also agree that including a sentence pertaining to cultivated drylands in the abstract is appropriate. This recommendation led us to revise our abstract by adding this sentence: "*We show that rapid groundwater level declines (>0.5 m/year) are widespread in the 21st-century, especially in dry regions with extensive croplands.*" We specifically include the numeric definition of what we mean by 'rapid', to communicate our definition of 'rapid' in clear and quantitative terms. This revision improved our manuscript by bringing into our abstract a statement pertaining to content from the section entitled, "Groundwater declines in cultivated drylands"; we thank the reviewer for their good and helpful recommendation.

I also have some more minor comments in response to the reviewer rebuttals which I outline below.

Response to reviewer 1:

Supplementary Fig. 11: Large parts of South Asia (e.g. west central, eastern and western areas) appear to fall out of your analysis when the most stringent criteria apply. What impact might this have, given the importance of this region from a groundwater perspective?

We thank the reviewer for this helpful point. It is clear to us that the reviewer carefully examined the lengthy supplementary information linked to this work. We have added a new paragraph just above Supplementary Fig. 11 to draw attention to the spatial patterns of the monitoring wells that meet different filtering criteria. The new paragraph reads as follows:

"There are noteworthy differences in the spatial patterns and densities of monitoring wells meeting these three different criteria (i.e., minimum temporal offsets between the earliest and most-recent 21st century measurements in the time series of (i) 3 years, (ii) 5 years, or (iii) 8 years). For example, there are areas in South Asia (specifically, west central and eastern portions of South Asia) where many monitoring wells have adequate data for analyses of groundwater level trends over 5-7 years, but not for >8 year-long time intervals (i.e., red points in the following figure). Groundwater supplies play a key role in this region. Because the analyses on our main text are based solely on monitoring wells where the minimum temporal offset between the earliest and most-recent 21st century measurement is at least 8 years, many monitoring wells in these regions are excluded from our statistical analyses presented in the main text. However, our comparison of aquifer-scale trends in depth to groundwater using different thresholds for the minimum temporal offset between the earliest and most-recent 21st century measurement (Supplementary Figs. 12 and 13) suggest that, for the great majority of our studied aquifer systems, median aquifer-scale Theil-Sen slopes are largely unchanged when we reduce the stringency of our filtering criteria."

Supplementary Fig. 12 and 13: It would be interesting to label the outliers on these plots, so we can get a feeling for areas where there are significant differences between different filtering criteria.

Thank you for this good suggestion. We agree and have updated both of these figures. The original figure is shown on the left and the revised figures are shown on the right:

Supplementary Fig. 4 and Supplementary Fig. 35: It would also be helpful to present a figure showing how many of the wells have complete time series for each period, what percentage of wells have missing data in each year for your 8-, 5- and 3-san time series.

Perhaps I misunderstand Supplementary Fig. 35, but while it goes some way towards this, I don't think it quite addressed the specific point about missing data in each year.

Thank you for this helpful comment. We agree that highlighting the proportion of our monitoring wells that are incomplete* (defined as monitoring wells where one or more calendar years between the earliest and most-recent groundwater level measurement lacked data) is critical to provide a thorough and transparent sense of the quality of the underlying data that we analyse. We thank the reviewer for their recommendation. We feel that their recommendation is sufficiently important to add a new statement within the main text itself (instead of just the supplementary); specifically, we have added the following new sentence to the main text methods section entitled 'Limitations' to provide the reader with a clear sense that some of the time series we analyse are incomplete:

"Additionally, ~41% of the analysed monitoring wells have discontinuous time series of annual groundwater levels (where 'discontinuous' time series are defined as those lacking a groundwater level measurement for at least one of the calendar years that lie between the earliest and most-recent 21st century groundwater level measurements; for an example of a discontinuity in an annual groundwater level time series see Supplementary Fig. 3c)."

Additionally, we added a new sentence to our Supplementary Information next to Supplementary Fig. 3 that provides an example of a discontinuous groundwater level time series. Specifically, we added the following sentence to the Supplementary Fig. 3 caption (underlined text represents the newly added text):

“The

Supplementary Fig. 3. Calculation of annual median groundwater levels from well water level time series. (a) Location of a monitoring well in northern Iran (monitoring well is located within the boundaries of the West Qazvin Plain aquifer system). (b) Time series of groundwater levels recorded at this monitoring well. Each grey circle represents one recorded well water level measurements (units on y-axis are meters below the land surface). X-axis values represent the measurement date. (c) Calculation of annual median groundwater levels. Each black circle represents the median well water level for a given calendar year. The original time series of well water level measurements are presented as grey circles displayed behind the black circles. The black circles are equally spaced (one point per calendar year) with the exception of missing values (for calendar years during which no well water level measurements were made). The figure exemplifies the discontinuous nature of 21st-century groundwater level time series evident in (see gap in groundwater level measurements for the year 2003, for sample, in panels b and c). Such discontinuities occur in 41% of the monitoring wells presented in main text Fig. 1.”

Further, we made a third update to our Supplementary Information in response to this comment. We have added a new statement next to the Supplementary Figure that the reviewer highlights in their comment. Specifically, we added the following to the caption of Supplementary Fig. 3: “We stress that many of these monitoring wells do not have a complete time series of annual groundwater levels (that is, one or more calendar years between the earliest and most-recent calendar years in the time series lack a groundwater level measurement; ~41% of the monitoring wells have such discontinuities.”

Supplementary page 915: Previous studies have looked at depth dependant groundwater level trends, the most notable to my knowledge is:

Malakar, P., Mukherjee, A., Bhanja, S.N., Ganguly, A.R., Ray, R.K., Zahid, A., Sarkar, S., Saha, D. and Chattopadhyay, S., 2021. Three decades of depth-dependent groundwater response to climate variability and human regime in the transboundary Indus-Ganges-Brahmaputra-Meghna mega river basin aquifers. *Advances in Water Resources*, 149, p.103856.

We added this citation to our main text methods following the statement: “We stress that groundwater-level trends may differ between deeper and shallower wells (e.g., ref. Malakar et al. 2021)....”

Supplementary Fig. 282: It would be useful to provide a map of the global distribution of these points, particularly given that previous results are available for India and it would be good to be able to see how your results from the Indian data within your dataset compare to the paper above.

This is an excellent recommendation. We agree. We have created a map and updated our Supplementary Fig. 282 to include this analysis of the global distribution of these aquifer systems. The revised figure appears as follows:

Regarding the reviewer’s point regarding data for South Asia, we did not find total well depth data within the dataset that we downloaded from <https://www.indiawatertool.in> (at the time that we downloaded these data (October 1, 2020) from <https://www.indiawatertool.in> as detailed in Supplementary Table 1). We understand that other studies have found total well depth information, thus enabling an analysis of shallow versus deep monitoring wells and their trends in depth to groundwater. In an effort to include specific reference to (i) the publication mentioned by the reviewer in their comment, and also (ii) acknowledgement that our analyses presented in this Supplementary Note 19.1 can only focus on monitoring wells where we were able to

compile well depth data, we have added the following new paragraph to our Supplementary Information:

*“Previous works have compared trends in depth to groundwater among shallow versus deep monitoring wells in regions that are not included in the analysis presented in the Supplementary Fig. 282 shown above (e.g., ref.¹³⁵⁹: Malakar, P., Mukherjee, A., Bhanja, S.N., Ganguly, A.R., Ray, R.K., Zahid, A., Sarkar, S., Saha, D., Chattopadhyay, S. Three decades of depth-dependent groundwater response to climate variability and human regime in the transboundary Indus-Ganges-Brahmaputra-Meghna mega river basin aquifers. *Advances in Water Resources*, 149, 103856 (2021)). We acknowledge that our data compilation may not include well depth data in all possible cases, and that the analyses presented here focus only on areas where we were able to compile well depth data (in addition to groundwater level time series).”*

Response to reviewer 2:

In the abstract the authors state: “...groundwater level declines have accelerated over the past four decades in a disproportionate share of the world’s regional aquifers”. I would suggest providing the actual numbers here. I appreciate your definition of the term “disproportionate” later in the text but I think the use of the term here could still be misleading (particularly given my comments about the title above).

Thank you. We agree. We revised our abstract (which previously read as *“Critically, we also show that groundwater level declines have accelerated over the past four decades in a disproportionate share of the world’s regional aquifers.”*) to now read as *“Critically, we also show that groundwater level declines have accelerated over the past four decades in 30% of the world’s regional aquifers.”* (underline specifies change made to avoid ‘disproportionate’)

I appreciate the definition that you have added on lines 132 – 147, however when it is followed so closely by the following text it somewhat muddies the water:

“Specifically, in half (49%) of the 542 aquifer systems in our analysis, groundwater level declines have decelerated (i.e., slowed; orange in Fig. 3; 20%) or reversed (blue in Fig. 3; 16%), or groundwater levels have continued to rise (purple in Fig. 3; 13%).”

Even with your improved definition of terminology, it’s hard to make the case that 51% of aquifers represents a “disproportionate” amount.

We agree, and thank the reviewer for pointing this out. In addition to the change we have made to the abstract to avoid the term ‘disproportionate’, we made two more revisions such that the term ‘disproportionate’ no longer appears in the main text:

1. We revised our previous text (which read as *“groundwater-level declines have accelerated in a disproportionate fraction of our aquifer systems.”*) to now read as follows: *“groundwater-level declines have accelerated in a substantial share of the analysed aquifer systems.”* (underline specifies change made to avoid ‘disproportionate’).

2. We made yet another change to avoid the term 'disproportionate'. In the final section of our main text ('Depleting and recovering groundwater resources') we revised our main text (which previously read as: "...groundwater declines are accelerating in a disproportionate share of aquifer systems") to instead read as follows in this revised submission: "*groundwater declines are accelerating in many aquifer systems around the world*"

Supplementary note 22: Similar data are available for Pakistan and India. It's surprising these data are not included (even though they might not represent "completion events" in the same way as you have defined here) given their prominence as areas of large-scale groundwater extraction for irrigation.

Thank you. We agree with the reviewer that these regions are prominent groundwater extraction areas, and are very important to understanding groundwater availability. We were therefore pleased to be able to analyze groundwater level trends in these areas as part of this research. We are not aware of national scale well completion data for these regions, despite inquiring about the potential existence of such data over several years as part of a previous project on global well completion (i.e., ref.¹⁷ in this work). Though we could not identify a downloadable and readily accessible well completion dataset for these regions, we are aware of census data from important works that have been published for parts of these regions. For example, increases over time in the number of groundwater structures regional research has been demonstrated by (i) Fig. 7 within Laghari, A. N., Vanham, D., & Rauch, W. The Indus basin in the framework of current and future water resources management. *Hydrology and Earth System Sciences*, 16, 1063-1083 (2012), and (ii) Fig. 1 within Mukherji, A., Rawat, S., & Shah, T. Major insights from India's minor irrigation censuses: 1986-87 to 2006-07. *Economic and Political Weekly*, 115-124 (2013).

Though we could not access well completion data for these regions (to report in a methodologically consistent way as presented in the supplementary information section referred to by the Reviewer), we agree that it is important to highlight previous research that has found increases in groundwater structures over time in this region. Therefore, we have added a sentence to our main text methods citing these works (and another publication focused on well completion in the Qazvin Plain of Iran); the new sentence (in main text methods section entitled "Limitations") reads as follows:

"Many of the aquifer systems exhibiting rapid groundwater level declines are being accessed by wells, as evidenced by recorded well completion events throughout the early 21st century (Supplementary Note 22; data described in refs.^{17,1369,1370,1371}) and by regional-scale research^{108,1372,1373}."

Referee #3 (Remarks to the Author):

Authors have addressed all the queries raised and the revised manuscript looks significantly improved and recommend for acceptance

We thank Referee #3. The recommendations they provided in their review led to numerous changes to our work (reflected in the revised manuscript) that each represented, in our view, improvements to the manuscript. Thank you.